# EdgeMask-HGNN: Learning to Sparsify Hypergraphs for Scalable Node Classification in Hypergraph Neural Networks

## Abstract

Hypergraph Neural Networks (HGNNs) have achieved remarkable performance in various learning tasks involving hypergraphs— a data model for higher-order relationships across diverse domains and applications. However, the scalability of HGNNs is limited by the computational and memory demands incurred by dense hypergraph structures. Existing unsupervised sparsifiers address the scalability issue but sacrifice downstream predictive performance. To address this, we propose **EdgeMask-HGNN**, a novel framework that introduces a learnable, task-aware sparsification mechanism to reduce the hypergraph size while preserving predictive performance. EdgeMask-HGNN offers two distinct masking: a fine-grained node-hyperedge masking and a coarse-grained hyperedge-level masking, both trained end-to-end using supervision from the downstream task. We provide theoretical analysis showing that our approach (i) yields stable model outputs under stochastic masking, and (ii) ensures convergence of retention probabilities under gradient descent. Extensive experiments on multiple node classification benchmarks demonstrate that EdgeMask-HGNN reduces or maintains memory usage on both small- and large-scale hypergraphs without sacrificing accuracy, and in some cases outperforms HGNNs trained on full hypergraphs. EdgeMask-HGNN also consistently outperforms unsupervised sparsification baselines such as random, degree-based, and spectral sparsification.

## 1 Introduction

Hypergraph Neural Networks (HGNNs) have emerged as powerful tools for learning on hypergraphs, where a single edge can connect multiple nodes (beyond pairwise connections), unlike traditional graphs. There are numerous real-world instances of such relations involving multiple entities: set of researchers collaborating on a paper (Han et al., 2009), set of products purchased in a shopping cart (Xia et al., 2021), group of legislators co-sponsoring a bill (Benson et al., 2018), or set of molecules participating together in biological processes (Gaudelet et al., 2018). With the rapid growth of data, the scale of real-world hypergraphs has also expanded dramatically. For instance, the open-source bibliographic database OpenAlex has $\sim$209 million scholarly works (hyperedges) from $\sim$13 million authors (node) across more than 450 topics[1]. A wide range of learning problems arises in this setting, from node classification (Yadati et al., 2019; Duta et al., 2023) to node clustering (Chodrow et al., 2021) to hyperlink prediction (Yadati et al., 2020). HGNNs have proven to be effective for addressing these tasks, demonstrating strong empirical performance across diverse domains such as social networks (Wang et al., 2018), recommendation systems (Wang et al., 2021), and bioinformatics (Deng et al., 2024).

Despite the success of Hypergraph neural networks, they incur high computational and memory complexity in terms of scaling to large-scale hypergraphs due to dense incidence structures. These costs arise from sparse-dense matrix multiplications during message-passing and the storage of intermediate activations during forward and backward passes. *Sparsifying the hypergraph by pruning redundant node–hyperedge links offers a promising solution by significantly reducing computational and memory overhead while retaining key structural information.*

---

[1] https://en.wikipedia.org/wiki/OpenAlex

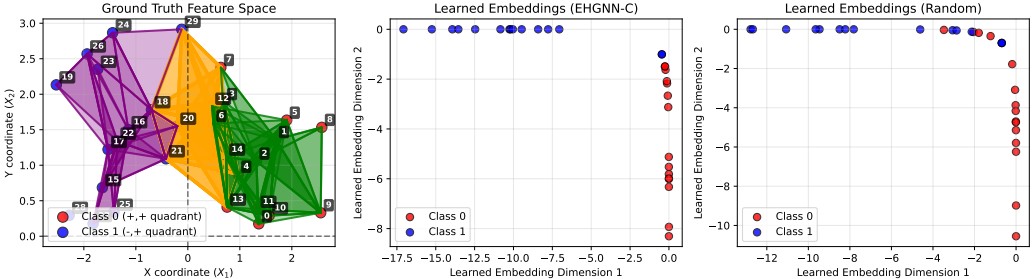

Figure 1: A synthetic 3-uniform hypergraph (#nodes=30, #edges=134) with their (x,y)-coordinates in $\mathbb{R}^2$ as ground truth node features (left). 50% hyperedges were pruned to learn node embeddings via EdgeMask-HGNN (middle) and via Random sparsification (right). The node embeddings learned by EdgeMask-HGNN are well separable and more aligned with the true class separation.

A naïve way to sparsify would be dropping hyperedges uniformly at random, often termed as *random sparsification* in graph literature (Das et al., 2025). However, this method does not exploit the structure of the hypergraph. Inspired by graph literature, *degree-proportionate sampling* (Leskovec & Faloutsos, 2006) samples nodes based on their degree in the hypergraph, constructing a sparse subhypergraph. However, this method may not preserve the spectral properties of the hypergraph and may remove nodes that are important for message passing for downstream tasks. *spectral sparsifiers* (Soma & Yoshida, 2019) sample nodes based on hypergraph effective resistance to preserve eigenvalues and eigenvectors of the original hypergraph. However, this approach may preserve task-irrelevant hyperedges since it does not exploit label information.

**Our contributions.** (I) In this paper, we propose EdgeMask-HGNN to fill this gap of a lack of task-specific sparsification in HGNNs. EdgeMask-HGNN offers two distinct masking strategies: a fine-grained node–hyperedge masking and a coarse-grained hyperedge-level masking, both trained end-to-end using supervision from the downstream task to facilitate task-aware sparsification. Unlike unsupervised methods (e.g., random, degree-based, and spectral sparsification) that rely on fixed hypergraph structures sampled a priori, EdgeMask-HGNN dynamically selects a subset of hyperedges during training that helps the downstream HGNN perform better on the learning task. Figure 1 highlights the importance of leveraging task-specific signal during sparsification: under the same sparsification budget, embeddings learned by EdgeMask-HGNN yield better class separation than the random sparsifier that does not utilize label information.

(II) We theoretically analyze EdgeMask-HGNN to show that it has a probabilistic interpretation in terms of sampling the mode of the distribution over sparse subhypergraphs. We also show that the sparsification (a) produces stable model outputs, and (b) has an $\mathcal{O}(1/\epsilon)$ convergence rate to an $\epsilon$-confidence threshold on the learned retention probabilities.

(III) Empirical evaluation and analysis show that EdgeMask-HGNN is more effective than alternative sparsifiers such as random, degree distribution-based, and effective resistance sparsifiers, while also being competitive or superior in terms of memory and runtime. Compared to training on the full hypergraph, EdgeMask-HGNN sacrifices little accuracy and often achieves better predictive performance, while maintaining or reducing memory usage on both small- and large-scale hypergraphs.

## 2 RELATED WORKS

**Hypergraph Neural Networks (HGNNs).** Feng et al. (2019) proposed HGNN by generalizing spectral graph convolutions to hypergraphs through the hypergraph Laplacian. Subsequently, several variants have been proposed, such as HyperGCN (Yadati et al., 2019), and HNHN (Dong et al., 2020). HyperGCN reformulates hypergraph convolution using clique expansion, and HNHN introduces attention mechanisms for hyperedge importance. *spatial* HGNNs typically define two-stage message aggregation: node to hyperedge and hyperedge to node. HyperSAGE (Arya et al., 2020), HGNN+ (Gao et al., 2022), and UniGCN (Huang & Yang, 2021) belong to this category. Recently, Chien et al. (2021) showed that propagation rules of many existing spatial HGNNs can be represented as a composition of two multiset functions, and proposed two different multiset encoding functions: DeepSets and SetTransformer. In addition, several recent frameworks, such as

Wang et al. (2025); Saxena et al. (2024), focus on general and expressive message-passing designs for hypergraphs. Our work targets resource-efficient learning by identifying and preserving only task-relevant hyperedges, making it a practical augmentation for such general-purpose designs. Interested readers may refer to recent surveys for a more in-depth discussion on HGNNs (Antelmi et al., 2023; Kim et al., 2024).

**Unsupervised and Spectral Hypergraph Sparsification.** Graph sparsification has a long legacy, starting with Benczúr & Karger (1996) for preserving cuts via edge sampling. Subsequently, significant progress was made by Spielman & Teng (2011) and Spielman & Srivastava (2008), who introduced spectral sparsification via effective resistance. These methods ensure that a small subset of edges preserves global graph properties. Among earlier works on hypergraph sparsification, Deveci et al. (2013) proposed hyperedge sampling heuristics to reduce hyperedges while preserving cut structure for hypergraph partitioning tasks. Going beyond heuristics, Soma & Yoshida (2019) extended spectral sparsification to hypergraphs, defining a nonlinear Laplacian quadratic form and constructing $\epsilon$-spectral sparsifiers of size $\mathcal{O}(n^3/\epsilon^2)$ in polynomial time. Subsequent works such as Bansal et al. (2019); Kapralov et al. (2021); Lee (2023) attempted to improve this bound. Most recently, Kapralov et al. (2022) proved a significant result showing that it is possible to obtain an $\epsilon$-spectral sparsifier of linear size. Unlike EdgeMask-HGNN, these methods are unsupervised in nature and do not incorporate node labels into their sparsification decision. Since sparsification is done a priori to training, they are downstream task-unaware. Hence, they may not be suitable for representation learning tasks on hypergraphs that require preserving task-relevant hyperedges.

**Sparse GNN training.** Several works study sparsification of neural network parameters, such as the unified lottery ticket hypothesis for GNNs Chen et al. (2021); Hui et al. (2023) and graph gradual pruning Liu et al. (2023). These methods prune weights in the GNN (and occasionally graph edges) to reduce parameter count and training cost, with the primary goal of model compression. Our work is complementary: we keep the HGNN parameters dense and instead learn a structural sparsifier that selectively retains a subset of node–hyperedge incidences. Thus, while prior work focuses on sparsifying the neural network, our method sparsifies the hypergraph structure itself to improve efficiency during message passing.

## 3 PROBLEM STATEMENT

We consider the semi-supervised node classification task on a hypergraph $\mathcal{H} \triangleq (V, E, \boldsymbol{X}) \equiv (\boldsymbol{H}, \boldsymbol{X})$, where $V$ is the set of nodes, $E$ is the set of hyperedges, $\boldsymbol{H} \in \{0,1\}^{|V|\times|E|}$ is the incidence matrix of $\mathcal{H}$, and $\boldsymbol{X} \in \mathbb{R}^{n \times F}$ is the node feature matrix. A hyperedge $e \in E$ is a subset of $V$, i.e., $H_{v,e} = 1$ if $v \in e$, or 0 otherwise. Let $V_L, V_U$ be the set of labeled and unlabeled nodes, respectively, and $\boldsymbol{y}_L \in \{1, \ldots, C\}^{|V_L|}$ be the label vector for the labeled nodes. The goal is to learn a *model* $f_{\theta^*} : V_U \to \{1, \ldots, C\}$ that predicts labels for nodes in $V_U$ based on the labels $\boldsymbol{y}_L$ of the labeled nodes by minimizing a loss function $\mathcal{L}_{\text{task}}$ (e.g. cross-entropy): $\theta^* = \arg\min_\theta \mathcal{L}_{\text{task}}(f_\theta(\boldsymbol{H}, \boldsymbol{X}), \boldsymbol{y}_L)$.

Given a budget, the goal of supervised sparsification is to construct a sparse hypergraph $\hat{\mathcal{H}} = (\tilde{\boldsymbol{H}}, \boldsymbol{X})$ such that the prediction accuracy of the downstream task can be preserved or even improved by using $\hat{\mathcal{H}}$ (instead of $\mathcal{H}$) as input. This can be formulated as the following optimization problem:

$$\theta^* = \arg\min_\theta \mathcal{L}_{\text{task}}(f_\theta(\tilde{\boldsymbol{H}}, \boldsymbol{X}), \boldsymbol{y}_L),$$

where $\tilde{\boldsymbol{H}}$ is the masked incidence matrix produced by a learnable sparsification module. The downstream task loss could be any classification loss, e.g., the cross-entropy loss:

$$\mathcal{L}_{\text{CE}} = -\frac{1}{|V_L|} \sum_{v \in V_L} \sum_{c=1}^{|C|} \boldsymbol{Y}_{vc} \log \hat{\boldsymbol{Y}}_{vc},$$

where $Y_{vc}$ indicates the true probability of node $v$ belonging to class $c \in C$, and the predicted probabilities expressed as $\hat{\boldsymbol{Y}} = f_\theta(\tilde{\boldsymbol{H}}, \boldsymbol{X})$.

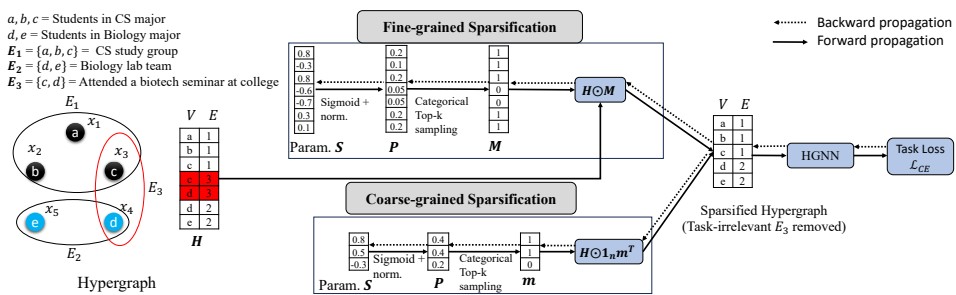

Figure 2: EdgeMask-HGNN on a small example where a task-irrelevant edge (red) is pruned.

Due to the uniqueness of hypergraphs, the budget constraint could be in the form of # incidence-pairs or # hyperedges. Let $k$ denotes the #node-hyperedge connections retained after sparsification. As edges in a hypergraph may contain more than 2 nodes, we shall denote by $\kappa$ the corresponding #hyperedges retained after sparsification. We denote the # node-hyperedge connections before sparsification by $t = \sum_{e \in \mathcal{H}} |e|$.

# 4 EDGEMASK-HGNN: LEARNABLE HYPERGRAPH SPARSIFICATION

In this section, we introduce EdgeMask-HGNN, a task-aware, learnable sparsification framework for hypergraph neural networks. The main idea is to devise a differentiable module that can learn to selectively mask hyperedges based on their relevance to the downstream learning task. Unlike prior hypergraph pruning methods that use fixed heuristics, EdgeMask-HGNN is trained end-to-end, allowing the model to jointly optimize both the hypergraph structure and parameters of HGNNs. Figure 2 illustrates the schematic of EdgeMask-HGNN with an example.

Since hyperedges may contain more than two nodes, two types of learnable masking strategies emerge: *Incidence-level Masking* and *Edge-Level Masking*. Incidence-level Masking learns a score for each (node, hyperedge) pair in the hypergraph, whereas Edge-Level Masking learns a score for each hyperedge. The scores are used to compute the sparsified hypergraph, which in turn is used to train the downstream HGNN. Based on the task loss of the downstream HGNN, the gradients are backpropagated down to compute the updated incidence-level or edge-level scores.

## 4.1 FINE-GRAINED INCIDENCE-LEVEL MASKING (EHGNN-F)

Fine-grained incidence-level masking focuses on learning a soft selection probability for individual node–edge connections so as to achieve a fine-grained control over the sparsified hypergraph structure. For each incidence pair $(v, e)$, we maintain a learnable score parameter $s_{v,e}$, which is converted to a soft selection probability as

$$p_{v,e} = \sigma(s_{v,e}) / \sum_{v,e} \sigma(s_{v,e}), \tag{1}$$

where $\sigma$ is the sigmoid function. $\boldsymbol{p} = (p_{v,e})$ can be interpreted as the marginal probabilities of a categorical distribution over all incidence entries, from which we aim to sample a prescribed number of active pairs, denoted by constraint $k$. In other words, if the sparsifier samples $(v, e)$, the hard binary mask $\hat{m}_{v,e} = 1$ and 0, otherwise. Mathematically,

$$\hat{\boldsymbol{m}} \sim \text{Top-k Categorical}(\boldsymbol{p}), \quad \sum_{v,e} \hat{m}_{v,e} = k. \tag{2}$$

This sampling process is not differentiable during the backward pass. To enable differentiable learning, we apply a straight-through estimator (Bengio et al., 2013). During the forward pass, we compute $p_{v,e}$ for each node–edge pair and use top-$k$ sampling to generate a hard binary mask $\hat{m}_{v,e}$, where only the top-$k$ values are retained. During the backward pass, we define the final mask as:

$$m_{v,e} = \text{stop\_grad}(\hat{m}_{v,e}) + p_{v,e} - \text{stop\_grad}(p_{v,e}). \tag{3}$$

This ensures that the forward pass uses the hard mask $\hat{m}_{v,e}$, while the backpropagation treats it as if it were the continuous probability $p_{v,e}$, allowing gradient-based optimization. The sparse matrix is:

$$\tilde{\boldsymbol{H}} = \boldsymbol{H} \odot \boldsymbol{M}, \tag{4}$$

where $\boldsymbol{M} \in \{0,1\}^{n \times m}$ is the sampled incidence-level mask, and $\odot$ indicates element-wise multiplication. To avoid notational conflict, in subsequent theoretical sections we define $\boldsymbol{P} \in [0,1]^{n \times m}$ as the matrix of marginal retention probabilities, and write: $\boldsymbol{\mathcal{M}} \sim q(\boldsymbol{\mathcal{M}} \mid \boldsymbol{P})$ to denote a probability distribution over binary incidence matrices, where $\boldsymbol{P}$ specifies the marginal inclusion probabilities such that $p_{v,e} = \mathbb{P}(\boldsymbol{\mathcal{M}}_{v,e} = 1)$. A realization of this random variable ($\boldsymbol{\mathcal{M}}$) is denoted by $\boldsymbol{M} \in \{0,1\}^{n \times m}$, used in the sparsified incidence matrix $\tilde{\boldsymbol{H}}$ (Equation 4).

**Discussion on differentiability.** Although Top-$k$ selection is discrete, the masking mechanism remains differentiable during training because the straight-through estimator (STE) replaces the nondifferentiable hard mask with a continuous surrogate in the backward pass. Specifically, the surrogate mask $m$ satisfies

$$m = \text{stop\_grad}(\hat{m}) + p - \text{stop\_grad}(p),$$

which ensures

$$\frac{\partial \mathcal{L}}{\partial s} = \frac{\partial \mathcal{L}}{\partial m} \cdot \frac{\partial m}{\partial p} \cdot \frac{\partial p}{\partial s}.$$

This guarantees that gradients propagate through the sparsifier into the HGNN, preventing the gradient-blocking effect typically associated with stochastic sampling-based sparsification. Hard Top-$k$ masks are used only during inference, while training uses the differentiable relaxation $m$.

**Variant of EHGNN-F conditioned on node features.** EHGNN-F is feature-agnostic, each node–hyperedge incidence $(v, e)$ has a free learnable parameter $s_{v,e}$, which do not depend on node features, rather they are optimized via task loss. In semi-supervised setting, this may cause only the incidences near labeled nodes to benefit from supervision. To address this issue, feature-conditioned EHGNN-F (**EHGNN-F (cond.)**) learns the scorer as a parameter of node features as follows:

$$s_{v,e} = \texttt{MLP}(\boldsymbol{X}_v || \hat{\boldsymbol{X}}_e), \tag{5}$$

where $\boldsymbol{X}_v$ is the feature vector of node $v$ and the aggregated embedding of hyperedge $e$ is $\hat{\boldsymbol{X}}_e = \frac{1}{|e|} \sum_{v \in e} \boldsymbol{X}_v$. The shared MLP parameters allow limited labeled nodes to shape sparsification decisions across the whole hypergraph, alleviating the lack of supervision issue.

To summarize, EHGNN-F promotes competitive selection among node–edge connections, which ensures that only the most informative relationships are retained under a global sparsity constraint. This selective pressure improves the model's ability to identify task-relevant structure while avoiding over-retention of redundant or noisy incidences.

## 4.2 COARSE-GRAINED EDGE-LEVEL MASKING (EHGNN-C)

Although Incidence-level masking allows fine-grained control, it may result in a large parameter size since the network keeps one parameter per incidence pair. Coarse-grained Edge-Level masking alleviates this by maintaining one parameter per edge. For instance, if the input hypergraph is $k$-uniform (every edge contains $k$ nodes), edge-level scoring reduces the model parameters by $k$.

The learnable edge score parameter $s$ is converted to soft probabilities via sigmoid function:

$$p_e = \sigma(s_e) / \sum_e \sigma(s_e) \tag{6}$$

We treat $\boldsymbol{p} = (p_e)$ as the marginal probabilities of a categorical distribution over all hyperedges and sample binary hyperedge masks from it:

$$\boldsymbol{m} \sim \text{Top-k } \texttt{Categorical}(\boldsymbol{p}), \quad \sum_e m_e = \kappa.$$

The binary mask $m_e \in \{0,1\}$ determines whether the hyperedge $e$ is retained or dropped. This allows efficient sparsification of hyperedges with a low-parameter overhead and is especially well suited for large-scale hypergraphs. The sparsified incidence matrix becomes:

$$\tilde{\boldsymbol{H}} = \boldsymbol{H} \odot \boldsymbol{1}_n \boldsymbol{m}^T, \tag{7}$$

where $\mathbf{1}_n \in \mathbb{R}^n$ is a vector of ones, $\boldsymbol{m}^T \in \mathbb{R}^{1 \times m}$ is the binary mask.

**Variant of EHGNN-C conditioned on node features.** EHGNN-C is feature-agnostic, each edge has a free learnable parameter $s_e$, which do not depend on its constituent node features. In semi-supervised setting, this may cause only the edges containing labeled nodes to benefit from supervision. To address this issue, feature-conditioned EHGNN-C (**EHGNN-C (cond.)**) learns the scorer as a parameter of node features using a permutation-invariant operator, such as mean pooling:

$$\hat{\boldsymbol{X}}_e = \frac{1}{|e|} \sum_{v \in e} \boldsymbol{X}_v, \quad s_e = \texttt{MLP}(\hat{\boldsymbol{X}}_e) \tag{8}$$

**Training and Inference.** The masked incidence $\tilde{\boldsymbol{H}}$ is passed into any HGNN architecture (e.g., HyperGCN, UniGNN, etc.) and trained end-to-end with a task loss $\mathcal{L}_{\text{task}}$. This yields a sparse hypergraph $\tilde{\boldsymbol{H}}$ adapted to the node classification task. During inference, instead of stochastic sampling, we use deterministic top-$k$ (or top-$\kappa$) selection from $\boldsymbol{p}$ to select incidences (or edges) of $\tilde{\boldsymbol{H}}$.

Unlike prior sparsification approaches for graphs or hypergraphs, EdgeMask-HGNN introduces a fully learnable, task-aware sparsification module at two granularities, operating end-to-end with downstream supervision. The fine-grained variant (EHGNN-F) allows localized selection of informative node–edge connections, while the coarse-grained variant (EHGNN-C) offers a more parameter-efficient alternative by scoring entire hyperedges via permutation-invariant pooling. Although masking strategies have been explored in the graph sparsification literature, such as $L_0$-based sparsifiers (Ye & Ji, 2021), EdgeMask-HGNN addresses the unique structure of hypergraphs, where sparsification potentially operates over exponentially many possible incidence patterns.

**Remark.** In our experiments, the MLP-based feature-conditioned scorer defined in 5 and 8 is prohibitively memory-intensive on large datasets as they introduce additional activation buffers due to pooling node features at the incidence or hyperedge level. To ensure scalability, we adopt a lightweight low-rank parameterization that preserves feature-conditioning without the associated memory overhead, we adopted a low-rank bilinear parameterization (EHGNN-F(cond,LR) and EHGNN-C(cond,LR)) discussed in detail in Appendix C.

## 5 THEORETICAL GUARANTEES

We provide a formal analysis of EHGNN-F by examining stability, and convergence behavior. The proofs are presented in the Appendix for brevity.

**Stability and Robustness.** We prove that the model output remains stable under stochastic masking. Let $p_{v,e} \in [0,1]$ represent the marginal probability of retaining the connection $(v,e)$. A binary mask $\boldsymbol{M} \in \{0,1\}^{n \times m}$ is sampled as a realization of a random variable $\boldsymbol{\mathcal{M}} \sim q(\boldsymbol{\mathcal{M}} \mid \boldsymbol{P})$. Each realization corresponds to a subhypergraph $\tilde{\boldsymbol{H}} = \boldsymbol{H} \odot \boldsymbol{M}$, and defines a distribution over the subhypergraphs.

**Theorem 5.1** (Perturbation Stability). *If the HGNN $f_\theta(\boldsymbol{X}, \cdot)$ is L-Lipschitz w.r.t. the Frobenius norm,*

$$\mathbb{E}\left[\left\|f_\theta(\boldsymbol{X}, \tilde{\boldsymbol{H}}) - f_\theta(\boldsymbol{X}, \mathbb{E}[\tilde{\boldsymbol{H}}])\right\|_F\right] \leq L \cdot \sqrt{\sum_{v,e} H_{v,e}^2 p_{v,e}(1 - p_{v,e})}. \tag{9}$$

Here the expectation $\mathbb{E}[\tilde{\boldsymbol{H}}]$ quantifies the average connectivity across sparsified subhypergraphs sampled from the model. This theorem suggests that, as the learned mask becomes more deterministic ($p_{v,e} \to 1$ or $0$), the model's output becomes more stable.

**Convergence of retention probabilities $p_{v,e}$.**

**Theorem 5.2** (Convergence of Fine-Grained Mask Parameters). *Let the mask be defined by probabilities $\boldsymbol{P} = \sigma(\boldsymbol{S}) \in [0,1]^{n \times m}$, where each entry $p_{v,e} = \sigma(s_{v,e})$ is a sigmoid-transformed logit.*

*If the gradient of the loss with respect to each $p_{v,e}$ maintains a fixed sign and the logits are updated via gradient descent, then for any small $\epsilon > 0$, the time to convergence (either $p_{v,e} < \epsilon$ or $p_{v,e} > 1 - \epsilon$) follows*

$$\tau \geq \mathcal{O}\left(1/\epsilon\right)$$

Table 1: Dataset statistics.

| | Cora | Citeseer | Pubmed | Cora-CA | DBLP-CA | 20News | Mushroom | NTU2012 | ModelNet40 | Yelp | House | Walmart | Actor | Twitch | Pokec | Trivago |
|---|---|---|---|---|---|---|---|---|---|---|---|---|---|---|---|---|
| $|V|$ | 2708 | 3312 | 19717 | 2708 | 41302 | 16242 | 8124 | 2012 | 12311 | 50758 | 1290 | 88860 | 16255 | 16812 | 14998 | 172738 |
| $|E|$ | 1579 | 1079 | 7963 | 1072 | 22363 | 100 | 298 | 2012 | 12311 | 679302 | 341 | 69906 | 10164 | 2627 | 2406 | 233202 |
| # feature | 1433 | 3703 | 500 | 1433 | 1425 | 100 | 22 | 100 | 100 | 1862 | 100 | 100 | 50 | 7 | 65 | 300 |
| # class | 7 | 6 | 3 | 7 | 6 | 4 | 2 | 67 | 40 | 9 | 2 | 11 | 3 | 2 | 2 | 160 |
| $t$ | 4786 | 3453 | 34629 | 4585 | 99561 | 65451 | 40620 | 10060 | 61555 | 4523594 | 11843 | 460630 | 53372 | 16356 | 5502 | 726861 |
| Avg edge size ($\bar{e}_{avg}$) | 3.03 | 3.20 | 4.35 | 4.28 | 4.45 | 654.51 | 136.31 | 5.00 | 5.00 | 6.66 | 34.73 | 6.59 | 5.25 | 6.23 | 2.29 | 3.12 |
| Density ($\bar{e}_{avg}/|V|$) | 0.00112 | 0.00097 | 0.00022 | 0.00158 | 0.00011 | 0.04030 | 0.01678 | 0.00249 | 0.00041 | 0.00013 | 0.02692 | 0.00007 | 0.00032 | 0.00037 | 0.00015 | 0.00002 |
| Edge homophily ($\mathcal{H}_e$) | 0.75 | 0.68 | 0.78 | 0.78 | 0.87 | 0.61 | 0.94 | 0.79 | 0.87 | 0.29 | 0.49 | 0.60 | 0.46 | 0.49 | 0.45 | 0.98 |
| Node homophily ($\mathcal{H}_n$) | 0.76 | 0.68 | 0.76 | 0.76 | 0.86 | 0.53 | 0.91 | 0.77 | 0.85 | 0.24 | 0.51 | 0.50 | 0.48 | 0.50 | 0.45 | 0.98 |

Table 2: Accuracy ($\pm$ std) across different datasets for each algorithm at sparsity = 50%. OOM=Out-of-Memory. Bold (underline) denotes the best (2nd best) result per dataset.

| Algorithms | 20news | ModelNet40 | Mushroom | NTU2012 | Actor | CiteSeer | Cora-CA | DBLP-CA |
|---|---|---|---|---|---|---|---|---|
| Full | 81.39 ± 0.02 | 94.89 ± 0.06 | 97.75 ± 0.31 | 87.67 ± 0.28 | 64.53 ± 0.03 | 68.19 ± 0.39 | 81.48 ± 0.36 | 90.77 ± 0.07 |
| Random | 68.32 ± 0.39 | 81.77 ± 0.66 | 94.77 ± 0.52 | 75.59 ± 1.52 | 65.39 ± 0.45 | 31.64 ± 1.17 | 55.95 ± 1.08 | 71.28 ± 0.44 |
| Degdist | 60.99 ± 0.36 | 68.37 ± 0.82 | 94.65 ± 0.46 | 64.77 ± 1.86 | 63.44 ± 0.09 | 26.84 ± 1.70 | 51.52 ± 1.35 | 64.89 ± 0.51 |
| Spectral | 72.82 ± 0.04 | 82.52 ± 0.08 | 97.75 ± 0.12 | 65.21 ± 0.58 | 63.89 ± 0.01 | 35.39 ± 0.66 | 62.81 ± 0.12 | 84.83 ± 0.03 |
| HSL | 81.85 ± **0.10** | 86.36 ± 1.61 | **99.79 ± 0.03** | 80.45 ± 1.84 | 63.01 ± 0.24 | 62.20 ± 0.24 | 75.82 ± 1.26 | **90.11 ± 0.55** |
| EHGNN-F | 77.99 ± 0.07 | 95.25 ± 0.06 | 96.39 ± 0.39 | **87.51 ± 0.22** | 77.00 ± 0.07 | 68.12 ± 0.52 | 79.53 ± 0.55 | 88.11 ± 0.07 |
| EHGNN-C | 78.01 ± 0.34 | 93.74 ± 0.10 | 94.14 ± 0.60 | 84.65 ± 0.51 | 77.20 ± 0.07 | 67.46 ± 0.58 | 80.50 ± 0.15 | 89.30 ± 0.07 |
| EHGNN-F (cond) | 78.67 ± 0.06 | 95.59 ± 0.12 | 95.07 ± 0.69 | 87.32 ± 0.43 | **78.73 ± 0.05** | 68.36 ± 0.43 | 76.07 ± 0.66 | 86.27 ± 0.08 |
| EHGNN-C (cond) | 74.63 ± 0.39 | 94.54 ± 0.15 | 97.52 ± 0.12 | 85.45 ± 0.29 | 77.57 ± 0.06 | 67.20 ± 0.57 | 75.51 ± 0.16 | 87.10 ± 0.14 |
| EHGNN-F (cond,LR) | 76.34 ± 0.16 | **95.98 ± 0.01** | 96.91 ± 0.11 | 86.28 ± 0.20 | 78.00 ± 0.06 | 67.56 ± 0.42 | 76.93 ± 0.26 | 87.48 ± 0.09 |
| EHGNN-C (cond,LR) | 76.83 ± 0.57 | 95.12 ± 0.09 | 96.73 ± 0.32 | 86.72 ± 0.26 | 77.57 ± 0.06 | 67.58 ± 0.64 | 79.32 ± 0.18 | 88.27 ± 0.13 |

| | Cora | House | Pokec | PubMed | Twitch | Walmart | Yelp | Trivago | Avg. Rank |
|---|---|---|---|---|---|---|---|---|---|
| Full | 78.23 ± 0.31 | 73.75 ± 0.14 | 58.37 ± 0.04 | 85.60 ± 0.09 | 51.23 ± 0.05 | 94.78 ± 0.02 | 29.75 ± 1.11 | 61.68 ± 0.28 | |
| Random | 48.15 ± 1.25 | 60.99 ± 1.53 | 54.10 ± 0.27 | 47.83 ± 0.14 | 50.64 ± 0.55 | 64.70 ± 0.12 | 30.82 ± 0.43 | 26.78 ± 0.05 | 7.63 |
| Degdist | 39.38 ± 0.93 | 56.53 ± 2.42 | 53.43 ± 0.17 | 46.09 ± 0.27 | 50.55 ± 0.69 | 46.38 ± 0.12 | 28.41 ± 0.40 | 26.78 ± 0.05 | 9.38 |
| Spectral | 56.45 ± 0.07 | 57.52 ± 0.14 | 52.85 ± 0.04 | 49.07 ± 0.00 | 50.84 ± 0.11 | 70.54 ± 0.16 | OOM/28.67±1.03 | OOM/OOM | 7.75 |
| HSL | 74.10 ± 1.62 | 75.85 ± 0.31 | 58.03 ± 0.67 | 78.88 ± 1.16 | 51.15 ± 0.41 | 63.59 ± 0.21 | 24.89 ± 0.03 | **68.69 ± 1.06** | 5.75 |
| EHGNN-F | 74.33 ± 0.41 | 87.31 ± 1.58 | 58.78 ± 0.07 | **85.56 ± 0.04** | 50.74 ± 0.05 | 95.17 ± 0.02 | 32.31 ± 0.15 | 50.38 ± 1.40 | **3.38** |
| EHGNN-C | 73.18 ± 0.36 | 84.09 ± 1.17 | 58.79 ± 0.04 | 85.42 ± 0.06 | 50.57 ± 0.08 | 95.68 ± 0.01 | 31.75 ± 0.29 | 51.41 ± 0.72 | 4.75 |
| EHGNN-F (cond) | 73.94 ± 0.39 | **100.00 ± 0.00** | **59.17 ± 0.09** | 85.53 ± 0.06 | 50.84 ± 0.06 | **98.37 ± 0.01** | OOM | 46.22 ± 1.21 | 4.13 |
| EHGNN-C (cond) | **76.22 ± 0.35** | 72.14 ± 0.73 | 59.02 ± 0.03 | 85.52 ± 0.05 | 50.67 ± 0.03 | 94.72 ± 0.04 | 29.21 ± 0.36 | 50.14 ± 0.72 | 4.63 |
| EHGNN-F (cond,LR) | 74.77 ± 0.19 | **100.00 ± 0.00** | 58.67 ± 0.05 | **85.56 ± 0.06** | 50.47 ± 0.04 | 97.10 ± 0.01 | **32.48 ± 0.19** | 49.87 ± 0.95 | 3.75 |
| EHGNN-C (cond,LR) | 75.27 ± 0.39 | 85.14 ± 0.22 | 58.67 ± 0.03 | 85.53 ± 0.06 | **51.48 ± 0.09** | 95.71 ± 0.01 | 30.36 ± 0.66 | 48.18 ± 0.49 | 3.87 |

This theorem suggests that one can ensure $p_{v,e}(\tau)$ reaches within $\epsilon$ of its limiting value or at least $(1 - \epsilon)$; it suffices to train for #epochs proportional to $1/\epsilon$.

# 6 EXPERIMENTS

We focus on semi-supervised node classification task in the transductive setting. Note that, we have also evaluated EdgeMask-HGNN on unsupervised node clustering task (see Appendix H). We randomly split the nodes into training/validation/test samples using 50%/25%/25% splitting percentages Chien et al. (2021). Unless stated otherwise, we have used HGNN (Feng et al., 2019) as a backbone in our implementation. We follow Chien et al. (2021) to set the hyperparameters of various base HGNN models. We average the results of 10 experiments using multiple random splits and initializations. All experiments are run on a system with 512 GB RAM and a NVIDIA Tesla V100 GPU with 32 GB memory. All algorithms run for 500 epochs except Trivago (2000 epochs) with a retention rate of $k = 50\%$ ($\kappa = 50\%$).

**Datasets.** The hypergraph datasets and their statistics are presented in Table 1. It includes 3 recently proposed heterophilic benchmarks (Actor, Twitch, Pokec) from Li et al. (2025). The rest of the datasets are well-known in the hypergraph learning literature, originating from works such as Yadati et al. (2019); Chien et al. (2021) where they are discussed in detail.

**Baselines.** The baseline algorithms include i) *Full*: the HGNN model is trained on the entire hypergraph, ii) *Degdist*: top-$k$ nodes are sampled from the distribution Top-$k$ Categorical($\boldsymbol{d}$) with $d_v$ being the normalized degree of node $v$. All node, hyperedge connections involving the sampled nodes are kept in the sparsified hypergraph, iii) *Random*: drops a node-hyperedge connection $(u, e)$ if an i.i.d uniform random variable $r \sim \mathcal{U}[0, 1]$ satisfies $r < k/|E|$, and finally, iv) *Spectral*: We sample top-$\kappa$ hyperedges based on their effective resistance. We approximated effective resistance $R(e) \approx \sum_{v \in e} \left(L^\dagger\right)_{vv}$ for efficiency purposes. Here $L^\dagger$ indicates the Moore–Penrose pseudoinverse of the Hypergraph Laplacian (Zhou et al., 2006), and (v) Hypergraph Structure Learning, *HSL* (Cai et al., 2022). The model parameters, hyperparameters of the algorithms, ablation studies, convergence of retention probabilities, node-clustering experiments, and other supplementary experiments can be found in the Appendix. Our source codes: https://github.com/toggled/ehgnn.

## 6.1 EXPERIMENTAL EVALUATION

Table 3: Mean peak GPU memory usage (GB) during training (% change is w.r.t Full). (**Bold**=best)

| Algorithms | 20news | ModelNet40 | Mushroom | NTU2012 | Actor | CiteSeer | Cora-CA | DBLP-CA |
|---|---|---|---|---|---|---|---|---|
| Full | 0.43 | 0.41 | 0.25 | 0.08 | 0.31 | 0.14 | 0.09 | 1.07 |
| Random | **0.22 (49% ↓)** | **0.22 (45% ↓)** | **0.13 (49% ↓)** | **0.05 (38% ↓)** | **0.21 (32% ↓)** | 0.16 (14% ↑) | **0.08 (11% ↓)** | 0.86 (19% ↓) |
| Degdist | **0.22 (49% ↓)** | **0.22 (45% ↓)** | **0.13 (49% ↓)** | **0.05 (38% ↓)** | **0.21 (32% ↓)** | 0.16 (14% ↑) | **0.08 (11% ↓)** | 0.86 (19% ↓) |
| Spectral | 0.29 (33% ↓) | **0.22 (45% ↓)** | 0.19 (24% ↓) | **0.05 (38% ↓)** | 0.23 (26% ↓) | 0.16 (16% ↑) | 0.08 (8% ↓) | 0.96 (10% ↓) |
| HSL | 0.41 (4% ↓) | 0.40 (1% ↓) | 0.13 (47% ↓) | 0.08 (1% ↓) | 0.33 (6% ↑) | 0.17 (21% ↑) | 0.09 (1% ↑) | 1.18 (10% ↑) |
| EHGNN-F | 0.31 (28% ↓) | 0.29 (29% ↓) | 0.18 (30% ↓) | 0.06 (23% ↓) | 0.30 (2% ↓) | 0.13 (5% ↓) | 0.08 (10% ↓) | 0.89 (17% ↓) |
| EHGNN-C | 0.34 (22% ↓) | 0.29 (29% ↓) | 0.20 (21% ↓) | 0.06 (24% ↓) | 0.30 (2% ↓) | 0.13 (5% ↓) | 0.08 (10% ↓) | 0.92 (14% ↓) |
| EHGNN-F(cond) | 0.31 (28% ↓) | 0.29 (30% ↓) | 0.18 (30% ↓) | 0.06 (24% ↓) | 0.30 (4% ↓) | 0.30 (116% ↑) | 0.15 (69% ↑) | 2.51 (134% ↑) |
| EHGNN-C(cond) | 0.34 (22% ↓) | 0.29 (28% ↓) | 0.20 (22% ↓) | 0.06 (23% ↓) | 0.30 (2% ↓) | 0.15 (11% ↑) | **0.08 (11% ↓)** | 0.90 (16% ↓) |
| EHGNN-F (cond,LR) | 0.31 (28% ↓) | 0.28 (30% ↓) | 0.18 (30% ↓) | 0.06 (23% ↓) | 0.29 (5% ↓) | **0.13 (6% ↓)** | 0.08 (12% ↓) | **0.85 (21% ↓)** |
| EHGNN-C (cond,LR) | 0.35 (19% ↓) | 0.30 (28% ↓) | 0.20 (21% ↓) | 0.06 (20% ↓) | 0.31 (1% ↓) | 0.18 (26% ↑) | 0.09 (5% ↑) | 1.11 (4% ↑) |

| | Cora | House | Pokec | PubMed | Twitch | Walmart | Yelp | Trivago |
|---|---|---|---|---|---|---|---|---|
| Full | 0.09 | 0.08 | 0.19 | 0.40 | 0.25 | 2.79 | 19.60 | 6.20 |
| Random | **0.08 (12% ↓)** | **0.05 (40% ↓)** | **0.12 (38% ↓)** | 0.25 (36% ↓) | **0.12 (50% ↓)** | **1.40 (50% ↓)** | 11.01 (44% ↓) | **4.44 (28% ↓)** |
| Degdist | **0.08 (11% ↓)** | **0.05 (40% ↓)** | **0.12 (38% ↓)** | 0.25 (36% ↓) | **0.12 (50% ↓)** | **1.40 (50% ↓)** | 11.01 (44% ↓) | **4.44 (28% ↓)** |
| Spectral | **0.08 (12% ↓)** | 0.08 (4% ↓) | **0.12 (38% ↓)** | 0.28 (29% ↓) | 0.13 (47% ↓) | 1.71 (39% ↓) | OOM/16.68 (15%↓) | OOM/OOM |
| HSL | 0.09 (2% ↑) | 0.08 (4% ↓) | 0.17 (14% ↓) | **0.24 (39% ↓)** | 0.21 (16% ↓) | 10.70 (283% ↓) | 11.89 (39% ↓) | 5.53 (11% ↓) |
| EHGNN-F | 0.08 (10% ↓) | 0.05 (30% ↓) | 0.18 (5% ↓) | 0.33 (16% ↓) | 0.22 (12% ↓) | 1.93 (31% ↓) | 11.51 (41% ↓) | 4.55 (27% ↓) |
| EHGNN-C | 0.08 (11% ↓) | 0.06 (25% ↓) | 0.18 (5% ↓) | 0.33 (16% ↓) | 0.22 (11% ↓) | 1.94 (31% ↓) | 15.45 (21% ↓) | 4.84 (22% ↓) |
| EHGNN-F(cond) | 0.16 (72% ↑) | 0.05 (30% ↓) | 0.18 (6% ↓) | 0.34 (14% ↓) | 0.21 (13% ↓) | 1.90 (32% ↓) | OOM | 4.52 (27% ↓) |
| EHGNN-C(cond) | 0.08 (9% ↓) | 0.06 (25% ↓) | 0.18 (5% ↓) | 0.33 (15% ↓) | 0.22 (11% ↓) | 1.94 (31% ↓) | **10.09 (49% ↓)** | 4.84 (22% ↓) |
| EHGNN-F (cond,LR) | **0.08 (12% ↓)** | 0.05 (30% ↓) | 0.18 (8% ↓) | 0.32 (18% ↓) | 0.22 (13% ↓) | 1.89 (32% ↓) | 11.49 (41% ↓) | 4.53 (27% ↓) |
| EHGNN-C (cond,LR) | 0.10 (6% ↑) | 0.06 (29% ↓) | 0.19 (2% ↓) | 0.37 (7% ↓) | 0.22 (12% ↓) | 1.95 (30% ↓) | 11.56 (41% ↓) | 4.44 (22% ↓) |

**I. Effectiveness and Scalability.** Table 2 highlights the effectiveness of EdgeMask-HGNN over the baselines. We observe several things:

*Supervised vs Unsupervised sparsifiers:* On most datasets, EHGNN variants consistently outperform Random and degree-based sparsification. This highlights the importance of task-aware learning of sparsification masks, which can retain task-relevant incidences or hyperedges while discarding noisy ones. On large-scale datasets, spectral methods face memory issues due to the computation of the large matrix pseudoinverse. We also used an approximate method using a Hutchinson-type random projection estimator for $diag(L^\dagger)$, which enabled computation on Yelp, yet goes OOM on Trivago.

*Full training vs Sparsification:* Although it is not expected that a 50% sparsification to yield better accuracy than full training, EdgeMask-HGNNs outperform full training on several benchmarks, such as ModelNet40, Actor, Citeseer, House, Pokec, Walmart, and Yelp. This demonstrates that pruning of irrelevant pairs can enhance generalization by improving the signal-to-noise ratio during message passing. Deeper analysis (Figure 7, Appendix) reveals that EdgeMask-HGNN variants offer more accuracy gain (over Full) near low homophily/high heterophily regimes (roughly $\leq 0.6$).

*Fine-grained vs Coarse-grained EdgeMasking and variants:* EHGNN-C is slightly better than EHGNN-F, often outperforming on datasets like 20news, Cora, Cora-CA, DBLP-CA, and Walmart. To understand when coarse-grained versus fine-grained sparsification is preferable, we analyzed the accuracy gap (EHGNN-C - EHGNN-F) across all datasets and correlated it with hypergraph statistics, in particular the number of nodes and density (Table 1). The number of nodes shows a moderately strong positive rank correlation with accuracy gap (Spearman $\rho_n = 0.67$), while incidence density shows moderately strong negative correlation ($\rho_d = -0.61$). This indicates that EHGNN-C tends to be more effective on moderately sparse hypergraphs with large #nodes. On contrary, EHGNN-F performs better on denser hypergraphs, where fine-grained incidence-level masking can better suppress noisy connections inside large hyperedges.

**II. Memory efficiency.** Table 3 reports peak GPU memory across methods at the same sparsity budget. Note that, peak GPU usage during HGNN training is dominated by activations (messages, pooled edge features, and autograd buffers) that scale with # incidences after sparsification $k = \sum_{e \in \tilde{\mathcal{H}}} |e|$, and feature dimension $d$, not by the parameter overhead (see Table 6, Appendix B).

*On small-scale datasets:* On small datasets, all methods exhibit similar peak memory because the entire incidence structure fits comfortably on a GPU. The incidence and hyperedge counts are not large enough to pose scalability challenges, thus, sparsification strategies bring little difference in this regime. We also find that Spectral methods require noticeably large memory in this regime.

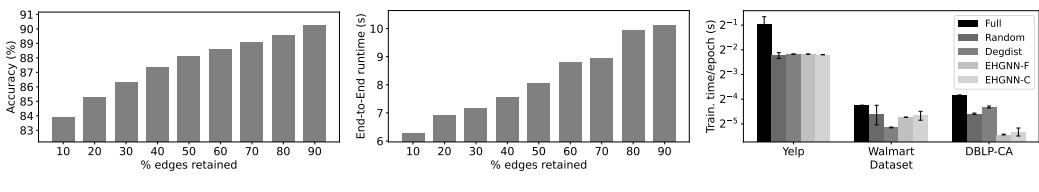

(a) Accuracy vs. Sparsity  (b) End-to-end runtime vs. Sparsity  (c) Training time

Figure 3: (a-b) Impact of sparsity on EHGNN-F's performance (DBLP-CA dataset) (c) Training time efficiency of the methods.

*On moderately large hypergraphs with medium incidence count (DBLP-CA, Walmart):* The differences emerge clearly on DBLP-CA ($t = 99{,}561$) and Walmart ($t = 460{,}630$), where activations begin to dominate memory usage but are not overwhelmingly large. In this region, Random and degree-based pruning yield good memory savings but at a significant accuracy cost. EHGNN-F and EHGNN-C are better alternatives, since with a memory footprint similar to Random and degree-based pruning, they yield better accuracy. Our correlation analysis supports this trend: the memory delta (vs. Full) for both EHGNN-F and EHGNN-C is strongly negatively correlated with the number of hyperedges and incidences (Pearson $\rho_m = -0.99$, $\rho_t = -0.98$ for EHGNN-C; $\rho_m = -0.99$, $\rho_t = -0.999$ for EHGNN-F), indicating that both sparsifiers become increasingly memory-efficient as the hypergraph grows.

*On extremely large hypergraphs with very high incidence count (Yelp, Trivago):* Yelp ($t = 4{,}523{,}594$) is the largest and most computationally demanding hypergraph in our benchmark. At this scale, even after sparsification, the retained incidence structure is so large that all learnable sparsifiers exhibit substantially higher peak activation memory on Yelp than on medium-scale datasets (DBLP-CA and Walmart). While Full HGNN training already incurs very high peak memory (19.60 GB), the learnable sparsifiers show mixed behavior: the feature-conditioned variant EHGNN-F(cond) runs out of memory, but the other EHGNN variants actually reduce peak memory by 21–49%. Trivago ($t = 726{,}861$) is also large but less extreme; all EHGNN variants reduce memory compared to Full, showing that learnable incidence pruning remains effective at this scale.

*Fine- versus coarse-grained masking:* Among our methods, EHGNN-F achieves the largest and most consistent memory reductions, since incidence-level pruning directly lowers the number of active node–edge pairs $k$, the dominant contributor to activation memory. We also note that feature-conditioned variants (EHGNN-F(cond) and EHGNN-C(cond)) introduce additional activation buffers due to pooling node features at the incidence or hyperedge level and evaluating scorer MLPs. These per-incidence operations scale with the retained incidence count $k$ and often lead to higher peak memory than their non-conditioned counterparts, explaining the OOM observed on Yelp and the larger memory footprint on several medium-scale datasets.

In summary, peak GPU memory is governed primarily by hypergraph scale, especially the total incidence count $t$. EHGNN-F provides the strongest memory reductions by directly minimizing the number of active incidences. EHGNN-C offers similar benefits but is more sensitive to edge cardinality. Feature-conditioned variants introduce additional activation overhead, which becomes increasingly pronounced at large $t$. Across datasets, both fine-grained and coarse-grained masking reliably reduce memory, with the greatest gains on large but manageable incidence regimes and with overhead appearing only when $t$ becomes extremely large or when feature conditioning is used.

**III. Accuracy/Runtime vs sparsity trade-off.** We analyze the trade-off between accuracy and sparsity due to edge pruning by our proposed EdgeMask-HGNN in Figure 3a, while the impact of sparsity on the end-to-end runtime (Training + Evaluation) is presented in Figure 3b. We find that as more edges are retained in the sparsification, the accuracy increases at the cost of a higher runtime.

**IV. Runtime efficiency.** Figure 3c compares the average training time per epoch across several large-scale datasets. On Yelp, Walmart, and DBLP-CA, the Full HGNN consistently incurs the highest cost, while both EHGNN-F and EHGNN-C achieve substantially lower runtimes. This aligns with the fact that incidence-level sparsification reduces the message-passing workload, which dominates the cost on large hypergraphs. Table 4 further shows that the runtime impact of sparsification closely follows hypergraph scale. On small datasets (e.g., Cora, Cora-CA, Citeseer), message pass-

Table 4: End-to-end execution time (in seconds) as mean ± std across datasets for each algorithm at sparsity = 50%.

| Algorithms | 20news | ModelNet40 | Mushroom | NTU2012 | Actor | Citeseer | Cora-CA | DBLP-CA | |
|---|---|---|---|---|---|---|---|---|---|
| Full | 19.52 ± 4.14 | 18.97 ± 2.22 | 2.94 ± 0.22 | 15.85 ± 1.08 | 19.81 ± 3.74 | 7.26 ± 1.35 | 10.56 ± 0.42 | 24.94 ± 0.52 | |
| Random | 3.05 ± 0.05 | 11.65 ± 0.12 | 10.44 ± 0.09 | 3.04 ± 0.20 | **3.05 ± 0.06** | **1.34 ± 0.06** | 1.64 ± 0.18 | 12.85 ± 0.22 | |
| Degdist | **2.99 ± 0.12** | **3.48 ± 0.04** | **2.38 ± 0.15** | **2.90 ± 0.09** | 3.20 ± 0.07 | 1.36 ± 0.07 | **1.06 ± 0.33** | 15.70 ± 0.29 | |
| Spectral | 337.16 ± 0.16 | 163.13 ± 0.20 | 38.41 ± 0.23 | 3.63 ± 0.20 | 419.58 ± 0.20 | 2.93 ± 0.13 | 3.04 ± 0.14 | 19.66 ± 0.30 | |
| HSL | 114.28 ± 5.97 | 93.89 ± 5.09 | 62.41 ± 1.36 | 23.79 ± 1.26 | 7.75 ± 0.14 | 8.84 ± 0.87 | 8.65 ± 0.40 | 109.77 ± 9.09 | |
| EHGNN-F | 4.36 ± 0.08 | 4.47 ± 0.09 | 3.44 ± 0.26 | 3.06 ± 0.08 | 4.64 ± 0.05 | 1.85 ± 0.08 | 2.05 ± 0.09 | 7.61 ± 0.24 | |
| EHGNN-C | 4.14 ± 0.37 | 4.29 ± 0.16 | 3.28 ± 0.07 | 3.14 ± 0.11 | 4.67 ± 0.09 | 1.62 ± 0.09 | 2.00 ± 0.10 | 9.00 ± 1.64 | |
| EHGNN-F (cond) | 4.80 ± 0.11 | 5.12 ± 0.13 | 3.60 ± 0.10 | 3.45 ± 0.17 | 5.02 ± 0.11 | 2.06 ± 0.09 | 2.22 ± 0.13 | 15.61 ± 0.76 | |
| EHGNN-C (cond) | 4.47 ± 0.39 | 4.60 ± 0.10 | 3.59 ± 0.10 | 3.13 ± 0.20 | 4.93 ± 0.20 | 1.78 ± 0.09 | 2.04 ± 0.12 | 10.68 ± 1.22 | |
| EHGNN-F (cond,LR) | 4.48 ± 0.27 | 4.65 ± 0.11 | 3.58 ± 0.08 | 3.29 ± 0.23 | 4.96 ± 0.11 | 1.88 ± 0.06 | 1.91 ± 0.12 | **7.53 ± 0.18** | |
| EHGNN-C (cond,LR) | 4.53 ± 0.22 | 4.70 ± 0.09 | 3.61 ± 0.14 | 3.32 ± 0.19 | 5.01 ± 0.09 | 1.90 ± 0.07 | 1.96 ± 0.09 | 8.01 ± 0.27 | |
| | Cora | House | Pokec | PubMed | Twitch | Walmart | Yelp | Trivago | Avg. Rank |
| Full | 1.64 ± 0.04 | 14.78 ± 2.42 | 17.20 ± 2.52 | 5.00 ± 0.12 | 13.56 ± 4.23 | 26.36 ± 0.09 | 260.99 ± 56.35 | 355.77 ± 37.70 | |
| Random | 1.57 ± 0.03 | 2.96 ± 0.05 | **1.38 ± 0.09** | **3.21 ± 0.08** | **0.93 ± 0.25** | 20.87 ± 5.58 | **106.79 ± 8.84** | **55.67 ± 6.56** | 2.94 |
| Degdist | **1.56 ± 0.10** | **2.53 ± 0.38** | 1.50 ± 0.11 | 3.33 ± 0.14 | 1.59 ± 0.60 | **14.18 ± 0.19** | 111.09 ± 0.32 | 57.34 ± 0.51 | 1.88 |
| Spectral | 2.59 ± 0.13 | 3.05 ± 0.12 | 8.74 ± 0.36 | 500.81 ± 0.23 | 4.48 ± 0.15 | 243.25 ± 0.85 | OOM | OOM | 9.09 |
| HSL | 7.27 ± 0.10 | 10.07 ± 2.99 | 52.28 ± 4.35 | 105.95 ± 8.33 | 76.10 ± 4.07 | 813.21 ± 30.74 | 1141.31 ± 55.37 | 2350.25 ± 184.43 | 9.50 |
| EHGNN-F | 1.95 ± 0.19 | 3.04 ± 0.06 | 3.46 ± 0.30 | 5.19 ± 0.08 | 3.03 ± 0.63 | 18.86 ± 0.07 | 221.55 ± 0.38 | 224.09 ± 45.80 | 4.25 |
| EHGNN-C | 1.83 ± 0.10 | 3.06 ± 0.07 | 3.21 ± 0.22 | 4.85 ± 0.09 | 2.61 ± 0.70 | 19.81 ± 2.47 | 218.62 ± 1.24 | 277.14 ± 33.84 | 3.94 |
| EHGNN-F (cond) | 2.04 ± 0.20 | 3.34 ± 0.21 | 3.69 ± 0.34 | 6.07 ± 0.14 | 2.87 ± 0.83 | 21.92 ± 0.57 | OOM | 312.40 ± 52.99 | 7.78 |
| EHGNN-C (cond) | 1.80 ± 0.19 | 3.22 ± 0.12 | 3.59 ± 0.20 | 5.34 ± 0.11 | 2.60 ± 0.57 | 20.41 ± 0.20 | 2230.27 ± 0.26 | 281.25 ± 50.43 | 5.44 |
| EHGNN-F (cond,LR) | 2.03 ± 0.14 | 3.16 ± 0.07 | 3.36 ± 0.38 | 5.52 ± 0.10 | 3.17 ± 0.71 | 18.92 ± 0.19 | 281.93 ± 125.73 | 262.60 ± 23.72 | 5.38 |
| EHGNN-C (cond,LR) | 1.88 ± 0.12 | 2.94 ± 0.11 | 3.30 ± 0.09 | 5.08 ± 0.08 | 2.56 ± 0.62 | 18.11 ± 0.15 | 276.91 ± 0.48 | 251.44 ± 21.33 | 4.81 |

Table 5: Comparing various state-of-the-art HGNNs and their EHGNN-F enhanced counterparts (sparsity = 50%).

| Models | Actor | Cora | Cora-CA | House | ModelNet40 | NTU | PubMed |
|---|---|---|---|---|---|---|---|
| AllSetTrans. | 68.77 ± 0.60 | **76.93 ± 0.41** | **83.34 ± 0.77** | **100.00 ± 0.00** | **97.80 ± 0.07** | **90.30 ± 0.71** | **88.58 ± 0.12** |
| AllSetTrans.+EHGNN-F | **85.73 ± 0.36** | 76.54 ± 0.75 | 79.41 ± 0.44 | 99.94 ± 0.14 | 97.43 ± 0.11 | 88.91 ± 0.43 | 88.49 ± 0.16 |
| CE-GAT | 70.84 ± 4.40 | **74.39 ± 0.67** | 73.26 ± 0.54 | 96.78 ± 2.69 | 90.65 ± 0.15 | **78.33 ± 1.13** | **84.76 ± 0.16** |
| CE-GAT+EHGNN-F | **74.67 ± 1.76** | 73.65 ± 0.70 | **73.83 ± 1.15** | **99.57 ± 0.28** | **92.27 ± 0.29** | 77.46 ± 1.06 | 84.45 ± 0.69 |
| CE-GCN | 57.27 ± 0.37 | 52.51 ± 0.19 | 50.66 ± 0.40 | 51.98 ± 0.39 | 43.21 ± 0.37 | 35.20 ± 0.43 | 61.01 ± 0.06 |
| CE-GCN+EHGNN-F | **62.76 ± 0.03** | **53.29 ± 0.12** | **52.44 ± 0.49** | **52.63 ± 0.31** | **44.58 ± 0.62** | **35.55 ± 0.45** | **61.08 ± 0.05** |
| ED-HNN | 67.55 ± 0.10 | **78.05 ± 0.51** | **82.39 ± 0.62** | 96.84 ± 2.4 | **97.42 ± 0.06** | **89.54 ± 0.61** | **88.04 ± 0.24** |
| ED-HNN+EHGNN-F | **82.26 ± 0.16** | 76.01 ± 0.76 | 80.56 ± 0.6 | **99.50 ± 0.28** | 97.38 ± 0.07 | 89.46 ± 0.37 | 87.87 ± 0.26 |

ing is inexpensive, so the additional scoring and masking operations make EHGNN-F and EHGNN-C comparable to or slightly slower than Full. As incidence counts grow (House, PubMed, Pokec, Twitch), message passing dominates the cost and both sparsifiers consistently outperform Full, often reducing runtime by 3–5×. On Walmart, Yelp, and Trivago, EHGNN-F/EHGNN-C achieves 1.2-1.6× faster training.

**V. Adaptability to existing HGNNs.** EdgeMask-HGNN is model-agnostic and easily adaptable to different hypergraph architectures. To demonstrate this, we have employed EHGNN-F into ED-HNN (Wang et al., 2023), AllSetTransformer, CE-GCN, and CE-GAT architectures Chien et al. (2021) and present the results in Table 5. We observe that EHGNN-F achieves comparable and sometimes better performance across these three HGNN backbones. This indicates that the sparsification mechanism is flexible and can act as a plug-in module without significantly degrading the existing HGNNs' representational power.

## 7 Conclusion and Future works

We introduced EdgeMask-HGNN, a novel task-aware sparsification framework that effectively reduces memory overhead without sacrificing predictive performance of HGNNs. We proposed two learnable masking strategies: fine-grained masking and coarse-grained masking– both trained end-to-end using feedback from downstream tasks. Furthermore, EdgeMask-HGNN is theoretically grounded in terms of stability and convergence of the learned sparsifiers.

Extensive experiments across diverse and challenging benchmarks and multiple HGNN backbones underscore the adaptability and effectiveness of EdgeMask-HGNN. In particular, the fine-grained variant not only improves accuracy over full hypergraph training in many cases, but also achieves slightly smaller execution time on large hypergraphs with a comparable memory footprint. Finally, our approach consistently outperforms unsupervised and spectral baselines in accuracy, with a comparable or better memory footprint.

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

APPENDIX

# A  THEORETICAL ANALYSIS

We first prove the following lemma that will be used later in Stability proof.

**Lemma A.1** (Expectation of Masked Structure). *Let $\boldsymbol{P} \in [0,1]^{n \times m}$ be the matrix of marginal inclusion probabilities, i.e., $p_{v,e} = \mathbb{P}(\boldsymbol{\mathcal{M}}_{v,e} = 1)$. Then,*

$$\mathbb{E}[\tilde{\boldsymbol{H}}] = \boldsymbol{H} \odot \boldsymbol{P}.$$

*Proof.*

$$\begin{aligned} \mathbb{E}[\tilde{H}_{v,e}] &= \mathbb{E}[H_{v,e} \cdot \mathcal{M}_{v,e}] \\ &= H_{v,e} \cdot \mathbb{E}[\mathcal{M}_{v,e}] \\ &= H_{v,e} \cdot p_{v,e} \\ \Rightarrow \quad \mathbb{E}[\tilde{\boldsymbol{H}}] &= \boldsymbol{H} \odot \boldsymbol{P} \end{aligned}$$

$\square$

**Theorem 5.1** (Perturbation Stability). *If $f_\theta(\boldsymbol{X}, \cdot)$ is L-Lipschitz w.r.t. the Frobenius norm,*

$$\mathbb{E}\left[\left\| f_\theta(\boldsymbol{X}, \tilde{\boldsymbol{H}}) - f_\theta(\boldsymbol{X}, \mathbb{E}[\tilde{\boldsymbol{H}}]) \right\|_F\right] \leq$$

$$L \cdot \sqrt{\sum_{v,e} H_{v,e}^2 p_{v,e}(1 - p_{v,e})}.$$

*Proof.* Since we assumed the HGNN $f_\theta$ to be $L$-Lipschitz,

$$\|f_\theta(\boldsymbol{X}, \tilde{\boldsymbol{H}}_1) - f_\theta(\boldsymbol{X}, \tilde{\boldsymbol{H}}_2)\|_F \leq L \cdot \|\tilde{\boldsymbol{H}}_1 - \tilde{\boldsymbol{H}}_2\|_F$$

for all incidence matrices $\tilde{\boldsymbol{H}}_1, \tilde{\boldsymbol{H}}_2$.

Let $\bar{\boldsymbol{H}} = \mathbb{E}[\tilde{\boldsymbol{H}}]$. As per Lemma A.1, $E[\tilde{\boldsymbol{H}}] = \boldsymbol{H} \odot \boldsymbol{P}$. Thus

$$\|f_\theta(\boldsymbol{X}, \tilde{\boldsymbol{H}}) - f_\theta(\boldsymbol{X}, \bar{\boldsymbol{H}})\|_F \leq L \cdot \|\tilde{\boldsymbol{H}} - \bar{\boldsymbol{H}}\|_F$$

Taking expectation

$$\mathbb{E}_{\tilde{\boldsymbol{H}}}\left[\|f_\theta(\boldsymbol{X}, \tilde{\boldsymbol{H}}) - f_\theta(\boldsymbol{X}, \bar{\boldsymbol{H}})\|_F\right] \leq L \cdot \mathbb{E}_{\tilde{\boldsymbol{H}}}\left[\|\tilde{\boldsymbol{H}} - \bar{\boldsymbol{H}}\|_F\right]$$

By Jensen's inequality, for any matrix $\boldsymbol{A}$

$$\mathbb{E}[\|\boldsymbol{A}\|] \leq \sqrt{\mathbb{E}[\|\boldsymbol{A}\|^2]}$$

$$\Rightarrow \mathbb{E}[\|\tilde{\boldsymbol{H}} - \bar{\boldsymbol{H}}\|_F] \leq \sqrt{\mathbb{E}[\|\tilde{\boldsymbol{H}} - \bar{\boldsymbol{H}}\|_F^2]}$$

Since the entries $\tilde{H}_{v,e}$ are independent:

$$\mathbb{E}[(\tilde{H}_{v,e} - \bar{H}_{v,e})^2] = \text{Var}[\tilde{H}_{v,e}] = H_{v,e}^2 \cdot p_{v,e}(1 - p_{v,e})$$

It follows that,

$$\mathbb{E}\left[\|\tilde{\boldsymbol{H}} - \bar{\boldsymbol{H}}\|_F^2\right] = \sum_{v,e} \text{Var}[\tilde{H}_{v,e}] = \sum_{v,e} H_{v,e}^2 p_{v,e}(1 - p_{v,e})$$

Thus,

$$\mathbb{E}_{\tilde{\boldsymbol{H}}}\left[\|f_\theta(\boldsymbol{X}, \tilde{\boldsymbol{H}}) - f_\theta(\boldsymbol{X}, \mathbb{E}[\tilde{\boldsymbol{H}}])\|_F\right]$$

$$\leq L \cdot \sqrt{\sum_{v,e} H_{v,e}^2 \cdot p_{v,e}(1 - p_{v,e})}$$

$\square$

**Theorem 5.2** (Convergence of Fine-Grained Mask Parameters). *Let the mask be defined by probabilities $\boldsymbol{P} = \sigma(\boldsymbol{S}) \in [0,1]^{n \times m}$, where each entry $p_{v,e} = \sigma(s_{v,e})$ is a sigmoid-transformed logit.*

*If the gradient of the loss with respect to each $p_{v,e}$ maintains a fixed sign and the logits are updated via gradient descent, then for any small $\epsilon > 0$, the time to convergence (either $p_{v,e} < \epsilon$ or $p_{v,e} > 1 - \epsilon$) follows*

$$\tau \geq \mathcal{O}\left(\frac{1}{\epsilon}\right)$$

*Proof.* Each probability $p_{v,e} \in (0,1)$ is parameterized as a sigmoid function:

$$p_{v,e} = \sigma(s_{v,e}) = \frac{1}{1 + e^{-s_{v,e}}}$$

For simplicity, we ignore the normalisation of the scores. Its derivative with respect to the logit is:

$$\frac{dp_{v,e}}{ds_{v,e}} = p_{v,e}(1 - p_{v,e})$$

This gradient is positive and bounded by $\frac{1}{4}$, ensuring smooth and monotonic updates.

Using chain rule, the gradient descent update on the logit becomes

$$\frac{ds_{v,e}}{d\tau} = -\eta \cdot \frac{\partial \mathcal{L}_{\text{task}}}{\partial p_{v,e}} \cdot \frac{dp_{v,e}}{ds_{v,e}} = -\eta \cdot g_{v,e} \cdot p_{v,e}(1 - p_{v,e}),$$

where

$$g_{v,e} := \text{sign}\left(\frac{\partial \mathcal{L}_{\text{task}}}{\partial p_{v,e}}\right).$$

is assumed fixed throughout training. There are two cases to consider.

**(I) $\mathbf{g_{v,e} > 0}$.** If $g_{v,e} > 0$, the gradient descent step decreases $s_{v,e}$, pushing $p_{v,e} \to 0$, which implies $(1 - p_{v,e}) \to 1$. Hence we are in the low-probability regime, where we can approximate $p_{v,e}(1 - p_{v,e}) \approx p_{v,e}$. Thus,

$$\begin{aligned}
\frac{dp_{v,e}}{d\tau} &= \frac{dp_{v,e}}{ds_{v,e}} \cdot \frac{ds_{v,e}}{d\tau} \\
&= p_{v,e}(1 - p_{v,e}) \cdot (-\eta g_{v,e} p_{v,e}(1 - p_{v,e})) \\
&= -\eta g_{v,e} \left(p_{v,e}(1 - p_{v,e})\right)^2 \\
&\approx -\eta g_{v,e} p_{v,e}^2
\end{aligned}$$

This is a first-order linear differential equation. Solving this differential equation yields

$$p_{v,e} \approx \frac{1}{\eta g_{v,e} \tau}.$$

To reach a desired threshold $p_{v,e} \leq \epsilon$, we solve:

$$\tau \geq \frac{1}{\eta g_{v,e} \epsilon} \approx \mathcal{O}(1/\epsilon).$$

**(II) $\mathbf{g_{v,e} < 0}$.** If $g_{v,e} < 0$, the gradient descent step increases $s_{v,e}$, pushing $p_{v,e} \to 1$, which implies $(1 - p_{v,e}) \to 0$. In this high-probability regime, we can approximate $p_{v,e}(1 - p_{v,e}) \approx 1 - p_{v,e}$. Thus,

$$\begin{aligned}
\frac{d(1 - p_{v,e})}{d\tau} &= -\frac{dp_{v,e}}{d\tau} \\
&\approx \eta |g_{v,e}|(1 - p_{v,e})^2
\end{aligned}$$

Solving this differential equation yields

$$1 - p_{v,e} \approx \frac{1}{\eta g_{v,e} \tau}$$

To reach a desired threshold $1 - p_{v,e} \leq \epsilon \Rightarrow p_{v,e} \geq 1 - \epsilon$, we require

$$\tau \geq \frac{1}{\eta g_{v,e} \epsilon} \approx \mathcal{O}(1/\epsilon)$$

Thus, convergence to an $\epsilon$-confidence threshold is in $\mathcal{O}(1/\epsilon)$. $\qquad\square$

# B  COMPUTATIONAL COMPLEXITY

We compare the per-layer computational complexity of full HGNNs with our fine-grained (EHGNN-F) and coarse-grained (EHGNN-C) sparsification variants. Let $n, m, t, d$, and $k$ denote the #nodes, #hyperedges, #node-hyperedge incidence pairs in $\boldsymbol{H}$, feature dimensions, and #node-hyperedge incidence pairs in $\tilde{\boldsymbol{H}}$ respectively. Table 6 shows that EHGNN-F reduces activation and computation overhead via incidence-level sparsification, but incurs a higher parameter cost. EHGNN-C reduces parameter overhead by scoring hyperedges via a shared MLP over pooled node features, sacrificing fine-grained control for scalability. Both variants are more scalable than full HGNN.

Let, $n = |V|$ denotes the number of nodes, $m = |E|$ denotes the number of hyperedges, $t = \sum_{e \in E} |e|$ denotes the number of incidence pairs (nonzeros in $\boldsymbol{H} \in \{0,1\}^{n \times m}$), $d$ denotes the input feature dimensionality, $k$ denotes the number of retained node–edge pairs after sparsification, $h$ denotes hidden layer size, and $\kappa$ denotes the number of hyperedges retained after sparsification, meaning, $\kappa \approx m.k/t$.

**Full HGNN.**  Node-to-edge and edge-to-node aggregations per layer is typically done via sparse matrix–dense matrix multiplication (SpMM). In particular, node-to-edge aggregation is done via $\boldsymbol{H}^T \boldsymbol{X} \in \mathbb{R}^{m \times d}$ to construct hyperedge embedding, while edge-to-node aggregation is done via $\boldsymbol{H}(\boldsymbol{H}^T \boldsymbol{X}) \in \mathbb{R}^{n \times d}$ to construct embedding of the nodes in the next layer. Both aggregations has a time complexity $\mathcal{O}(td)$. The space complexity to hold the intermediate results is $\mathcal{O}(td)$. This is because there are $t$ non-zero entries in $\boldsymbol{H}$, for each we need to conduct $d$ elementwise multiply-add operations. There is no additional parameter overhead involved in Full HGNN training.

**Fine-grained masking (EHGNN-F).**  *Time complexity:* EHGNN-F incurs computational cost in two main stages: sparsification and message passing. During sparsification, the model computes sigmoid scores for all $t$ incidence logits in $\mathcal{O}(t)$ time. To select the top-$k$ incidences, it uses top-$k$ Categorical sampling, costing $\mathcal{O}(t \log k)$. Once the top-$k$ mask is applied, message passing is executed over the reduced incidence matrix containing $k$ incidence pairs. Each such pair involves computations over feature vectors of dimension $d$, resulting in a total message passing cost of $\mathcal{O}(kd)$. Summing these components, the overall time complexity per forward pass is: $\mathcal{O}(kd + t \log k)$.

*Space complexity:* The space complexity of EHGNN-F consists of both the memory required for storing scores $s_{v,e}$ and intermediate activations stored during the forward pass. First, the model learns a scalar mask logit $s_{v,e}$ for every incidence pair (v,e), totaling $\mathcal{O}(t)$ persistent memory. During

Table 6: Time and space (activation) complexity, and parameter overhead comparison. Here $m = |E|$ is the original # hyperedges, $t = \sum_{e \in E} |e|$ is the original incidence size, $k \approx t.\kappa/m$ indicates the reduced incidence size after sparsification, and $\kappa$ indicates the #hyperedges after sparsification. Low-rank variants use rank $r \ll d$.

| Method | Time | Space/Activation | Param. overhead |
|---|---|---|---|
| Full HGNN | $\mathcal{O}(td)$ | $\mathcal{O}(td)$ | None |
| EHGNN-F (Fine) | $\mathcal{O}(t \log k + kd)$ | $\mathcal{O}(t + kd)$ | $\mathcal{O}(t)$ |
| EHGNN-F w/ cond (Fine) | $\mathcal{O}(t \log k + kd + td)$ | $\mathcal{O}(t + kd)$ | $\mathcal{O}(d^2)$ |
| EHGNN-F w/ cond (Fine, LR) | $\mathcal{O}(t \log k + kd + tr)$ | $\mathcal{O}(t + kd)$ | $\mathcal{O}(dr + mr)$ |
| EHGNN-C (Coarse) | $\mathcal{O}(td + m \log \kappa + kd)$ | $\mathcal{O}(m + kd)$ | $\mathcal{O}(m)$ |
| EHGNN-C w/ cond (Coarse) | $\mathcal{O}(td + md^2 + m \log \kappa + kd)$ | $\mathcal{O}(m + kd)$ | $\mathcal{O}(d^2)$ |
| EHGNN-C w/ cond (Coarse, LR) | $\mathcal{O}(td + mr + m \log \kappa + kd)$ | $\mathcal{O}(m + kd)$ | $\mathcal{O}(dr + mr)$ |

forward propagation, all $t$ scores are passed through a sigmoid and retained in memory for use in the straight-through estimator, contributing an additional $\mathcal{O}(t)$ temporary memory. The model then selects the top-$k$ incidences and performs message passing only over this pruned subset, requiring storage of $\mathcal{O}(kd)$ for the selected node or edge features. Thus, the total space complexity is: $\mathcal{O}(t + kd)$. This becomes prohibitive in large, dense hypergraphs where $t \gg n, m$; however, it is still more efficient than Full HGNN's space complexity $\mathcal{O}(td)$.

*Parameter overhead:* The parameter overhead in EHGNN-F originates from its fine-grained masks. For each incidence pair, the model maintains a dedicated scalar parameter. This leads to a total parameter count of: $\mathcal{O}(t)$.

**Coarse-grained masking (EHGNN-C w/ cond).** *Time Complexity:* The runtime overhead of EHGNN-C consists of edge scoring and sparse message passing. First, node features are aggregated to hyperedges via $\boldsymbol{H}^T \boldsymbol{X}$, requiring $\mathcal{O}(td)$ elementwise multiply-add operations. Then, the edge-level MLP scores each of the $m$ hyperedges in $\mathcal{O}(md^2)$ time if a single-layer MLP is used. Selecting the top-$\kappa$ edges costs $\mathcal{O}(m \log \kappa)$. Finally, sparse message passing occurs over $k$ incidence pairs costs $\mathcal{O}(kd)$. Thus, the total time complexity per forward pass is: $\mathcal{O}(td + md^2 + m \log \kappa + kd)$. This reflects an efficient sparsification process, in particular for dense hypergraphs where the number of hyperedges $m \ll t$.

*Space Complexity:* EHGNN-C requires memory primarily for intermediate activations and edge-level mask scores. During message passing, only the top-$\kappa$ hyperedges are retained which contribute to roughly $k$ incidence pairs that participate in message passing. Thus, message-passing activations are stored for $k$ node–edge pairs, each of feature size $d$, totaling $\mathcal{O}(kd)$. Additionally, the model stores $m$ scalar scores ($s_e$) for edge selection, which adds $\mathcal{O}(m)$ space. The total space complexity during forward propagation is: $\mathcal{O}(m + kd)$. This is significantly better than EHGNN-F's space complexity in dense hypergraphs where $m \ll t$.

*Parameter overhead:* The parameter overhead of EHGNN-C comes from its coarse-grained scoring MLP, which is applied once per hyperedge. For a single shared MLP (across all edges), the number of parameters is $\mathcal{O}(d^2)$, assuming one hidden layer. More importantly, the parameter count is independent of the number of node-edge incidences $t$. This makes EHGNN-C far more parameter-efficient than EHGNN-F, especially for large-scale hypergraphs with millions of incidences but only thousands of hyperedges.

## C    LOW-RANK PARAMETERIZATIONS OF FEATURE-CONDITIONED EHGNN VARIANTS

This section provides additional details on the lightweight, memory-efficient parameterizations of the feature-conditioned sparsifiers used in our revised experiments. These variants preserve the functional role of the original EHGNN-F(cond) and EHGNN-C(cond) scorers while significantly reducing memory usage on large and dense hypergraphs. The core sparsification mechanism (learning $p_{v,e}$ or $p_e$, task-guided top-$k$, straight-through estimation) remains unchanged, while only the parameterization of the scoring function is modified for scalability.

### C.1    LOW-RANK FEATURE-CONDITIONED INCIDENCE SCORING: EHGNN-F(COND,LR)

In the main paper, feature-conditioned EHGNN-F defines incidence scores by evaluating

$$p_{v,e} = \sigma\left(\texttt{MLP}_\theta(\boldsymbol{X}_v, \hat{\boldsymbol{X}}_e)\right), \tag{10}$$

where $\boldsymbol{X}_v \in \mathbb{R}^F$ is the feature of node $v$, $\hat{\boldsymbol{X}}_e$ is hyperedge representation pooled from $\{\boldsymbol{X}_v : v \in e\}$, and $\rho_\theta$ is a small MLP. Here, we assumed $e = \{v_1, v_2, \cdots, v_k\}$. When the number of incidences $|I| = \sum_e |e|$ is large, evaluating $\texttt{MLP}_\theta$ across all incidences induces high activation memory.

To obtain a scalable scorer while preserving the conditioning on node features, we replace $\texttt{MLP}_\theta$ with a rank-$r$ bilinear parameterization:

$$\boldsymbol{u}_v = \boldsymbol{W}_x \boldsymbol{X}_v \in \mathbb{R}^r, \qquad \boldsymbol{w}_e \in \mathbb{R}^r, \tag{11}$$

where $\boldsymbol{W}_x \in \mathbb{R}^{r \times F}$ is a learnable projection matrix and $\boldsymbol{w}_e$ is a learnable hyperedge-specific embedding.

The incidence logit is then computed as

$$\ell_{v,e} = \langle \boldsymbol{u}_v, \boldsymbol{w}_e \rangle = \boldsymbol{X}_v^\top \boldsymbol{W}_x^\top \boldsymbol{w}_e, \qquad p_{v,e} = \sigma(\ell_{v,e}). \tag{12}$$

This parameterization constrains the unnormalized score matrix

$$\boldsymbol{L} = \boldsymbol{X} \boldsymbol{W}_x^\top \boldsymbol{W}_e^\top \in \mathbb{R}^{|V| \times |E|}$$

to satisfy $rank(L) \leq r$. Thus, Equation 12 constitutes a low-rank approximation of the feature-conditioned scorer $\texttt{MLP}_\theta$ used in EHGNN-F(cond). The score computation still depends on node features $x_v$, each hyperedge retains a learnable embedding $w_e$, and the masking and straight-through estimator remain unchanged. Only the parametrization changes; the sparsifier logic stays identical.

**Relation to EHGNN-F (cond).** EHGNN-F (cond) computes

$$p_{v,e} = \sigma \left( \texttt{MLP}_\theta \left( [\boldsymbol{X}_v \parallel \hat{\boldsymbol{X}}_e] \right) \right), \tag{13}$$

where $\texttt{MLP}_\theta$ is an MLP.

The low-rank formulation in Equation 12 is a restricted version of Equation 13, where instead of an unconstrained MLP, the scorer is approximated by the bilinear form $\boldsymbol{X}_v^\top \boldsymbol{W}_x^\top \boldsymbol{w}_e$. This offers a significantly lower activation memory (no per-incidence hidden layers), and parameter efficiency ($\mathcal{O}(Fr + |E|r)$).

## C.2 Low-Rank Feature-Conditioned edge Scoring: EHGNN-C(cond,LR)

EHGNN-C(cond) computes a single score per hyperedge as follows:

$$p_e = \sigma \left( \texttt{MLP}_\theta \left( \phi(\boldsymbol{X}_e) \right) \right), \tag{14}$$

where $\boldsymbol{X}_e = \{ \boldsymbol{X}_v : v \in e \}$ are features of nodes incident to $e$, and $\phi$ is a permutation-invariant aggregator (e.g. mean aggregation). The hyperedge score $p_e$ is then used to compute the sparsified hypergraph.

To reduce memory, we replace $\texttt{MLP}_\theta$ with a rank-$r$ bilinear form. We first aggregate feature-projected node embeddings:

$$\boldsymbol{u}_e = \phi(\{ \boldsymbol{W}_x \boldsymbol{X}_v \mid v \in e \}) \in \mathbb{R}^r, \tag{C.6}$$

where $\boldsymbol{W}_x \in \mathbb{R}^{r \times F}$ and $\boldsymbol{w}_e$ is a learnable hyperedge-specific embedding.

Then we compute

$$\ell_e = \langle \boldsymbol{u}_e, \boldsymbol{w}_e \rangle, \qquad p_e = \sigma(\ell_e). \tag{15}$$

This produces hyperedge-level probabilities similar to original EHGNN-C (cond) but avoids the cost of evaluating an MLP on each hyperedge. The aggregation function $\phi$ remains the same, and the learned scores $p_e$ participate in top-$k$ hyperedge selection similar to EGHNN-C (cond).

## D    Parameter settings of EHGNN-C and EHGNN-F

Recall that EHGNN-C (cond.) and EHGNN-F (cond.) passes the node features of the nodes in a hyperedge to an MLP (shared with other hyperedges) to compute edge-level scores for each hyperedge. Table 7 reports the #neurons in the hidden layer of the MLP on various datasets. The reported setting produced the best accuracy on the test set.

## E    Ablation Studies

We analyze the various design choices for EHGNN-F. To that end, we conduct two studies: (a) whether it is worth normalizing $\sigma(s_{v,e})$ during the computation of probabilities $p_{v,e}$, and (b) whether the Bernoulli sampling performs better than Top-$k$ Categorical sampling.

Table 7: Parameter settings of EHGNN-C (cond.) and EHGNN-F (cond.)

| Dataset | # Hidden layers of MLP |
|---------|------------------------|
| 20news | 8 |
| Actor | 16 |
| CiteSeer | 16 |
| Cora | 32 |
| Cora-CA | 8 |
| DBLP-CA | 16 |
| House | 8 |
| ModelNet40 | 16 |
| Mushroom | 32 |
| NTU | 16 |
| Pokec | 32 |
| PubMed | 32 |
| Twitch | 16 |
| Walmart | 32 |
| Yelp | 16 |

Table 8: Comparison (accuracy $\pm$ std) of EHGNN-F and its variant without Probabilistic Normalization across datasets.

| Model | 20news | ModelNet40 | Mushroom | NTU2012 | Actor | Citeseer | Cora-CA | DBLP-CA |
|-------|--------|------------|----------|---------|-------|----------|---------|---------|
| EHGNN-F w/o Norm. | **77.96 ± 0.10** | **95.28 ± 0.05** | 96.30 ± 0.49 | 87.28 ± 0.42 | **76.82 ± 0.21** | 70.87 ± 0.72 | 80.71 ± 0.43 | **88.44 ± 0.08** |
| EHGNN-F | 77.94 ± 0.11 | 95.24 ± 0.07 | **96.35 ± 0.41** | **87.40 ± 0.18** | 76.68 ± 0.14 | **70.94 ± 0.37** | **80.77 ± 0.46** | 88.37 ± 0.07 |

| | Cora | House | Pokec | Pubmed | Twitch | Walmart | Yelp | |
|-------|------|-------|-------|--------|--------|---------|------|---|
| EHGNN-F w/o Norm. | **75.92 ± 0.43** | 87.80 ± 1.19 | 58.76 ± 0.03 | 85.57 ± 0.07 | 50.71 ± 0.07 | 95.16 ± 0.02 | **30.92 ± 0.14** | |
| EHGNN-F | 75.42 ± 0.57 | **87.93 ± 0.22** | **58.82 ± 0.09** | **85.59 ± 0.03** | **50.72 ± 0.11** | **95.17 ± 0.02** | 30.57 ± 0.46 | |

(a) In Table 8, we observe that in most of the datasets (9 out of 15), normalization slightly helps improve the performance. As normalization is a constant-time operation, we decided to do so in EHGNN-F for an additional boost in performance. (b) In Figure 4, we find that the Bernoulli sampling, along with enforcing the budget constraint via L2 regularization, does not boost accuracy, but rather consumes more memory. Thus, instead, we opted for Top-$k$ Categorical sampling in EHGNN-F.

## F  ADDITIONAL EXPERIMENTS

### F.1  EFFECTIVENESS ON HETEROPHILIC HYPERGRAPHS.

Li et al. (2025) proposed a synthetic dataset containing hypergraphs with various homophily ratios. We compare EHGNN-F and EHGNN-C with this dataset to understand the effectiveness of EdgeMask-HGNN on heterophilic hypergraphs. The results are presented in Figure 5.

On heterophilic hypergraphs (e.g., homophily ratio = 0.3), where connected nodes often have dissimilar labels, EHGNN-F slightly outperforms EHGNN-C. This suggests that a more flexible fine-grained masking may better preserve diverse cross-class connections, which are important for information propagation in heterophilic settings. In contrast, EHGNN-C may prematurely prune such informative edges due to its more selective sparsification criteria. The performance of EHGNN-C becomes slightly better than EHGNN-F as hypergraphs become more homophilic (e.g., homophily ratio = 0.9).

Furthermore, we have analyzed the performance gain of our method relative to Full training by examining the homophily ratios across various hypergraph datasets. Figure 7 suggests that the proposed models perform better in low homophily (high heterophily) regimes than in high homophily regimes. On EHGNN-C, the Pearson correlation coefficient between %improvement vs edge homophily and node homophily is $-0.69$ and $-0.64$, respectively, and on EHGNN-F it is $-0.67$ and $-0.62$, respectively.

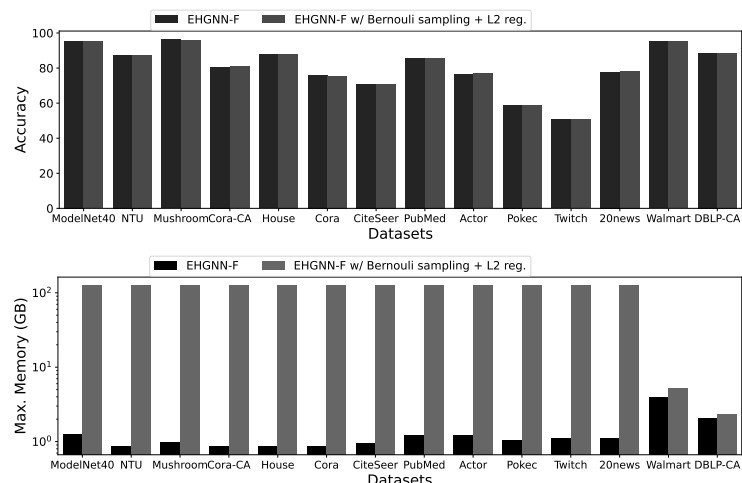

Figure 4: Ablation study of hard mask sampling from soft probabilities.

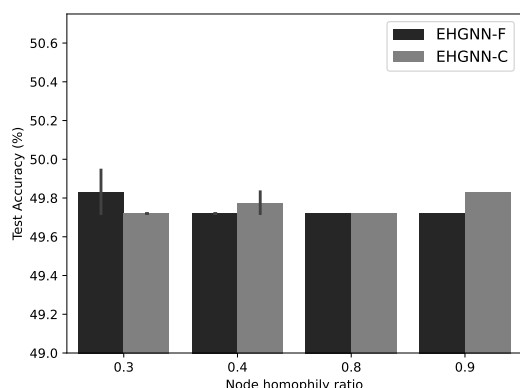

Figure 5: Performance of EdgeMask-HGNN on hypergraphs with various node homophily ratios.

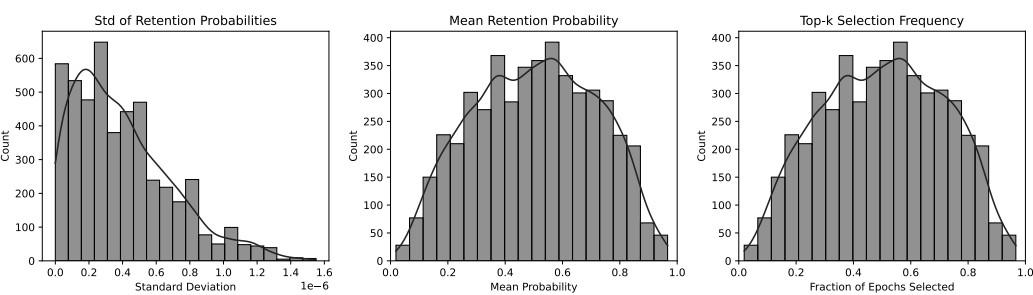

Figure 6: Analysing the retention probabilities of EHGNN-F on Cora (sparsity = 1%)

### F.2 CONVERGENCE OF RETENTION PROBABILITIES.

We analyze the retention probabilities $p_{v,e}$ in EHGNN-F to better understand their convergence behavior. Figure 6 presents the results regarding (a) the standard deviation of $p_{v,e}^{(t)}$ across epochs $t$ (left plot), (b) the mean $p_{v,e}^{(t)}$ across epochs $t$. (middle plot), and (c) the proportion of epochs where the pairs $(v, e)$ were selected to be in the top-$k$ (right plot).

(a) From the left plot, we observe that the vast majority of $(v, e)$ pairs have low standard deviation, indicating that their retention probabilities remain stable across training epochs. This demonstrates

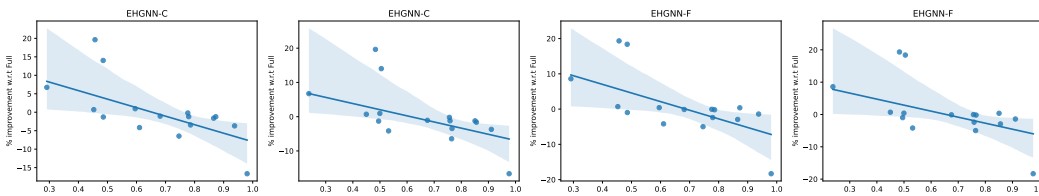

Figure 7: Correlation between Node and Edge homophily vs performance gain of EHGNN-C and EHGNN-F. EdgeMask-HGNN variants offer more accuracy gain near low homophily/high heterophily regimes ($< 0.6$).

Table 9: Correlation between accuracy difference, Accuracy(EHGNN variant) - Accuracy(Full) and hypergraph statistics.

| Metric | EHGNN-C Pearson | EHGNN-C Spearman | EHGNN-F Pearson | EHGNN-F Spearman |
|---|---|---|---|---|
| $\mathcal{H}_e$ | -0.657 | -0.791 | -0.616 | -0.665 |
| $\mathcal{H}_n$ | -0.603 | -0.818 | -0.558 | -0.680 |
| $\bar{e}_{edge}$ | -0.160 | 0.224 | -0.152 | 0.263 |
| Density | 0.097 | -0.206 | 0.169 | -0.088 |
| Sparsity | -0.171 | -0.035 | -0.172 | -0.122 |
| $n = |V|$ | -0.398 | 0.103 | -0.482 | -0.057 |
| $m = |E|$ | -0.045 | 0.229 | -0.072 | 0.185 |
| $t$ | 0.041 | 0.021 | 0.022 | 0.018 |

strong convergence of the learned mask, as the model consistently assigns very similar probabilities across epochs.

(b) The mean probabilities are evenly distributed across the full $[0, 1]$ range, forming a near-uniform bell shape. This suggests the model learns a broad spectrum of importance scores, effectively distinguishing task-relevant vs. irrelevant incidence pairs. This reflects the model's ability to differentiate between task-relevant and irrelevant pairs, reinforcing the value of the learned masking.

(c) The selection frequencies also follow a wide and symmetric distribution. Some edges are consistently selected, appearing in the top-$k$ mask in most epochs. These are likely critical incidence pairs, showing high agreement between learned probabilities and sampled mask selections. While others are rarely selected, reinforcing that the model has converged to a sparse and discriminative selection pattern.

### F.3 MODEL BEHAVIOR WITH RESPECT TO HYPERGRAPH STATISTICS.

In Table 9, we analyse the correlation between accuracy deltas (accuracy(EHGNN-C)-accuracy(Full)) and (accuracy(EHGNN-F) - accuracy(Full)) separately with various hypergraph properties such as edge homophily ratio ($\mathcal{H}_e$), node homophily ratio ($\mathcal{H}_n$), average edge size ($\bar{e}_{edge}$), hypergraph density, sparsity, the number of nodes ($n$), hyperedges ($m$) and incidences ($t$).

The results show a consistent pattern: Accuracy deltas for both EHGNN-C and EHGNN-F exhibit strong negative correlations with edge and node homophily. For EHGNN-C, the Spearman correlation with edge homophily is $-0.791$, and node homophily is $-0.818$. For EGHNN-F, the Spearman correlation with edge homophily is $-0.665$ and node homophily is $-0.680$. Scale-related statistics $(n, m, t)$ show near-zero correlation with accuracy deltas. Density, sparsity, and average edge size show weak or inconsistent effects, confirming that homophily is the dominant predictor of changes in accuracy.

Table 10: Comparison (Accuracy, peak memory) between exact and approximate effective resistance based sparsification.

| | DBLP-CA | Yelp | Walmart |
|---|---|---|---|
| **Spectral (exact)** | 84.83, 0.96 GB (10% reduction) | OOM | OOM |
| **Spectral (approximate)** | 81.95, 0.91 GB (15% reduction) | 28.67, 16.68 GB (15% reduction) | 70.68, 1.71 GB (39% reduction) |

Table 11: Number of model parameters (integers) across datasets for each algorithm at sparsity = 50%. OOM=Out-of-Memory.

| Algorithms | 20news | ModelNet40 | Mushroom | NTU2012 | Actor | Citeseer | Cora-CA | DBLP-CA |
|---|---|---|---|---|---|---|---|---|
| Full | 53764 | 72745 | 12802 | 86083 | 27651 | 1899526 | 737799 | 733190 |
| Random | 53764 | 72745 | 12802 | 86083 | 27651 | 1899526 | 737799 | 733190 |
| Degdist | 53764 | 72745 | 12802 | 86083 | 27651 | 1899526 | 737799 | 733190 |
| Spectral | 53764 | 72745 | 12802 | 86083 | 27651 | 1899526 | 737799 | OOM |
| HSL | 88718 | 91123 | 23024 | 92813 | 82153 | 557238 | 262203 | 261098 |
| EHGNN-F | 119215 | 134300 | 53422 | 96143 | 81023 | 1902979 | 742384 | 832751 |
| EHGNN-F (cond.) | 60229 | 79210 | 14275 | 92548 | 30916 | 2136583 | 829576 | 824455 |
| EHGNN-F (cond, LR) | 54564 | 122389 | 14082 | 94531 | 68507 | 1918654 | 747819 | 828342 |
| EHGNN-C | 119215 | 134300 | 53422 | 96143 | 81023 | 1902979 | 742384 | 832751 |
| EHGNN-C (cond.) | 57029 | 76010 | 13571 | 89348 | 29316 | 2018087 | 783720 | 778855 |
| EHGNN-C (cond, LR) | 54564 | 122389 | 14082 | 94531 | 68507 | 1918654 | 747819 | 828342 |
| | Cora | House | Pokec | Pubmed | Twitch | Walmart | Yelp | Trivago |
| Full | 737799 | 2562 | 34818 | 258051 | 5122 | 11787 | 958473 | 498848 |
| Random | 737799 | 2562 | 34818 | 258051 | 5122 | 11787 | 958473 | 236192 |
| Degdist | 737799 | 2562 | 34818 | 258051 | 5122 | 11787 | 958473 | 236192 |
| Spectral | 737799 | 2562 | 34818 | 258051 | 5122 | OOM | OOM | OOM |
| HSL | 262203 | 75848 | 84038 | 54637 | 76498 | 1094051 | 144919 | 124858 |
| EHGNN-F | 742585 | 14405 | 40320 | 292680 | 21478 | 472417 | 5482067 | 1225709 |
| EHGNN-F (cond.) | 829576 | 2755 | 39043 | 290116 | 5635 | 12556 | 1077706 | 518113 |
| EHGNN-F (cond, LR) | 749847 | 3934 | 44702 | 291903 | 15658 | 291455 | 3683129 | 1432856 |
| EHGNN-C | 742585 | 14405 | 40320 | 292680 | 21478 | 472417 | 5482067 | 732050 |
| EHGNN-C (cond.) | 783720 | 2691 | 36963 | 274116 | 5411 | 12204 | 1018122 | 508513 |
| EHGNN-C (cond, LR) | 749847 | 3934 | 44702 | 291903 | 15658 | 291455 | 3683129 | 1432856 |

### F.4 APPROXIMATE EFFECTIVE RESISTANCE SPARSIFICATION.

In addition to the exact effective-resistance-based sparsifier, which requires a dense pseudoinverse of the hypergraph Laplacian, we implemented an approximate spectral sparsifier using a Hutchinson-type random projection estimator for $\text{diag}(L^+)$. Specifically, we sample vectors $g$ from Rademacher distribution, solve $(L + \lambda I)z = g$ via Conjugate Gradient, and approximate $\text{diag}(L^+) \approx \mathbb{E}[z \odot g]$. These approximate resistances are then aggregated per hyperedge as in the exact method. This follows the standard line of random-projection-based spectral sparsification and avoids forming the pseudoinverse explicitly.

In practice, this approximation yields some loss in accuracy compared to full training while being significantly more scalable. Table 10 shows that on DBLP-CA it reduces memory consumption than exact sparsification, while enables training on Yelp and Walmart.

## G MODEL PARAMETER SIZES

We report the model parameter sizes in Table 11 for a complete understanding of the parameter overhead, which was discussed earlier theoretically. Recall that EHGNN-C and EHGNN-F have parameter overhead of $\mathcal{O}(d^2)$ and $\mathcal{O}(t)$, respectively, where $d$ represents the dimension of node features and $t = \sum_{e \in E} |e|$ is the number of node–hyperedge pairs. Note that $d$ does not depend on $|V|$ or $|E|$ and it's a constant, but $t$ does. Hence, we observe that EHGNN-C has a smaller parameter overhead than EHGNN-F in general.

EHGNN-F has a higher parameter overhead than Full, Random, Degdist, and Spectral due to the fact that it needs to learn the mask conditioned on the supervision signal from the downstream task, requiring additional parameters. While EHGNN-F introduces a higher parameter overhead, its mem-

ory usage ($\mathcal{O}(t + kd)$) is dominated by sparsified message passing layers. The substantial reduction in active node–hyperedge incidences ($k \ll t$) leads to a much smaller activation footprint. Thus, despite having more parameters, EHGNN-F consumes roughly equal and oftentimes less memory than the Full training (see Table 3).

## H  EVALUATING EDGEMASK-HGNN ON NODE CLUSTERING TASK

We adopt an unsupervised autoencoder framework, where the encoder learns node embeddings $\boldsymbol{Z} \in \mathbb{R}^{n \times d}$ from the hypergraph. The decoder reconstructs the incidence structure $\boldsymbol{H}$ from the embeddings. The encoder-decoder are trained based on reconstruction loss (e.g. Binary cross-entropy loss). Finally, we run a standard clustering algorithm (e.g., $k$-means) on the learned embedding $\boldsymbol{Z}$.

**Encoder.**  Given the hypergraph $\mathcal{H} = (\boldsymbol{X}, \boldsymbol{H})$, an HGNN (Feng et al., 2019) encoder $f_\theta$ produces node embeddings:

$$\boldsymbol{Z} = f_\theta(\boldsymbol{X}, \boldsymbol{H}) \in \mathbb{R}^{n \times d}.$$

The encoder may also include a sparsification mask (EHGNN-F or EHGNN-C), in which case the effective incidence is:

$$\tilde{\boldsymbol{H}} = \boldsymbol{H} \odot \boldsymbol{M}, \quad \boldsymbol{M} \in \{0,1\}^{n \times m},$$

where $\boldsymbol{M}$ is the learned binary mask as discussed in section 4.

**Decoder.**  The decoder reconstructs the incidence matrix $\boldsymbol{H}$ from $\boldsymbol{Z}$. Each hyperedge $e \in E$ is represented by aggregating its node embeddings:

$$\boldsymbol{h}_e = \frac{1}{|e|} \sum_{v \in e} \boldsymbol{Z}_v, \quad \boldsymbol{h}_e \in \mathbb{R}^d.$$

The probability that node $v$ belongs to hyperedge $e$ is modeled as:

$$\hat{\boldsymbol{H}}_{v,e} = \sigma\left(\langle \boldsymbol{W}_n \boldsymbol{Z}_v, \boldsymbol{W}_e \boldsymbol{h}_e \rangle\right),$$

where $W_n, W_e \in \mathbb{R}^{d \times d}$ are learnable projections of the decoder, $\langle \cdot, \cdot \rangle$ denotes the dot product, and $\sigma$ is the sigmoid activation function.

**Reconstruction Loss.**  We sample the same number of positive incidences $(v, e)$ where $H_{v,e} = 1$, and negatives $(v, e)$ where $H_{v,e} = 0$. The reconstruction loss is binary cross-entropy loss:

$$\mathcal{L}_{\text{recon}} = - \sum_{(v,e) \in \Omega^+} \log \hat{H}_{v,e} - \sum_{(v,e) \in \Omega^-} \log(1 - \hat{H}_{v,e}),$$

where $\Omega^+ = \{(v,e) : H_{v,e} = 1\}$ and $\Omega^-$ is a set of sampled negatives.

The encoder parameters $\theta$, and decoder projections $(W_n, W_e)$ are optimized jointly to minimize $\mathcal{L}_{\text{recon}}$.

**Observations.**  We observe that all algorithms (including Full) perform poorly on heterophilic hypergraph benchmarks (Actor, Pokec, twitch). We believe the reason is the following: on heterophilic hypergraphs node neighbors often have different labels which renders laplacian-style averaging over incident nodes under homophily assumption ineffective. As there are currently no studies on the node-clustering performance of HGNNs on heterophilic hypergraphs, addressing this issue not only under full training but also sparsified training setting would be an interesting future work.

Table 12: Node clustering Accuracy (± std) across datasets. Bold denotes the best non-*Full* result per dataset. Avg. Rank computed over datasets where all non-*Full* algorithms report accuracies (10 datasets: actor, citeseer, cora, house-committees, ModelNet40, Mushroom, NTU2012, pokec, pubmed, twitch).

| Algorithms | actor | citeseer | cora | house-committees | ModelNet40 | Mushroom | NTU2012 | pokec | pubmed | twitch | Avg. Rank |
|---|---|---|---|---|---|---|---|---|---|---|---|
| Full | 0.41 ± 0.02 | 0.41 ± 0.02 | 0.41 ± 0.03 | 0.67 ± 0.00 | 0.93 ± 0.00 | 0.84 ± 0.05 | 0.69 ± 0.01 | 0.51 ± 0.01 | 0.52 ± 0.08 | 0.51 ± 0.00 | |
| EHGNN-F | 0.41 ± 0.03 | 0.40 ± 0.05 | **0.42 ± 0.04** | **0.61 ± 0.03** | **0.92 ± 0.02** | 0.69 ± 0.07 | **0.67 ± 0.02** | **0.51 ± 0.00** | 0.53 ± 0.07 | 0.51 ± 0.00 | 2.00 |
| EHGNN-C | **0.43 ± 0.03** | 0.41 ± 0.04 | 0.41 ± 0.02 | 0.58 ± 0.05 | 0.90 ± 0.02 | **0.75 ± 0.10** | 0.67 ± 0.02 | 0.51 ± 0.00 | 0.51 ± 0.00 | 0.51 ± 0.00 | 2.60 |
| EHGNN-F (cond.) | 0.41 ± 0.03 | 0.39 ± 0.04 | 0.40 ± 0.04 | 0.59 ± 0.02 | 0.91 ± 0.01 | 0.70 ± 0.08 | 0.66 ± 0.01 | 0.51 ± 0.01 | **0.53 ± 0.07** | **0.51 ± 0.00** | 2.90 |
| EHGNN-C (cond.) | 0.42 ± 0.03 | **0.41 ± 0.04** | 0.41 ± 0.02 | 0.61 ± 0.05 | 0.91 ± 0.01 | 0.75 ± 0.10 | 0.65 ± 0.01 | 0.51 ± 0.00 | 0.53 ± 0.07 | 0.51 ± 0.00 | 2.50 |

Table 13: Node clustering NMI (± std) across datasets. Bold denotes the best non-*Full* result per dataset. Avg. Rank computed over datasets where all non-*Full* algorithms report accuracies (10 datasets: actor, citeseer, cora, house-committees, ModelNet40, Mushroom, NTU2012, pokec, pubmed, twitch).

| Algorithms | actor | citeseer | cora | house-committees | ModelNet40 | Mushroom | NTU2012 | pokec | pubmed | twitch | Avg. Rank |
|---|---|---|---|---|---|---|---|---|---|---|---|
| Full | 0.00 ± 0.00 | 0.20 ± 0.03 | 0.24 ± 0.02 | 0.22 ± 0.00 | 0.92 ± 0.00 | 0.41 ± 0.11 | 0.82 ± 0.00 | 0.00 ± 0.00 | 0.12 ± 0.06 | 0.00 ± 0.00 | |
| EHGNN-F | 0.00 ± 0.00 | 0.18 ± 0.04 | 0.22 ± 0.03 | 0.14 ± 0.04 | 0.92 ± 0.00 | 0.14 ± 0.07 | 0.80 ± 0.00 | 0.00 ± 0.00 | 0.13 ± 0.06 | 0.00 ± 0.00 | 2.30 |
| EHGNN-C | 0.00 ± 0.00 | 0.18 ± 0.03 | 0.23 ± 0.03 | 0.10 ± 0.07 | 0.91 ± 0.00 | 0.24 ± 0.12 | 0.80 ± 0.00 | 0.00 ± 0.00 | 0.12 ± 0.07 | 0.00 ± 0.00 | 2.50 |
| EHGNN-F (cond.) | 0.00 ± 0.00 | 0.17 ± 0.04 | 0.22 ± 0.04 | 0.12 ± 0.03 | 0.92 ± 0.00 | 0.15 ± 0.08 | 0.80 ± 0.01 | 0.00 ± 0.00 | 0.14 ± 0.06 | 0.00 ± 0.00 | 2.90 |
| EHGNN-C (cond.) | 0.00 ± 0.00 | 0.19 ± 0.03 | 0.23 ± 0.03 | 0.14 ± 0.08 | 0.91 ± 0.00 | 0.24 ± 0.12 | 0.80 ± 0.00 | 0.00 ± 0.00 | 0.13 ± 0.06 | 0.00 ± 0.00 | 2.30 |

Table 14: Node clustering ARI (± std) across datasets. Bold denotes the best non-*Full* result per dataset. Avg. Rank computed over datasets where all non-*Full* algorithms report accuracies (10 datasets: actor, citeseer, cora, house-committees, ModelNet40, Mushroom, NTU2012, pokec, pubmed, twitch).

| Algorithms | actor | citeseer | cora | house-committees | ModelNet40 | Mushroom | NTU2012 | pokec | pubmed | twitch | Avg. Rank |
|---|---|---|---|---|---|---|---|---|---|---|---|
| Full | 0.01 ± 0.00 | 0.16 ± 0.02 | 0.18 ± 0.03 | 0.12 ± 0.00 | 0.90 ± 0.00 | 0.48 ± 0.13 | 0.65 ± 0.01 | 0.00 ± 0.00 | 0.11 ± 0.06 | 0.00 ± 0.00 | |
| EHGNN-F | 0.01 ± 0.00 | 0.13 ± 0.03 | 0.16 ± 0.03 | 0.05 ± 0.02 | 0.91 ± 0.01 | 0.16 ± 0.10 | 0.62 ± 0.03 | 0.00 ± 0.00 | 0.13 ± 0.07 | 0.00 ± 0.00 | 2.10 |
| EHGNN-C | 0.01 ± 0.00 | 0.14 ± 0.04 | 0.16 ± 0.04 | 0.04 ± 0.05 | 0.88 ± 0.02 | 0.29 ± 0.16 | 0.63 ± 0.04 | 0.00 ± 0.00 | 0.11 ± 0.08 | 0.00 ± 0.00 | 2.60 |
| EHGNN-F (cond.) | 0.01 ± 0.00 | 0.12 ± 0.03 | 0.15 ± 0.03 | 0.04 ± 0.02 | 0.90 ± 0.01 | 0.18 ± 0.12 | 0.61 ± 0.02 | 0.00 ± 0.00 | 0.13 ± 0.07 | 0.00 ± 0.00 | 2.80 |
| EHGNN-C (cond.) | 0.01 ± 0.00 | 0.14 ± 0.03 | 0.16 ± 0.03 | 0.06 ± 0.05 | 0.90 ± 0.01 | 0.29 ± 0.16 | 0.60 ± 0.02 | 0.00 ± 0.00 | 0.12 ± 0.07 | 0.00 ± 0.00 | 2.50 |

# I   STATISTICAL SIGNIFICANCE TESTS

In order to evaluate whether the proposed EdgeMask variants differ meaningfully from the full hypergraph training baseline, we conduct paired (related) samples t-tests comparing Full vs. EHGNN-C and Full vs. EHGNN-F across all datasets (Table 15). Here, $* = p < 0.05$, $** = p < 0.01$, and $- = p > 0.05$ indicate statistical significance levels. All Edgemask-HGNN variants operate under a 50% sparsification budget.

The statistical tests reveal several clear trends. First, sparsification does not universally degrade performance. On many datasets, particularly Actor, House, Walmart, and Yelp, we observe that both EHGNN-C and EHGNN-F significantly outperform the full hypergraph model, despite using only 50% of the hyperedges. This indicates that dense hypergraphs often contain redundant or noisy high-order relations, and that principled sparsification can improve generalization by acting as a structural regularizer.

Second, datasets with strong homophily, such as Cora, Cora-CA, CiteSeer, and DBLP-CA, show significant performance drops after sparsification. Here, full connectivity provides a useful neighborhood signal, and removing hyperedges disrupts information flow.

Third, the two Edgemask variants exhibit complementary strengths. We find that EHGNN-C tends to be more effective on moderately sparse hypergraphs with large #nodes. On the contrary, EHGNN-F performs better on denser hypergraphs, where fine-grained incidence-level masking can better suppress noisy connections inside large hyperedges. This observation aligns with our correlation analysis, where the number of nodes shows a moderately strong positive rank correlation with accuracy gap (Spearman $\rho_n = 0.67$), while incidence density shows a moderately strong negative correlation ($\rho_d = -0.61$).

Overall, the results demonstrate that well-designed hypergraph sparsification can preserve or improve predictive performance while substantially reducing hyperedge count, especially in settings with noisy, redundant hyperedges.

Table 15: Statistical test by conducting paired (related) samples t-tests: Full vs. EHGNN-C and Full vs. EHGNN-F. Here $* = p < 0.05, ** = p < 0.01$ and $- = p > 0.05$ indicates no statistical significance. Edgemask variants are run with 50% sparsification.

| Dataset | Full | EHGNN-C | | | | EHGNN-F | | | | Better Method |
|---|---|---|---|---|---|---|---|---|---|---|
| | | Mean Acc. | $\Delta$ | p-value | Sig. | Mean Acc. | $\Delta$ | p-value | Sig. | |
| 20news | 81.39 | 78.01 | $-3.38$ | 2.57e-05 | ** | 77.99 | $-3.41$ | 2.60e-08 | ** | Full |
| Actor | 64.53 | 77.20 | **+12.67** | 3.90e-10 | ** | 77.00 | **+12.48** | 1.18e-10 | ** | EHGNN-C |
| CiteSeer | 68.19 | 67.46 | $-0.72$ | 4.29e-02 | * | 68.12 | $-0.07$ | 8.16e-01 | - | Full |
| Cora | 78.23 | 73.18 | $-5.05$ | 1.47e-05 | ** | 74.33 | $-3.90$ | 6.17e-05 | ** | Full |
| Cora-CA | 81.48 | 80.50 | $-0.97$ | 1.09e-02 | * | 79.53 | $-1.95$ | 9.92e-04 | ** | Full |
| DBLP-CA | 90.77 | 89.30 | $-1.46$ | 2.21e-06 | ** | 88.11 | $-2.65$ | 8.62e-07 | ** | Full |
| House | 73.75 | 84.09 | **+10.34** | 3.11e-05 | ** | 87.31 | **+13.56** | 3.60e-05 | ** | EHGNN-F |
| ModelNet40 | 94.89 | 93.74 | $-1.16$ | 7.90e-06 | ** | 95.25 | **+0.35** | 7.35e-04 | ** | EHGNN-F |
| Mushroom | 97.75 | 94.14 | $-3.61$ | 7.57e-05 | ** | 96.39 | $-1.37$ | 3.00e-05 | ** | Full |
| NTU2012 | 87.67 | 84.65 | $-3.02$ | 4.52e-05 | ** | 87.51 | $-0.16$ | 3.74e-01 | - | Full |
| Pokec | 58.37 | 58.79 | **+0.42** | 1.55e-04 | ** | 58.78 | **+0.41** | 1.16e-04 | ** | EHGNN-C/F |
| PubMed | 85.60 | 85.42 | $-0.17$ | 7.05e-03 | ** | 85.56 | $-0.04$ | 2.58e-01 | - | Full |
| Twitch | 51.23 | 50.57 | $-0.66$ | 2.35e-04 | ** | 50.74 | $-0.50$ | 2.47e-05 | ** | Full |
| Trivago | 61.68 | 51.41 | $-10.27$ | 2.08e-06 | ** | 50.38 | $-11.30$ | 5.20e-05 | ** | Full |
| Walmart | 94.78 | 95.68 | **+0.90** | 8.82e-08 | ** | 95.17 | **+0.39** | 5.28e-06 | ** | EHGNN-C |
| Yelp | 29.75 | 31.75 | **+2.00** | 6.30e-03 | ** | 32.31 | **+2.56** | 5.58e-03 | ** | EHGNN-F |

Table 16: Recommended variants based on the # node-hyperedge pairs in the hypergraph ($t$).

| Regime | Recommended Variant | Expected Performance | Representative Dataset |
|---|---|---|---|
| **Small t** $(< 20k)$ | Full HGNN | Sparsification may remove important learning signal; full model already efficient. | Cora $(t = 4,786)$ |
| **Medium** $t$ (20k–200k) | EHGNN-C or EHGNN-F | Accuracy matches or exceeds Full; moderate memory gains. | ModelNet40 $(t = 61,555)$ |
| **Large** $t$ (200k–800k) | EHGNN-F | Strong accuracy and memory gains; incidence/edge pruning reduces activation cost. | Walmart $(t = 460,630)$ |
| **Extremely large** $t$ $(> 800k)$ | EHGNN-F or EHGNN-F (cond, LR) | Feature-conditioned variants may require low-rank scorer to curb activation memory. | Yelp $(t = 4,523,594)$ |

## J    RECOMMENDED VARIANTS

In order to support decision-making regarding which variants to use, we refer to Table 16 as a guideline. Table 16 presents our observation based on the benchmarks we studied.

We also elucidate when sparsification may help, and when it may underperform:

*(I) When sparsification improves accuracy:* 1) The hypergraph is heterophilic (e.g., Actor, Pokec, House), in particular, when Edge/node homophily is roughly $< 0.6$ (See Figure 7), and 2) the hypergraph is noisy.

*(II) When sparsification maintains accuracy while reducing memory/runtime:* Large-scale but not extremely dense hypergraphs (e.g., DBLP-CA, Walmart, Yelp).

*(III) When sparsification may slightly hurt accuracy:* Highly homophilic (e.g., Cora, Citeseer, PubMed) and/or small-scale hypergraphs (e.g., 20news).

## K    EMPIRICAL CONVERGENCE AND STABILITY OF LEARNED MASKS

A central concern in learnable sparsification is whether the discrete masks $\hat{m}_{v,e}$ converge to a stable selection of hyperedges or incidences. Since the forward pass uses a hard selection (top-$k$) but the backpropagation uses a straight-through estimator (STE), instability of these surrogate gradients could lead to erratic mask behavior.

In order to verify that the learned masks truly converge during training, we track the mask flips, meaning, hyperedges whose hard mask value changes between consecutive epochs for EHGNN-C:

$$\% \text{Flip}^{(t)} = \frac{100}{|E|} \sum_{e \in E} \mathbf{1}\left[\hat{m}_e^{(t)} \neq \hat{m}_e^{(t-1)}\right],$$

where $\hat{m}_e^{(t)} \in \{0, 1\}$ denotes whether hyperedge $e$ is kept by the sparsifier at epoch $t$. If the learned sparsifier is unstable, we expect frequent flips throughout training (a higher %Flip).

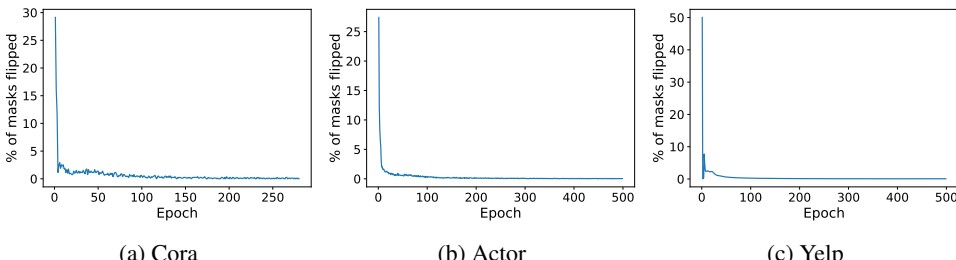

(a) Cora         (b) Actor         (c) Yelp

Figure 8: % of Learned mask flipped per epoch during training of EHGNN-C with 50% pruning rate.

Figures 8a, 8b, and 8c show the %Flip for three datasets (Cora, Actor, and Yelp), where we observe that %Flip rapidly decreases after the initial epochs and stabilizes to very low values ($< 1\%$). This demonstrates that the learned discrete mask converges to a stable sparsification pattern, with only small local perturbations as training progresses.

More importantly, mask stability is a stricter notion of convergence than gradient sign stability; in other words, if the STE gradients were noisy or inconsistent, the resulting binary decisions would fluctuate significantly. The observed mask convergence therefore provides strong empirical evidence that the STE-based optimization yields a stable update signal for the mask logits.

