., 2006). Note that there are no task-relevant methods in the HGNN literature, hence our comparison had to be limited to unsupervised approaches. The model parameters, hyperparameters of the algorithms, ablation studies, convergence of retention probabilities, and node-clustering experiments can be found in the Appendix. Our source codes: `https://github.com/toggled/ehgnn`.

## 6.1 EXPERIMENTAL EVALUATION

**I. Effectiveness and Scalability.** Table 2 highlights the effectiveness of EdgeMask-HGNN over the baselines. We observe several things:

*Supervised vs Unsupervised sparsifiers:* On most datasets, EHGNN variants consistently outperform Random and degree-based sparsification. This highlights the importance of task-aware learning of sparsification masks, which can retain task-relevant incidences or hyperedges while discarding noisy ones. On large-scale datasets, spectral methods face memory issues, which highlight the computational challenges to preserving the Laplacian spectrum.

*Full training vs Sparsification:* On several (8 out of 15) datasets, EdgeMask-HGNNs actually outperforms full HGNN training, such as ModelNet40, Actor, DBLP-CA, House, Pokec, Pubmed, Walmart, and Yelp. This demonstrates that pruning of irrelevant pairs can enhance generalization by improving the signal-to-noise ratio during message passing.

*Fine-grained vs Coarse-grained EdgeMasking and variants:* EHGNN-C is slightly better than EHGNN-F, often outperforming on datasets like 20news, CiteSeer, Cora, Cora-CA, DBLP-CA, and PubMed. Since co-authorship and co-citation datasets contain semantically coherent hyperedges (papers) and rich bag-of-words features, the model does not need fine-grained incidence-level filtering to extract meaningful training signals. Instead, pruning at the hyperedge level is sufficient. In contrast, EHGNN-F excels on ModelNet40, NTU2012 (where class counts are large), and Walmart, House, Actor, and Pokec, where hyperedges are more noisy, and may group together items or people that aren't strongly related to the prediction task. In such conditions, fine-grained incidence pruning provides a stronger signal-to-noise filtering.

**II. Memory efficiency.** Table 3 reports peak GPU memory across methods at the same sparsity budget. Note that, peak GPU usage during HGNN training is dominated by activations (messages, pooled edge features, and autograd buffers) that scale with # incidences after sparsification $k = \sum_{e \in \tilde{\mathcal{H}}} |e|$, and feature dimension $d$, not by the parameter overhead (see Table 6, Appendix B).

*On small-scale datasets:* On small datasets, all methods exhibit similar peak memory because the entire incidence structure fits comfortably on a GPU. The incidence and hyperedge counts are not large enough to pose scalability challenges, thus, sparsification strategies bring little practical dif-

Table 2: Accuracy ($\pm$ std) across different datasets for each algorithm at sparsity = 50%. OOM=Out-of-Memory. Bold (underline) denotes the best (2nd best) result per dataset not counting Full.

| Algorithms | 20news | ModelNet40 | Mushroom | NTU2012 | Actor | CiteSeer | Cora-CA | DBLP-CA |
|---|---|---|---|---|---|---|---|---|
| Full | 81.40 ± 0.04 | 94.88 ± 0.06 | 97.76 ± 0.31 | 87.67 ± 0.24 | 64.58 ± 0.03 | 69.93 ± 0.46 | 82.25 ± 0.22 | 91.14 ± 0.04 |
| Random | 55.26 ± 0.66 | 64.03 ± 1.01 | 84.19 ± 1.25 | 59.80 ± 2.55 | 65.86 ± 0.67 | 27.29 ± 1.14 | 42.72 ± 3.33 | 52.63 ± 1.90 |
| Degdist | 59.02 ± 0.60 | 50.25 ± 1.05 | 74.19 ± 1.25 | 44.53 ± 1.45 | 63.57 ± 0.16 | 40.39 ± 0.20 | 52.64 ± 2.68 | 57.64 ± 0.26 |
| Spectral | 72.80 ± 0.02 | 82.53 ± 0.03 | **97.74 ± 0.12** | 65.45 ± 0.43 | 63.93 ± 0.03 | 35.85 ± 0.28 | 64.14 ± 0.28 | OOM |
| EHGNN-F | 78.00 ± 0.04 | 95.25 ± 0.08 | 96.42 ± 0.41 | **87.40 ± 0.18** | 76.99 ± 0.03 | 67.92 ± 0.43 | 79.38 ± 0.39 | 88.00 ± 0.10 |
| EHGNN-C | **81.10 ± 0.28** | 94.90 ± 0.08 | 96.23 ± 1.13 | 87.36 ± 0.30 | 74.07 ± 0.05 | **69.03 ± 0.33** | **82.25 ± 0.28** | **91.33 ± 0.04** |
| EHGNN-F (cond.) | 78.54 ± 0.08 | **95.52 ± 0.05** | 94.92 ± 0.61 | 87.28 ± 0.54 | **78.75 ± 0.04** | 68.43 ± 0.48 | 75.95 ± 0.59 | 86.28 ± 0.09 |
| EHGNN-C (cond.) | 74.82 ± 0.26 | 94.52 ± 0.10 | 97.46 ± 0.30 | 85.33 ± 0.41 | 77.55 ± 0.04 | 67.05 ± 0.16 | 75.39 ± 0.34 | 87.11 ± 0.18 |

| | Cora | House | Pokec | PubMed | Twitch | Walmart | Yelp | Avg. Rank |
|---|---|---|---|---|---|---|---|---|
| Full | 78.35 ± 0.34 | 73.99 ± 0.44 | 58.55 ± 0.05 | 85.61 ± 0.05 | 51.22 ± 0.05 | 94.78 ± 0.02 | 30.48 ± 0.68 | |
| Random | 41.57 ± 1.28 | 61.73 ± 0.71 | 53.34 ± 0.29 | 46.24 ± 0.31 | 49.88 ± 0.78 | 55.70 ± 0.20 | 29.78 ± 0.28 | 6.29 |
| Degdist | 59.32 ± 1.01 | 52.14 ± 5.18 | 54.15 ± 0.05 | 49.04 ± 0.03 | 50.44 ± 0.59 | 56.28 ± 0.14 | 27.82 ± 0.65 | 6.00 |
| Spectral | 56.90 ± 0.12 | 57.59 ± 0.54 | 52.92 ± 0.06 | 49.07 ± 0.00 | 51.05 ± 0.12 | 65.90 ± 0.03 | OOM | 5.21 |
| EHGNN-F | 74.00 ± 0.43 | 87.93 ± 0.22 | 58.70 ± 0.04 | 85.59 ± 0.03 | 50.67 ± 0.03 | 95.16 ± 0.01 | **30.57 ± 0.46** | 2.58 |
| EHGNN-C | **77.52 ± 0.28** | 74.74 ± 0.17 | 58.46 ± 0.09 | **85.69 ± 0.05** | **51.10 ± 0.04** | 93.72 ± 0.09 | 30.12 ± 0.60 | **2.21** |
| EHGNN-F (cond.) | 73.65 ± 0.44 | **100.00 ± 0.00** | **59.14 ± 0.07** | 85.55 ± 0.07 | 50.89 ± 0.04 | **98.38 ± 0.01** | 29.13 ± 0.44 | 2.43 |
| EHGNN-C (cond.) | 75.86 ± 0.47 | 73.07 ± 0.95 | 59.01 ± 0.05 | 85.48 ± 0.04 | 50.67 ± 0.03 | 94.73 ± 0.04 | 29.17 ± 0.18 | 3.29 |

Table 3: Maximum GPU memory usage (in GB) across datasets for each algorithm at sparsity = 50%. Avg. Rank computed over all datasets with OOM (Out-of-memory) treated as the worst rank.

| Algorithms | 20news | ModelNet40 | Mushroom | NTU2012 | Actor | CiteSeer | Cora-CA | DBLP-CA |
|---|---|---|---|---|---|---|---|---|
| Full | 1.3 ± 0.0 (1.3) | 1.3 ± 0.0 (1.3) | 1.0 ± 0.0 (1.0) | 0.9 ± 0.0 (0.9) | 1.1 ± 0.0 (1.1) | 0.9 ± 0.0 (0.9) | 0.9 ± 0.0 (0.9) | 3.9 ± 0.0 (3.9) |
| Random | 1.0 ± 0.0 (1.0) | 1.0 ± 0.0 (1.0) | 0.9 ± 0.0 (0.9) | 0.8 ± 0.0 (0.8) | 1.0 ± 0.0 (1.0) | 0.9 ± 0.0 (0.9) | 0.8 ± 0.0 (0.8) | 1.5 ± 0.0 (1.5) |
| Degdist | 1.1 ± 0.0 (1.1) | 1.1 ± 0.0 (1.1) | 0.9 ± 0.0 (0.9) | 0.8 ± 0.0 (0.8) | 1.1 ± 0.0 (1.1) | 0.9 ± 0.0 (0.9) | 0.8 ± 0.0 (0.8) | 1.5 ± 0.0 (1.5) |
| Spectral | 8.0 ± 0.1 (8.1) | 7.7 ± 1.4 (8.5) | 3.0 ± 0.1 (3.0) | 1.2 ± 0.0 (1.2) | 12.0 ± 2.1 (13.3) | 1.5 ± 0.0 (1.5) | 1.3 ± 0.0 (1.3) | 64.6 ± 19.8 (76.1) |
| EHGNN-F | 1.1 ± 0.0 (1.1) | 1.1 ± 0.0 (1.1) | 1.0 ± 0.0 (1.0) | 0.8 ± 0.0 (0.8) | 1.1 ± 0.0 (1.1) | 0.9 ± 0.0 (0.9) | 0.9 ± 0.0 (0.9) | 1.7 ± 0.0 (1.7) |
| EHGNN-C | 1.2 ± 0.1 (1.3) | 1.2 ± 0.1 (1.2) | 1.0 ± 0.0 (1.0) | 0.9 ± 0.0 (0.9) | 1.2 ± 0.1 (1.2) | 0.9 ± 0.0 (0.9) | 0.9 ± 0.0 (0.9) | 2.3 ± 0.5 (2.6) |
| EHGNN-F (cond.) | 1.1 ± 0.0 (1.1) | 1.1 ± 0.0 (1.1) | 1.0 ± 0.0 (1.0) | 0.8 ± 0.0 (0.8) | 1.1 ± 0.0 (1.1) | 1.1 ± 0.0 (1.1) | 0.9 ± 0.0 (0.9) | 5.5 ± 0.0 (5.5) |
| EHGNN-C (cond.) | 1.1 ± 0.0 (1.1) | 1.1 ± 0.0 (1.1) | 1.0 ± 0.0 (1.0) | 0.8 ± 0.0 (0.8) | 1.1 ± 0.0 (1.1) | 0.9 ± 0.0 (0.9) | 0.9 ± 0.0 (0.9) | 2.4 ± 0.0 (2.4) |

| | Cora | House | Pokec | PubMed | Twitch | Walmart | Yelp | Avg. Rank |
|---|---|---|---|---|---|---|---|---|
| Full | 0.9 ± 0.0 (0.9) | 0.9 ± 0.0 (0.9) | 1.0 ± 0.0 (1.0) | 1.2 ± 0.0 (1.2) | 1.0 ± 0.0 (1.0) | 13.3 ± 0.0 (13.3) | 98.9 ± 21.2 (111.2) | 4.25 |
| Random | 0.8 ± 0.0 (0.8) | 0.8 ± 0.0 (0.8) | 0.9 ± 0.0 (0.9) | 1.0 ± 0.0 (1.0) | 0.9 ± 0.0 (0.9) | 5.2 ± 0.7 (5.6) | 38.9 ± 0.1 (38.9) | 1.92 |
| Degdist | 0.9 ± 0.0 (0.9) | 0.8 ± 0.0 (0.8) | 0.9 ± 0.0 (0.9) | 1.1 ± 0.0 (1.1) | 0.9 ± 0.0 (0.9) | 8.0 ± 0.0 (8.0) | 104.4 ± 48.1 (130.1) | 2.83 |
| Spectral | 1.3 ± 0.0 (1.3) | 1.1 ± 0.0 (1.1) | 8.6 ± 1.2 (9.3) | 15.5 ± 2.5 (17.0) | 10.4 ± 1.4 (11.3) | 135.8 ± 2.2 (135.2) | OOM | 7.75 |
| EHGNN-F | 0.9 ± 0.0 (0.9) | 0.8 ± 0.0 (0.8) | 1.0 ± 0.0 (1.0) | 1.1 ± 0.0 (1.1) | 1.0 ± 0.0 (1.0) | 7.2 ± 0.0 (7.2) | 110.0 ± 47.2 (135.7) | 3.00 |
| EHGNN-C | 0.9 ± 0.0 (0.9) | 0.9 ± 0.0 (0.9) | 1.0 ± 0.0 (1.0) | 1.2 ± 0.0 (1.2) | 1.0 ± 0.0 (1.0) | 7.4 ± 0.0 (7.4) | 104.9 ± 22.3 (117.8) | 3.50 |
| EHGNN-F (cond.) | 0.9 ± 0.0 (0.9) | 0.8 ± 0.0 (0.8) | 1.0 ± 0.0 (1.0) | 1.2 ± 0.0 (1.2) | 1.0 ± 0.0 (1.0) | 7.1 ± 0.0 (7.1) | 138.2 ± 0.3 (138.1) | 3.83 |
| EHGNN-C (cond.) | 0.9 ± 0.0 (0.9) | 0.8 ± 0.0 (0.8) | 1.0 ± 0.0 (1.0) | 1.1 ± 0.0 (1.1) | 1.0 ± 0.0 (1.0) | 83.3 ± 65.5 (101.7) | 118.1 ± 37.3 (139.5) | 3.92 |

ference in this regime. We also find that Spectral methods require noticeably large memory in this regime.

*On large, but less-dense hypergraphs (DBLP-CA, Walmart):* The differences emerge clearly on large hypergraphs where the activations dominate the memory footprint. Spectral sparsification is the most memory-hungry due to Laplacian computations on dense adjacency matrix. Random and degree-based pruning yield good memory savings, but their accuracy significantly degrades compared to Full training. EHGNN-F and EHGNN-C are better alternatives, since with a similar memory footprint as Random and degree-based pruning, they yield better accuracy.

*On large, but more-dense hypergraph (Yelp):* As density increases, the additional buffers for mask parameters and stochastic sampling cause peak activation memory to also increase. As a result, all EHGNN variants consume more memory than the full HGNN on Yelp. The effect is pronounced for feature-conditioned variants, where pooling and scorer MLPs introduce extra activations, while coarse EHGNN-C without conditioning shows smaller overhead but still exceeds the Full baseline.

*Fine- versus coarse-grained masking:* Among our methods, EHGNN-F attains the best average rank (3.00) and shows the most stable reductions overall, since incidence-level pruning directly lowers the number of active node–edge pairs $k$. EHGNN-C (3.50) can also save memory by keeping fewer parameters and retaining smaller edges, but due to randomness in sampling, sometimes it may cause large activations (large $k$) by retaining large edges, consequently increasing the memory footprint.

To summarize, memory footprint is determined by the # retained incidences $k$ and per-layer activations. EHGNN-F directly reduces $k$ and is therefore the most reliable way to shrink peak memory. EHGNN-C reduces $m$ (#edges) but can retain large hyperedges, keeping $k$ high. Feature conditioning at the hyperedge level adds pooling/MLP activations and can further increase peak usage.

**III. Accuracy/Runtime vs sparsity trade-off.** We analyze the trade-off between accuracy (on DBLP-CA) and sparsity due to edge pruning by our proposed EdgeMask-HGNN in Figure 3a, while the impact of sparsity on the end-to-end runtime (Training + Evaluation time) is presented in Figure 3b. We observe that with more edges retained in the sparsification, the accuracy increases at the cost of a higher runtime.

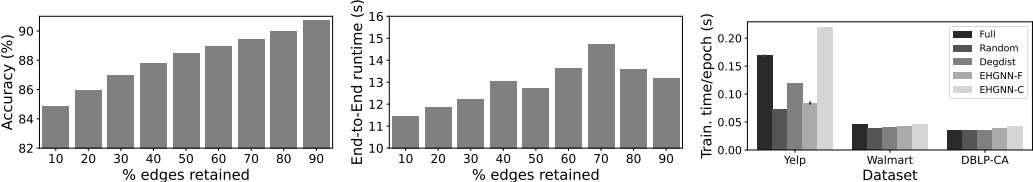

(a) Accuracy vs. Sparsity      (b) End-to-end runtime vs. Sparsity      (c) Training time

Figure 3: (a-b) Impact of sparsity on EHGNN-F's performance (DBLP-CA dataset) (c) Training time efficiency of the methods.

**IV. Runtime efficiency.** Figure 3c compares the average training time/epoch across three large-scale datasets. On Yelp, the full model incurs the highest cost ($\sim$0.17s), while EHGNN-F reduces

Table 4: End-to-end execution time (in seconds) as mean ± std across datasets for each algorithm at sparsity = 50%.

| Algorithms | 20news | ModelNet40 | Mushroom | NTU2012 | Actor | Citeseer | Cora-CA | DBLP-CA |
|---|---|---|---|---|---|---|---|---|
| Full | 18.37 ± 0.37 | 17.43 ± 1.08 | 17.39 ± 0.99 | 16.98 ± 1.10 | 16.90 ± 0.40 | 8.20 ± 0.14 | 10.39 ± 1.11 | 12.75 ± 0.33 |
| Random | 17.99 ± 0.38 | **16.90 ± 0.32** | 17.02 ± 0.44 | **16.78 ± 0.26** | **16.91 ± 0.32** | **7.57 ± 0.25** | **8.92 ± 0.61** | **11.29 ± 0.12** |
| Deg | 16.84 ± 0.40 | 17.10 ± 0.32 | **16.96 ± 0.27** | 16.92 ± 0.36 | 18.35 ± 0.55 | 7.76 ± 0.17 | 9.92 ± 0.63 | 12.43 ± 0.44 |
| Spectral | 74.98 ± 0.30 | 45.34 ± 1.28 | 25.17 ± 1.02 | 18.38 ± 1.35 | 88.38 ± 0.21 | 8.42 ± 0.19 | 10.66 ± 1.26 | 18.64 ± 0.32 |
| EHGNN-F | 17.89 ± 0.49 | 18.52 ± 0.72 | 18.22 ± 0.79 | 18.06 ± 0.72 | 18.50 ± 0.79 | 9.16 ± 0.49 | 10.84 ± 0.34 | 12.93 ± 0.70 |
| EHGNN-C | 17.50 ± 0.61 | 19.08 ± 0.34 | 17.52 ± 0.34 | 18.27 ± 0.32 | 19.28 ± 0.19 | 8.86 ± 0.24 | 11.15 ± 0.79 | 13.76 ± 0.78 |
| EHGNN-F (cond.) | 19.18 ± 0.13 | 18.39 ± 0.46 | 18.15 ± 0.47 | 17.90 ± 0.44 | 18.57 ± 0.73 | 9.32 ± 0.21 | 10.68 ± 0.43 | 14.51 ± 0.50 |
| EHGNN-C (cond.) | **16.22 ± 1.27** | 18.53 ± 0.37 | 18.15 ± 0.32 | 17.89 ± 0.36 | 18.49 ± 0.24 | 8.64 ± 0.45 | 11.90 ± 0.62 | 13.77 ± 0.43 |
| | Cora | House | Pokec | PubMed | Twitch | Walmart | Yelp | Avg. Rank |
| Full | 10.14 ± 1.23 | 18.23 ± 1.13 | 15.99 ± 1.09 | 17.25 ± 0.77 | 11.58 ± 3.04 | 22.82 ± 3.70 | 84.42 ± 1.25 | |
| Random | **8.94 ± 0.39** | 17.84 ± 0.26 | **7.62 ± 0.47** | **17.61 ± 0.46** | **7.98 ± 0.93** | **19.30 ± 0.35** | 35.88 ± 0.28 | **1.53** |
| Deg | 9.65 ± 0.10 | **16.01 ± 1.86** | 7.66 ± 0.51 | 17.64 ± 0.34 | 12.47 ± 4.02 | 20.19 ± 0.32 | 59.95 ± 2.60 | 2.13 |
| Spectral | 10.04 ± 0.26 | 18.48 ± 0.64 | 12.83 ± 0.38 | 98.80 ± 0.89 | 9.70 ± 0.24 | 62.17 ± 0.94 | OOM | 5.43 |
| EHGNN-F | 11.41 ± 0.52 | 17.36 ± 0.58 | 17.72 ± 0.29 | 18.50 ± 0.84 | 12.58 ± 3.32 | 20.84 ± 0.57 | 41.97 ± 2.92 | 4.47 |
| EHGNN-C | 10.28 ± 0.66 | 17.91 ± 0.72 | 15.91 ± 2.28 | 17.95 ± 0.30 | 14.32 ± 4.27 | 21.15 ± 0.45 | 39.75 ± 0.36 | 4.53 |
| EHGNN-F (cond.) | 10.95 ± 0.34 | 19.11 ± 0.56 | 17.90 ± 0.73 | 18.56 ± 0.15 | 14.19 ± 3.93 | 21.79 ± 0.71 | 162.61 ± 11.45 | 5.43 |
| EHGNN-C (cond.) | 11.43 ± 1.02 | 17.73 ± 0.26 | 17.71 ± 1.12 | 18.13 ± 0.26 | 11.60 ± 3.38 | 22.52 ± 0.73 | 109.33 ± 36.23 | 4.37 |

Table 5: Comparing various state-of-the-art HGNNs and their EHGNN-F enhanced counterparts (sparsity = 50%).

| Models | Actor | Cora | Cora-CA | House | ModelNet40 | NTU | PubMed |
|---|---|---|---|---|---|---|---|
| AllSetTrans. | 68.77 ± 0.60 | **76.93 ± 0.41** | **83.34 ± 0.77** | **100.00 ± 0.00** | **97.80 ± 0.07** | **90.30 ± 0.71** | **88.58 ± 0.12** |
| AllSetTrans.+EHGNN-F | **85.73 ± 0.36** | 76.54 ± 0.75 | 79.41 ± 0.44 | 99.94 ± 0.14 | 97.43 ± 0.11 | 88.91 ± 0.43 | 88.49 ± 0.16 |
| CE-GAT | 70.84 ± 4.40 | **74.39 ± 0.67** | 73.26 ± 0.54 | 96.78 ± 2.69 | 90.65 ± 0.15 | **78.33 ± 1.13** | **84.76 ± 0.16** |
| CE-GAT+EHGNN-F | **74.67 ± 1.76** | 73.65 ± 0.70 | **73.83 ± 1.15** | **99.57 ± 0.28** | **92.27 ± 0.29** | 77.46 ± 1.06 | 84.45 ± 0.69 |
| CE-GCN | 57.27 ± 0.37 | 52.51 ± 0.19 | 50.66 ± 0.40 | 51.98 ± 0.39 | 43.21 ± 0.37 | 35.20 ± 0.43 | 61.01 ± 0.06 |
| CE-GCN+EHGNN-F | **62.76 ± 0.03** | **53.29 ± 0.12** | **52.44 ± 0.49** | **52.63 ± 0.31** | **44.58 ± 0.62** | **35.55 ± 0.45** | **61.08 ± 0.05** |

training time substantially, with Random being the fastest. On DBLP-CA, the EHGNN variants take slightly longer than Full HGNN. This is because our methods introduce mask-learning steps (scoring, sampling, straight-through estimation) that add a fixed computational overhead. Table 4 shows that, on small datasets (ModelNet40, NTU2012, Cora, Citeseer), message passing itself is inexpensive, so this additional overhead dominates, leading to a slightly higher runtime than Full. *On larger datasets, such as Walmart and Yelp, EHGNN achieves training times comparable to or better than Full, as incidence-level sparsification reduces the dominant message-passing cost.*

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

| EHGNN-F | 77.94 $\pm$ 0.11 | 95.24 $\pm$ 0.07 | **96.35 $\pm$ 0.41** | **87.40 $\pm$ 0.18** | 76.68 $\pm$ 0.14 | **70.94 $\pm$ 0.37** | **80.77 $\pm$ 0.46** | 88.37 $\pm$ 0.07 |

| Model | Cora | House | Pokec | Pubmed | Twitch | Walmart | Yelp |
|---|---|---|---|---|---|---|---|
| EHGNN-F w/o Norm. | **75.92 $\pm$ 0.43** | 87.80 $\pm$ 1.19 | 58.76 $\pm$ 0.03 | 85.57 $\pm$ 0.07 | 50.71 $\pm$ 0.07 | 95.16 $\pm$ 0.02 | **30.92 $\pm$ 0.14** |
| EHGNN-F | 75.42 $\pm$ 0.57 | **87.93 $\pm$ 0.22** | **58.82 $\pm$ 0.09** | **85.59 $\pm$ 0.03** | **50.72 $\pm$ 0.11** | **95.17 $\pm$ 0.02** | 30.57 $\pm$ 0.46 |

# D    ABLATION STUDIES

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

**Encoder.**  Given the hypergraph $\mathcal{H} = (X, H)$, an HGNN (Feng et al., 2019) encoder $f_\theta$ produces node embeddings:

$$Z = f_\theta(X, H) \in \mathbb{R}^{n \times d}.$$

The encoder may also include a sparsification mask (EHGNN-F or EHGNN-C), in which case the effective incidence is:

$$\tilde{H} = H \odot M, \quad M \in \{0, 1\}^{n \times m},$$

where $M$ is the learned binary mask as discussed in section 4.

**Decoder.**  The decoder reconstructs the incidence matrix $H$ from $Z$. Each hyperedge $e \in E$ is represented by aggregating its node embeddings:

$$h_e = \frac{1}{|e|} \sum_{v \in e} Z_v, \quad h_e \in \mathbb{R}^d.$$

Table 10: Node clustering Accuracy ($\pm$ std) across datasets. Bold denotes the best non-*Full* result per dataset. Avg. Rank computed over datasets where all non-*Full* algorithms report accuracies (10 datasets: actor, citeseer, cora, house-committees, ModelNet40, Mushroom, NTU2012, pokec, pubmed, twitch).

| Algorithms | actor | citeseer | cora | house-committees | ModelNet40 | Mushroom | NTU2012 | pokec | pubmed | twitch | Avg. Rank |
|---|---|---|---|---|---|---|---|---|---|---|---|
| Full | $0.41 \pm 0.02$ | $0.41 \pm 0.02$ | $0.41 \pm 0.03$ | $0.67 \pm 0.00$ | $0.93 \pm 0.00$ | $0.84 \pm 0.05$ | $0.69 \pm 0.01$ | $0.51 \pm 0.01$ | $0.52 \pm 0.08$ | $0.51 \pm 0.00$ | |
| EHGNN-F | $0.41 \pm 0.03$ | $0.40 \pm 0.05$ | $\mathbf{0.42 \pm 0.04}$ | $\mathbf{0.61 \pm 0.03}$ | $\mathbf{0.92 \pm 0.02}$ | $0.69 \pm 0.07$ | $\mathbf{0.67 \pm 0.02}$ | $\mathbf{0.51 \pm 0.00}$ | $0.53 \pm 0.07$ | $0.51 \pm 0.00$ | 2.00 |
| EHGNN-C | $\mathbf{0.43 \pm 0.03}$ | $0.41 \pm 0.04$ | $0.41 \pm 0.02$ | $0.58 \pm 0.05$ | $0.90 \pm 0.02$ | $\mathbf{0.75 \pm 0.10}$ | $0.67 \pm 0.02$ | $0.51 \pm 0.00$ | $0.52 \pm 0.08$ | $0.51 \pm 0.00$ | 2.60 |
| EHGNN-F (cond.) | $0.41 \pm 0.03$ | $0.39 \pm 0.04$ | $0.40 \pm 0.04$ | $0.59 \pm 0.02$ | $0.91 \pm 0.01$ | $0.70 \pm 0.08$ | $0.66 \pm 0.01$ | $0.51 \pm 0.01$ | $\mathbf{0.53 \pm 0.07}$ | $\mathbf{0.51 \pm 0.00}$ | 2.90 |
| EHGNN-C (cond.) | $0.42 \pm 0.03$ | $\mathbf{0.41 \pm 0.04}$ | $0.41 \pm 0.02$ | $0.61 \pm 0.05$ | $0.91 \pm 0.01$ | $0.75 \pm 0.10$ | $0.65 \pm 0.01$ | $0.51 \pm 0.00$ | $0.53 \pm 0.07$ | $0.51 \pm 0.00$ | 2.50 |

Table 11: Node clustering NMI ($\pm$ std) across datasets. Bold denotes the best non-*Full* result per dataset. Avg. Rank computed over datasets where all non-*Full* algorithms report accuracies (10 datasets: actor, citeseer, cora, house-committees, ModelNet40, Mushroom, NTU2012, pokec, pubmed, twitch).

| Algorithms | actor | citeseer | cora | house-committees | ModelNet40 | Mushroom | NTU2012 | pokec | pubmed | twitch | Avg. Rank |
|---|---|---|---|---|---|---|---|---|---|---|---|
| Full | $0.00 \pm 0.00$ | $0.20 \pm 0.03$ | $0.24 \pm 0.02$ | $0.22 \pm 0.00$ | $0.92 \pm 0.00$ | $0.41 \pm 0.11$ | $0.82 \pm 0.00$ | $0.00 \pm 0.00$ | $0.12 \pm 0.06$ | $0.00 \pm 0.00$ | |
| EHGNN-F | $0.00 \pm 0.00$ | $0.18 \pm 0.04$ | $0.22 \pm 0.03$ | $0.14 \pm 0.04$ | $0.92 \pm 0.00$ | $0.14 \pm 0.07$ | $0.80 \pm 0.00$ | $0.00 \pm 0.00$ | $0.13 \pm 0.06$ | $0.00 \pm 0.00$ | 2.30 |
| EHGNN-C | $0.00 \pm 0.00$ | $0.18 \pm 0.03$ | $0.23 \pm 0.03$ | $0.10 \pm 0.07$ | $0.91 \pm 0.00$ | $0.24 \pm 0.12$ | $0.80 \pm 0.00$ | $0.00 \pm 0.00$ | $0.12 \pm 0.07$ | $0.00 \pm 0.00$ | 2.50 |
| EHGNN-F (cond.) | $0.00 \pm 0.00$ | $0.17 \pm 0.04$ | $0.22 \pm 0.04$ | $0.12 \pm 0.03$ | $0.92 \pm 0.00$ | $0.15 \pm 0.08$ | $0.80 \pm 0.01$ | $0.00 \pm 0.00$ | $0.14 \pm 0.06$ | $0.00 \pm 0.00$ | 2.90 |
| EHGNN-C (cond.) | $0.00 \pm 0.00$ | $0.19 \pm 0.03$ | $0.23 \pm 0.03$ | $0.14 \pm 0.08$ | $0.91 \pm 0.00$ | $0.24 \pm 0.12$ | $0.80 \pm 0.00$ | $0.00 \pm 0.00$ | $0.13 \pm 0.06$ | $0.00 \pm 0.00$ | 2.30 |

Table 12: Node clustering ARI ($\pm$ std) across datasets. Bold denotes the best non-*Full* result per dataset. Avg. Rank computed over datasets where all non-*Full* algorithms report accuracies (10 datasets: actor, citeseer, cora, house-committees, ModelNet40, Mushroom, NTU2012, pokec, pubmed, twitch).

| Algorithms | actor | citeseer | cora | house-committees | ModelNet40 | Mushroom | NTU2012 | pokec | pubmed | twitch | Avg. Rank |
|---|---|---|---|---|---|---|---|---|---|---|---|
| Full | $0.01 \pm 0.00$ | $0.16 \pm 0.02$ | $0.18 \pm 0.03$ | $0.12 \pm 0.00$ | $0.90 \pm 0.00$ | $0.48 \pm 0.13$ | $0.65 \pm 0.01$ | $0.00 \pm 0.00$ | $0.11 \pm 0.06$ | $0.00 \pm 0.00$ | |
| EHGNN-F | $0.01 \pm 0.00$ | $0.13 \pm 0.03$ | $0.16 \pm 0.03$ | $0.05 \pm 0.02$ | $0.91 \pm 0.01$ | $0.16 \pm 0.10$ | $0.62 \pm 0.03$ | $0.00 \pm 0.00$ | $0.13 \pm 0.07$ | $0.00 \pm 0.00$ | 2.10 |
| EHGNN-C | $0.01 \pm 0.00$ | $0.14 \pm 0.03$ | $0.16 \pm 0.04$ | $0.04 \pm 0.05$ | $0.88 \pm 0.02$ | $0.29 \pm 0.16$ | $0.63 \pm 0.04$ | $0.00 \pm 0.00$ | $0.11 \pm 0.08$ | $0.00 \pm 0.00$ | 2.60 |
| EHGNN-F (cond.) | $0.01 \pm 0.00$ | $0.12 \pm 0.03$ | $0.15 \pm 0.03$ | $0.04 \pm 0.02$ | $0.90 \pm 0.01$ | $0.18 \pm 0.12$ | $0.61 \pm 0.02$ | $0.00 \pm 0.00$ | $0.13 \pm 0.07$ | $0.00 \pm 0.00$ | 2.80 |
| EHGNN-C (cond.) | $0.01 \pm 0.00$ | $0.14 \pm 0.03$ | $0.16 \pm 0.03$ | $0.06 \pm 0.05$ | $0.90 \pm 0.01$ | $0.29 \pm 0.16$ | $0.60 \pm 0.02$ | $0.00 \pm 0.00$ | $0.12 \pm 0.07$ | $0.00 \pm 0.00$ | 2.50 |

The probability that node $v$ belongs to hyperedge $e$ is modeled as:

$$\hat{\boldsymbol{H}}_{v,e} = \sigma\left(\langle \boldsymbol{W}_n \boldsymbol{Z}_v, \boldsymbol{W}_e \boldsymbol{h}_e \rangle\right),$$

where $W_n, W_e \in \mathbb{R}^{d \times d}$ are learnable projections of the decoder, $\langle \cdot, \cdot \rangle$ denotes the dot product, and $\sigma$ is the sigmoid activation function.

**Reconstruction Loss.** We sample the same number of positive incidences $(v, e)$ where $H_{v,e} = 1$, and negatives $(v, e)$ where $H_{v,e} = 0$. The reconstruction loss is binary cross-entropy loss:

$$\mathcal{L}_{\mathrm{recon}} = - \sum_{(v,e) \in \Omega^+} \log \hat{H}_{v,e} - \sum_{(v,e) \in \Omega^-} \log(1 - \hat{H}_{v,e}),$$

where $\Omega^+ = \{(v, e) : H_{v,e} = 1\}$ and $\Omega^-$ is a set of sampled negatives.

The encoder parameters $\theta$, and decoder projections $(W_n, W_e)$ are optimized jointly to minimize $\mathcal{L}_{\mathrm{recon}}$.

**Observations.** We observe that all algorithms (including Full) perform poorly on heterophilic hypergraph benchmarks (Actor, Pokec, twitch). We believe the reason is the following: on heterophilic hypergraphs node neighbors often have different labels which renders laplacian-style averaging over incident nodes under homophily assumption ineffective. As there are currently no studies on the node-clustering performance of HGNNs on heterophilic hypergraphs, addressing this issue not only under full training but also sparsified training setting would be an interesting future work.