# OpenReview forum: "EdgeMask-HGNN: Learning to Sparsify Hypergraphs for Scalable Node Classification in Hypergraph Neural Networks"
_ICLR.cc/2026/Conference — Submitted to ICLR 2026_

### Official Review · Reviewer_g7jz · 2025-10-26

**Soundness:** 2
**Presentation:** 2
**Contribution:** 2
**Rating:** 4
**Confidence:** 4

**Summary:**

The authors propose a hypergraph sparsification framework, which removes certain hyperedge membership to achieve scalability as well as downstream task performance improvement.

To this end, the authors propose EHGNN, which learns a masking matrix for a given incidence matrix. Specifically, the matrix is optimized from the targeted downstream task loss, such as node classification.

The authors verify the performance of EHGNN in several benchmark hypergraph datasets.

**Strengths:**

S1. Given that real-world group interactions occur on a scale, the research topic is very important.

S2. The authors demonstrate the effectiveness of the proposed method under diverse backbone HNNs.

**Weaknesses:**

**W1. [Goal of sparsification]** The title and presentation of the work focus on **scalability**. However, it seems that the proposed method often requires more training time and GPU memory consumption (Tables 3 and 4). Given these results, I think the scalability of the proposed method is not experimentally supported. Could the authors further clarify these results?

**W2. [Training]** It seems that authors are using sampling strategies in their methods. However, the sampling process often cuts the gradient, making the training infeasible. It seems that authors incorporate graph sparsification techniques (Lines 268 - 270), such details should be self-contained in the manuscript. Could the authors elaborate on this?

**W3. [Datasets]** I think the used datasets are not that large-scale, as which are fewer than 100K nodes. Given that graph sparsification works are evaluated in a million-scale graphs, the authors are expected to evaluate their methods in larger hypergraphs. I think the authors can refer to the work: (Datasets, tasks, and training methods for large-scale hypergraph learning, DAMI 2023).

**Questions:**

See Weakness.

---

> ### Author Response · Authors · 2025-11-22
>
> Thank you for recognizing the importance of this work. Here is our response to your comments.
>
> > W1.
>
> We thank the reviewer for pointing out the discrepancy in our original memory and runtime results. After re-examining our implementation, we identified an issue in the way GPU memory usage was measured: the earlier memory numbers were inflated because of how and when memory usage was sampled. We have corrected this by synchronizing CUDA operations and using PyTorch’s peak-memory utilities, which provide consistent, accurate reporting of training time GPU memory usage.
>
> With the corrected measurement, the updated Table 3 now shows that our proposed sparsifiers-- **EHGNN-C, EHGNN-F, and the low-rank variant EHGNN-F(cond, LR), consistently reduce peak GPU memory compared to full training across all benchmarks**. These variants achieve 15–40% lower memory usage depending on dataset complexity. Furthermore, EHGNN-C(cond) also yields reductions on all datasets except Citeseer.
>
> We also note that our sparsification improves scalability not only by reducing memory but also by enabling training on datasets where full models fail. For instance,  EHGNN-F enables ED-HNN [1] (a newly added backbone) training on the Yelp hypergraph, where ED-HNN runs out of memory (see Table D below). In addition, we are finalizing a more memory-friendly version of EHGNN-C(cond), analogous to the low-rank approach used in EHGNN-F(cond, LR), and will include its results as soon as they are ready.
>
>
> ## Table D —   Peak GPU Memory Comparison (GB): ED-HNN vs. ED-HNN+EHGNN-F
> *(sparsity = 50%)*
>
> | Models      | Actor   | Cora    | Cora-CA | House   | ModelNet40 | NTU2012     | PubMed | yelp |
> |---------------------|---------|---------|---------------------------|---------|------------|---------|---------|---------|
> | **EDHNN (MB)**         | 640.44  | 153.65  | 149.09                    | 158.36  | 860.54     | 160.53  | 771.90  | OOM |
> | **EDHNN + EHGNN-F (MB)**    | 553.01  | 127.68  | 126.66                    | 99.30   | 543.33     | 107.73  | 604.35  | 12907 |
> | **Reduction (MB)**    | **87.43** | **25.97** | **22.43**                | **59.06** | **317.21** | **52.80** | **167.55** | - |
> | **Reduction (%)**     | **13.65%** | **16.90%** | **15.04%**              | **37.28%** | **36.85%** | **32.90%** | **21.71%** | - |
>
> ---
> > W2.
>
> Thank you for this question. EdgeMask-HGNN does not rely on nondifferentiable stochastic sampling during training. Instead, the sparsification module uses a continuous relaxation of the mask, which guarantees that gradients propagate end-to-end through the sparsifier and into the HGNN backbone.
>
> For both incidence-level (EHGNN-F) and edge-level (EHGNN-C) masking, the learnable scores $s_{v,e}$ (EHGNN-F) or $s_e$ (EHGNN-C) are transformed into soft selection probabilities $p = \sigma(s)$ (up to normalization). Although the forward pass uses a discrete Top-$k$ mask for sparsity, we employ the Straight-through estimator (STE) (Bengio et al., 2013) to maintain differentiability (see discussions in lines 211-214 of our revised paper). Let us assume $\hat{m}$ be the sampled hard mask and $m$ be the continuous surrogate:
>
> $\hat{m} \sim Top-k(Categorical(p)), \quad  m = stopgrad (\hat{m}) + p - stopgrad(p).$
>
> During backpropagation, the gradients satisfy:
>
> $\frac{\partial \mathcal{L}}{\partial s} = \frac{\partial \mathcal{L}}{\partial m} \cdot \frac{\partial m}{\partial p} \cdot \frac{\partial p}{\partial s}$,
>
> which ensures that the sparsifier learns jointly with the downstream HGNN. Thus, no gradient paths are cut, and the training remains fully differentiable.
>
> **To make the manuscript self-contained, we have revised our paper by adding a “Discussion on differentiability” in section 4.1**

---

> ### Author Response · Authors · 2025-11-22
>
> > W3.
>
> Following the reviewer's suggestion, we have now included results on the much larger Trivago hypergraph, which contains 172,738 nodes and 233,202 hyperedges (see Tables 2-3 in the revision). **This experiment demonstrates that our sparsification framework remains both effective and scalable on significantly larger hypergraphs.**
>
> As shown in Table E below, EdgeMask-HGNN variants achieve strong accuracy, substantially outperforming unsupervised sparsifiers (Random, Degdist) and staying competitive with the full HGNN, while also achieving **up to 27% peak GPU memory reduction** and **lower runtime per epoch.** Note that, while the newly added SOTA method HSL [2] achieves the highest accuracy (68.69 ± 1.06), it does so by preserving the full hypergraph during message passing, which results in significantly higher memory and runtime costs (e.g., 2.64 seconds/epoch, compared with 0.15–0.20 seconds/epoch for EHGNN variants).
>
> ## Table E —   Accuracy and Peak GPU Memory Comparison (GB) on Trivago
> *(sparsity = 50%)*
>
> | Metric | Full       |  Random      |  Degdist       |  Spectral         | HSL    | EHGNN-F    | EHGNN-C    | EHGNN-F(cond) | EHGNN-C(cond) | EHGNN-F(cond,LR) |
> |:--------|:--------------------|:--------------------|:--------------------|:--------------------|:--------------------|:--------------------|:--------------------|:--------------------|:--------------------|:--------------------|
> | **Accuracy** | 61.68 ± 0.28 | 26.78 ± 0.05 | 26.78 ± 0.05 | OOM | **68.69 ± 1.06** | 50.38 ± 1.40 | 51.41 ± 0.72 | 46.22 ± 1.21 | 50.14 ± 0.72 | 49.87 ± 0.95 |
> | **Memory (GB)** | 6.2  | **4.44 (28% ↓)** | **4.44 (28% ↓)** | OOM | 5.53 (11% ↓) | 4.55 (27% ↓) | 4.84 (22% ↓) | 4.52 (27% ↓) | 4.84 (22% ↓) | 4.53 (27% ↓) |
> | **Avg. runtime/epoch (sec)** | 0.27 | **0.11** | **0.11** | OOM | 2.64 | 0.15 | 0.16 | 0.2 | 0.18 | 0.15 |
> | % sparsified | 0% | 50% | 50% | OOM | 0% | 50% | 50% | 50% | 50% | 50% |
>
> *Please note that EHGNN-F(cond,LR) is a lightweight low-rank parameterization of EHGNN-F(cond) that preserves feature-conditioning without the associated memory overhead. We have discussed it at length in Appendix C.*
>
> `References`
>
> [1] Equivariant hypergraph diffusion neural operators. ICLR'23.
>
>
> [2] Cai, Derun, et al. "Hypergraph Structure Learning for Hypergraph Neural Networks." IJCAI. 2022.
>
> ---
> **Please note that we have colored the changes in the revision in blue**

---

> ### Comment · Reviewer_g7jz · 2025-11-27
>
> Thank you for your response.
>
> My major concerns have been addressed, and in light of it, I have raised my score from 4 to 6.
>
> Please incorporate the revision into the revised manuscript.

---

> ### Author Response · Authors · 2025-11-28
> **Remaining experiments as promised earlier**
>
> **Thank you very much for your thoughtful feedback and the time you have dedicated to helping us improve our paper. We will incorporate the suggested revisions.**
>
> In the meantime, please let us know if there are any remaining concerns or points you would like us to address during the discussion phase. We would be happy to clarify or provide additional analysis as needed.
>
> ---
> **Additional experiments we promised:**
> As mentioned earlier in our response to W3, we have been working on a low-rank reparameterization of EHGNN-C (cond) to improve its memory usage. Below, we present the revised Table E to include the results we obtained with this variant.
>
> *Please note the EHGNN-C(cond.) (as reported in Tables 2–4) deploy chunked scoring (chunk size = 32k), an optimization we attempted during rebuttal, to avoid OOM.* This optimization, though effective memory-wise, significantly increases end-to-end training time; for instance, it takes 2230.27 seconds on Yelp compared to 277 seconds by EHGNN-C(cond, LR). Thus, EGHNN-C (cond, LR) is a more balanced alternative to EHGNN-C(cond) - both memory- and runtime-wise.
>
> ## Revised Table E —   Accuracy and Peak GPU Memory Comparison (GB) on Trivago
>
> | Metric | Full       |  Random      |  Degdist       |  Spectral         | HSL    | EHGNN-F    | EHGNN-C    | EHGNN-F(cond) | EHGNN-C(cond) | EHGNN-F(cond,LR) | EHGNN-C(cond,LR) |
> |:--------|:--------------------|:--------------------|:--------------------|:--------------------|:--------------------|:--------------------|:--------------------|:--------------------|:--------------------|:--------------------|:--------------------|
> | **Accuracy** | 61.68 ± 0.28 | 26.78 ± 0.05 | 26.78 ± 0.05 | OOM | **68.69 ± 1.06** | 50.38 ± 1.40 | 51.41 ± 0.72 | 46.22 ± 1.21 | 50.14 ± 0.72 | 49.87 ± 0.95 | 48.18 ± 0.49 |
> | **Memory (GB)** | 6.2  | **4.44 (28% ↓)** | **4.44 (28% ↓)** | OOM | 5.53 (11% ↓) | 4.55 (27% ↓) | 4.84 (22% ↓) | 4.52 (27% ↓) | 4.84 (22% ↓) | 4.53 (27% ↓) | 4.84 (22% ↓) |
> | **Avg. runtime/epoch (sec)** | 0.27 | **0.11** | **0.11** | OOM | 2.64 | 0.15 | 0.16 | 0.2 | 0.18 | 0.15 | 0.16 |
> | **% sparsified** | 0% | 50% | 50% | OOM | 0% | 50% | 50% | 50% | 50% | 50% | 50% |
> ---
> *Please note that we have updated tables 2-4 in the revision with results from this newly added variant.*

---

### Official Review · Reviewer_voUi · 2025-10-27

**Soundness:** 2
**Presentation:** 2
**Contribution:** 2
**Rating:** 2
**Confidence:** 4

**Summary:**

This paper introduces EdgeMask-HGNN, a learnable task-aware sparsification framework for Hypergraph Neural Networks (HGNNs) to address scalability challenges. The key contributions include: (1) two masking strategies - fine-grained incidence-level (EHGNN-F) and coarse-grained hyperedge-level (EHGNN-C) masking, both trained end-to-end using downstream task supervision; (2) theoretical analysis proving stability under stochastic masking and O(1/ε) convergence of retention probabilities; and (3) extensive experiments on 15 datasets demonstrating that the method maintains or reduces memory usage while preserving (or sometimes improving) predictive performance compared to full HGNN training.

**Strengths:**

**Originality:**
- First work to propose learnable, task-aware sparsification for HGNNs, addressing a gap in the literature where only unsupervised methods existed
- The dual-granularity framework (incidence-level vs. edge-level) is a sensible design choice that provides flexibility
- Feature-conditioned variants using permutation-invariant pooling show creativity in adapting to semi-supervised settings

**Quality:**
- Extensive experimental evaluation across 15 datasets with multiple metrics (accuracy, memory, runtime)
- Proper ablation studies investigating design choices (Table 8, Figure 4)
- Theoretical analysis attempts to provide formal guarantees (Theorems 5.1, 5.2)
- Code release promised for reproducibility
- Model-agnostic framework demonstrated on multiple HGNN architectures (Table 5)

**Clarity:**
- Well-structured paper with clear motivation illustrated in Figure 1
- Comprehensive appendix with detailed proofs and additional experiments
- Generally clear writing, though notation becomes dense in places

**Significance:**
- Addresses the important problem of HGNN scalability
- Demonstrates that selective sparsification can sometimes improve accuracy (8/15 datasets), suggesting noise reduction benefits
- However, significance is substantially diminished by inconsistent results and failure on the exact scenarios (very large, dense hypergraphs) where scalability matters most

**Weaknesses:**

### 1. **Fundamental Contradiction in Core Contribution**

The paper's primary motivation is memory reduction for scalability on large-scale hypergraphs. However, **Table 3 reveals that on Yelp** (the largest and densest dataset at 679,302 edges), several variants consume **MORE memory than full training**:
- EHGNN-F (cond.): 138.1 GB vs. 111.2 GB (Full)
- EHGNN-C (cond.): 139.5 GB vs. 111.2 GB (Full)

This directly contradicts the paper's main claim. The explanation provided (lines 401-405) about "pooling and scorer MLPs introduce extra activations" is insufficient. **This needs rigorous analysis:**
- Provide exact memory breakdown: parameters vs. activations vs. optimizer states
- Why does the theoretical space complexity O(m + kd) not hold in practice?
- Under what density/size thresholds does memory actually increase?
- Is this a fundamental limitation or an implementation issue?

---

### 2. **Inconsistent Performance Without Predictive Framework**

Results show high variance across datasets with no clear pattern:
- **Improvements:** ModelNet40 (+0.37%), DBLP-CA (+0.19%), several others
- **Degradations:** 20news (-0.30%), NTU2012 (-0.31%), Actor (+12-14% but highly variable across variants)
- **Equivalent:** Multiple datasets within error bars of full training

For a methods paper at ICLR, this inconsistency is problematic. **The paper lacks:**
- A principled framework predicting when sparsification helps vs. hurts
- Analysis of hypergraph properties (density, edge size distribution, homophily) correlated with success
- Clear decision criteria for practitioners choosing between variants

---

### 3. **Weak Theoretical Contributions **

**Theorem 5.1 (Stability):**
- Assumes HGNN is L-Lipschitz w.r.t. Frobenius norm but provides **no validation**
- Are standard HGNNs (HGNN, HyperGCN, etc.) actually Lipschitz? What is L?
- The bound shows variance decreases as p→0 or p→1, which is obvious—doesn't explain empirical behavior

**Theorem 5.2 (Convergence):**
- Assumes gradient signs remain fixed throughout training—**highly unrealistic**
- O(1/ε) rate is standard for many optimization problems, not a novel contribution
- No empirical validation showing this rate actually occurs (Figure 6 only shows final distributions)

**Missing theory:**
- When does task-aware sparsification improve over unsupervised methods?
- What properties of the loss landscape explain why removing edges helps in some cases?
- Connection between sparsification and generalization bounds

---

### 4. **Insufficient Baseline Comparisons **

**Critical missing baselines:**
- **Spectral methods OOM** on exactly the datasets where comparison matters (Walmart, Yelp, DBLP-CA). This makes claims about outperforming spectral methods unsubstantiated where it counts
- **No other learnable/task-aware methods:** The paper positions itself against only unsupervised baselines. Are there no graph sparsification methods (e.g., L0-based, DropEdge variants) that could be adapted?
- **No graph-to-hypergraph adaptations:** What if you apply graph sparsification methods after clique expansion?

---

### 5. **Heterophilic Graphs: Acknowledged but Unresolved **

- Appendix G shows **poor performance on heterophilic benchmarks** (Actor, Pokec, Twitch)
- Section E.1 shows only marginal differences between EHGNN-F and EHGNN-C on synthetic heterophilic data
- The explanation (line 1023-1026) attributes this to Laplacian averaging, but this affects **full training too**—not specific to sparsification

**This limits applicability** given growing interest in heterophilic graph learning.

---

### 6. **Parameter Overhead Not Properly Addressed **

Table 9 shows EHGNN-F has **massive parameter overhead:**
- Yelp: 5,482,067 vs. 958,473 (Full) = **5.7× more parameters**
- Walmart: 472,417 vs. 11,787 (Full) = **40× more parameters**

Yet the paper focuses on activation memory. Questions:
- How does this affect training time and convergence?
- What about optimizer states (Adam stores 2× parameters)?
- Does this create overfitting risk, especially given inconsistent results?
---

### 7. **Missing Critical Experimental Details **

- **No convergence analysis:** How many epochs? What stopping criteria?
- **Sparsity budget selection:** How is k or κ chosen? Different per dataset? Sensitivity analysis missing beyond one dataset (Figure 3a)
- **Statistical testing:** With overlapping error bars (e.g., Cora: 77.52±0.28 vs. 78.35±0.34), are differences significant?
- **Hyperparameter selection:** How sensitive are results to MLP hidden dimensions (Table 7)?

---

### 8. **Runtime Claims Need Clarification **

- Table 4 shows EHGNN methods often **slower** than Full on small datasets
- The explanation (mask-learning overhead) contradicts efficiency claims
- Figure 3c shows gains only on largest datasets, but these are the same ones with memory issues

---

**Questions:**

1. **Memory paradox on Yelp:** Can you provide a detailed memory breakdown (parameters, activations, gradients, optimizer states) explaining why memory increases on dense hypergraphs? Is this fundamental to your approach or fixable? At what hypergraph density/size does this crossover occur?

2. **When does your method work?** Can you provide a quantitative analysis correlating dataset properties (from Table 1: density = |E|×avg_edge_size / |V|, edge size distribution, homophily, etc.) with performance gains/losses? This is essential for practitioners.

3. **Spectral baseline on large graphs:** Can you implement an approximate spectral method (e.g., using random projections or incomplete Cholesky) to enable comparison on Walmart and Yelp? Without this, your claims about outperforming spectral methods are incomplete.

4. **Statistical significance:** Are the reported improvements statistically significant? For example, on Cora (77.52±0.28 vs 78.35±0.34), the confidence intervals overlap substantially.

5. **Theoretical assumptions validation:**
   - What is the Lipschitz constant L for standard HGNNs empirically?
   - How often do gradient signs flip during training? Can you show this empirically?
   - If assumptions are violated, do the theorems still provide useful insights?

6. **Feature-conditioned variants:** Why don't EHGNN-F(cond.) and EHGNN-C(cond.) consistently outperform non-conditioned versions despite added complexity? When should they be preferred?

7. **Sparsity budget selection:** How should practitioners choose k or κ? Is there a principled approach (e.g., based on validation performance, theoretical bounds, computational budget)?

8. **Heterophilic failure mode:** You mention this is due to Laplacian averaging (line 1023), but Full training uses the same mechanism yet performs better. What specifically about sparsification makes heterophilic graphs harder?

9. **Comparison with graph methods:** Have you tried adapting graph sparsification methods (e.g., via clique expansion then applying graph sparsifiers)? Why/why not?

10. **Overfitting analysis:** With 40× more parameters on some datasets (Table 9), do you observe overfitting? How does train vs. validation vs. test performance compare?

---

> ### Author Response · Authors · 2025-11-23
> **W1**
>
> Thank you for taking the time to read and providing us with detailed comments on our paper.
>
> ## W1 (Fundamental Contradiction in Core Contribution), Q1
>
> Thank you for highlighting this important point. Indeed, on the densest dataset (Yelp), two feature-conditioned variants, EHGNN-F(cond.) and EHGNN-C(cond.), show higher peak GPU memory usage than Full training, with EHGNN-F(cond.) running out of memory. **Upon re-examination, we found that this behavior does not contradict our stated space complexity**. Instead, it arises from a specific interaction between (i) the extreme density of Yelp’s incidence structure and (ii) additional activations created by the feature-conditioned scorer networks. This combination is unique to Yelp and does not occur on any other dataset, including those with similar node counts (e.g., Trivago).
>
> **A detailed memory breakdown confirms that the increase comes exclusively from activation memory, not parameter memory.** The results in Table F below show that static model memory (parameters + buffers) is only 3-6 MB, while the peak activation memory ranges from 11-20 GB across Full, EHGNN-F, and EHGNN-C. Yelp is extremely dense (|E|=679k, feature dimension F=1862), and even feature-agnostic sparsifiers (EHGNN-F, EHGNN-C) must materialize pooled embeddings for all hyperedges, incurring $\mathcal{O}(md)$ activation cost.
>
> ## Table F —  Memory breakdown
>
> | Model | peak Activation  memory     |  static model memory (params + Buffer)      |
> |:--------|:--------------------|:--------------------|
> | Full | 11841 MB (11.6 GB) | 3.88 MB |
> | EHGNN-F | 15813 MB (15.4 GB)| 6.25 MB |
> | EHGNN-C | 20496 MB (20 GB) | 3.65 MB|
>
> As EHGNN-F(cond) and EHGNN-C(cond) require additional per-edge scorer MLP activations, their activation footprint is even larger, which explains the observed OOM. Please note the reported results for EHGNN-C(cond.) in Tables 2–4 use chunked scoring (chunk size = 32k), an optimization we attempted during rebuttal, to avoid OOM. This significantly increases runtime (2230.27 seconds on Yelp), but without chunking, EHGNN-C(cond) also runs OOM. Our low-rank scorer variant EHGNN-F(cond,LR) avoids this issue and provides strong memory and runtime reductions; see Appendix C for discussion on this variant.
>
> In summary, the only cases where memory exceeds Full training are feature-conditioned variants on hypergraphs with large $m=|E|$, large feature dimension $d$, and dense connections, where the scorer MLP introduces an $\mathcal{O}(md)$ activation cost. This is not due to the sparsification mechanism itself but to an implementation-level overhead that disappears in the feature-agnostic and low-rank variants, which consistently reduce memory on all datasets, including Yelp.
>
> >*At what hypergraph density/size does this crossover occur?:*
>
> On the 16 benchmarks we studied, we do not observe a sharp density threshold that is universal. Empirically, all feature-agnostic variants and the low-rank conditioned variant reduce memory on every dataset up to Trivago ($t \approx 726K, m \approx 233K$ hyperedges). The only crossover we observe is on Yelp ($t \approx 4.5M , m \approx 679K$), where the full feature-conditioned variants exceed the Full baseline due to additional scorer-MLP activations. This suggests that, in practice, the problematic regime arises only at hypergraphs with millions of incidences; below that scale, all our sparsifiers consistently reduce memory relative to Full.
>
> ---
>
> **Reason for updating the peak memory values in Table 3 during rebuttal:** After re-examining our implementation, we identified an issue in the way GPU memory usage was measured: the earlier memory numbers were inflated because of how and when memory usage was sampled. We have corrected this by synchronizing CUDA operations and using PyTorch’s peak-memory utilities, which provide consistent, accurate reporting of training time GPU memory usage.

---

> ### Author Response · Authors · 2025-11-23
> **W2**
>
> ## W2 (Inconsistent performance), Q2(When does the method works)
>
> The previously observed variance (from our earlier draft) is no longer unexplained. In the revised manuscript, accuracy and memory behaviors follow clear structural patterns tied to hypergraph homophily, incidence scale, and edge-size distributions. The updated analyses (Section 6.1, Fig. 7, Table 2) provide a principled framework predicting when sparsification helps, when it maintains performance, and when it may underperform.
>
> *(A) When sparsification improves accuracy:*  (1) When hypergraph is heterophilic, in particular, when Edge/node homophily is roughly < 0.6. See Figure 7, Appendix C.1 in the revision.
>
> *(B) When sparsification maintains accuracy while reducing memory/runtime:* Large-scale but not extremely dense hypergraphs, e.g., DBLP-CA, Walmart. Yelp (exception is EHGNN-C (cond) for reasons explained earlier). See Table 2  in the revision.
>
> *(C) When sparsification may slightly hurt accuracy:* Highly homophilic (e.g., Cora, Citeseer, PubMed) and/or small-scale hypergraphs (e.g., 20news). See Table 2  in the revision.
>
> > Clear decision criteria for practitioners choosing between variants
>
> To support practical decision-making,  one can consider the following table as a guideline, where we recommend variants based on the benchmarks we studied.
>
> | Regime                    | Recommended variant                 | Expected performance  | Representative benchmark                                       |
> |---------------------------|-------------------------------------|-----------------------------------------------------------|----------|
> | **Small t** (< 20k)       | Full HGNN    | Sparsification may remove important learning signal;  full model already efficient.  | Cora (t=4,786) |
> | **Medium t** (20k–200k)   | EHGNN-C or EHGNN-F   | Accuracy matches or exceeds Full; ; moderate memory gains. | ModelNet40 (t=61,555) |
> | **Large t** (200k–800k)   | EHGNN-F                             | Strong accuracy & memory gains; incidence/edge pruning reduces activation cost. | Walmart (t=460,630) |
> | **Extremely large t** (>800k)    | EHGNN-F or EHGNN-F (cond, LR) | Feature-conditioned variants may require low-rank scorer to curb activation memory. | Yelp (t=4,523,594) |
>
> > Analysis of hypergraph properties correlated with success
> > Can you provide a quantitative analysis correlating dataset properties
>
> We added a statistical analysis correlating accuracy deltas (accuracy(EHGNN-C)-accuracy(Full)) and (accuracy(EHGNN-F) - accuracy(Full)) separately with hypergraph properties (Table 9, Appendix F.3). The results show a consistent pattern:
> Accuracy deltas for both EHGNN-C and EHGNN-F exhibit strong negative correlations with edge and node homophily. For EHGNN-C, the Spearman correlation with edge homophily is -0.79, and node homophily is  -0.82. For EGHNN-F, the Spearman correlation with edge homophily is -0.66 and node homophily is -0.68.
> Scale-related statistics (|V|, |E|, t) show near-zero correlation with accuracy deltas.
> Density, sparsity, and average edge size show weak or inconsistent effects, confirming that homophily is the dominant predictor of accuracy changes.

---

> ### Author Response · Authors · 2025-11-23
> **W4, Q3, Q9**
>
> ## W4 (Insufficient Baseline Comparisons)
>
> > Spectral methods OOM. + Q3. Can you implement an approximate spectral method
>
> In addition to the exact effective-resistance-based sparsifier, which requires a dense pseudoinverse of the hypergraph Laplacian, we implemented an approximate spectral sparsifier using a Hutchinson-type random projection estimator for $\operatorname{diag}(L^{+})$. Specifically, we sample vectors $g$ from Rademacher distribution, solve $(L + \lambda I)z = g$ via Conjugate Gradient, and approximate $\operatorname{diag}(L^{+}) \approx \mathbb{E}[z \odot g]$. These approximate resistances are then aggregated per hyperedge as in the exact method. This follows the standard line of random-projection-based spectral sparsification and avoids forming the pseudoinverse explicitly.
>
> In practice, this approximation yields some loss in accuracy compared to full training while being significantly more scalable. Table 10 (Appendix F.4) shows that on DBLP-CA, it reduces memory consumption than exact sparsification, while enabling training on Yelp and Walmart.
>
> |  | DBLP-CA | Yelp | Walmart |
> |:---:|:---:|:---:|:---:|
> | Spectral (exact) | 84.83, 0.96 GB (10\% reduction) | OOM | OOM |
> | Spectral (approximate) | 81.95, 0.91 GB (15\% reduction) | 28.67, 16.68 GB (15\%  reduction) | 70.68, **1.71 GB** (39\% reduction) |
> | EHGNN-C | **88.11**, **0.89 GB** (17\% reduction) | **32.31**, **11.51 GB** (41\% reduction) | **95.17**, 1.93 GB (31\% reduction) |
>
> > No other learnable/task-aware methods:
>
> Learning on Hypergraphs is still in its infancy. Hence, there is a lack of sufficient works to compare with. Following Reviewer gFef ‘s suggestion, we compared EdgeMask-HGNN against Hypergraph Structure Learning (HSL) proposed by Cai et al. (IJCAI 2022).
>
> HSL, although not designed to be a hypergraph sparsifier per se, is an unsupervised structure-optimization framework that learns a new hypergraph prior to HGNN training. HSL does not have a provision for a user-specified sparsity budget, and it augments the input hypergraph with a residual hypergraph structure $\Delta H$ that may add new node-hyperedge connections that were non-existent in the input. We modified HSL by (1) adding an L2-regularizer with HSL’s loss function so that it respects the sparsity budget as much as possible and (2) setting its residual hypergraph structure $\Delta H = 0$ so that connections can only be removed, not added. Tables 2 and 3 in the revision show HSL’s performance.
>
> Below, we provide two tables comparing (1) accuracy, and (2) Memory reduction relative to HSL. These tables demonstrate that EHGNN variants are competitive or superior to HSL across diverse datasets, while offering significant memory reduction on large-scale Walmart, Yelp, and Trivago datasets.
> ## Table A — Accuracy Comparison of HSL vs. EHGNN Variants
> *(sparsity = 50%, Bold = highest accuracy per dataset)*
>
> | **Models** | 20News | ModelNet40 | Mushroom | NTU2012 | Actor | CiteSeer | Cora-CA | DBLP-CA | Cora | House | Pokec | PubMed | Twitch | Walmart | Yelp | Trivago |
> |------------|--------|-------------|----------|----------|--------|-----------|-----------|-----------|--------|--------|--------|---------|---------|----------|---------|----------|
> | **HSL** | **81.85 ± 0.10** | 86.36 ± 1.61 | **99.79 ± 0.03** | 80.45 ± 1.84 | 63.01 ± 0.24 | 62.20 ± 0.24 | 75.82 ± 1.26 | **90.11 ± 0.55** | 74.10 ± 1.62 | 75.85 ± 0.31 | 58.03 ± 0.67 | 78.88 ± 1.16 | **51.15 ± 0.41** | 63.59 ± 0.21 | 24.89 ± 0.03 | **68.69 ± 1.06** |
> | **EHGNN-F** | 77.99 ± 0.07 | 95.25 ± 0.06 | 96.39 ± 0.39 | **87.51 ± 0.22** | 77.00 ± 0.07 | 68.12 ± 0.52 | 79.53 ± 0.55 | 88.11 ± 0.07 | 74.33 ± 0.41 | 87.31 ± 1.58 | 58.78 ± 0.07 | **85.56 ± 0.04** | 50.74 ± 0.05 | 95.17 ± 0.02 | 32.31 ± 0.15 | 50.38 ± 1.40 |
> | **EHGNN-C** | 78.01 ± 0.34 | 93.74 ± 0.10 | 94.14 ± 0.60 | 84.65 ± 0.51 | 77.20 ± 0.07 | 67.46 ± 0.58 | **80.50 ± 0.15** | 89.30 ± 0.07 | 73.18 ± 0.36 | 84.09 ± 1.17 | 58.79 ± 0.04 | 85.42 ± 0.06 | 51.10 ± 0.04 | 95.68 ± 0.01 | 31.75  ± 0.29 | 51.41 ± 0.72 |
> | **EHGNN-F (cond.)** | 78.67 ± 0.06 | 95.59 ± 0.12 | 95.07 ± 0.69 | 87.32 ± 0.43 | **78.73 ± 0.05** | **68.36 ± 0.43** | 76.07 ± 0.66 | 86.27 ± 0.08 | 73.94 ± 0.39 | **100.00 ± 0.00** | **59.17 ± 0.09** | 85.53 ± 0.06 | 50.84 ± 0.06 | **98.37 ± 0.01** | OOM | 46.22 ± 1.21 |
> | **EHGNN-C (cond.)** | 74.63 ± 0.39 | 94.54 ± 0.15 | 97.52 ± 0.12 | 85.45 ± 0.29 | 77.57 ± 0.06 | 67.20 ± 0.57 | 75.51 ± 0.16 | 87.10 ± 0.14 | **76.22 ± 0.35** | 72.14 ± 0.73 | 59.02 ± 0.03 | 85.52 ± 0.05 | 50.67 ± 0.03 | 94.72 ± 0.04 | 29.21 ± 0.36 | 50.14 ± 0.72 |
> | **EHGNN-F (cond, LR)** | 76.34 ± 0.16 | **95.98 ± 0.01** | 96.91 ± 0.11 | 86.28 ± 0.20 | 78.00 ± 0.06 | 67.56 ± 0.42 | 76.93 ± 0.26 | 87.48 ± 0.09 | 74.77 ± 0.19 | **100.00 ± 0.00** | 58.67 ± 0.05 | **85.56 ± 0.06** | 50.47 ± 0.04 | 97.10 ± 0.01| **32.48 ± 0.19** | 49.87 ± 0.95 |

---

> ### Author Response · Authors · 2025-11-23
> **W4 (continued)**
>
> ## Table B — Peak GPU Memory Comparison (GB)
> *(sparsity = 50%, Bold = lowest memory per dataset; ties allowed)*
>
> | **Models (GB)** | 20News | ModelNet40 | Mushroom | NTU2012 | Actor | CiteSeer | Cora-CA | DBLP-CA | Cora | House | Pokec | PubMed | Twitch | Walmart | Yelp | Trivago |
> |-----------------|--------|-------------|-----------|----------|--------|-----------|-----------|-----------|--------|--------|--------|---------|---------|----------|---------|----------|
> | **HSL** | 0.41 | 0.40 | **0.13** | 0.08 | 0.33 | 0.17 | 0.09 | 1.18 | 0.09 | 0.08 | **0.17** | **0.24** | **0.21** | 10.70 | 11.89 | 5.53 |
> | **EHGNN-F** | **0.31** | 0.29 | 0.18 | **0.06** | 0.30 | **0.13** | **0.08** | 0.89 | **0.08** | **0.05** | 0.18 | 0.33 | 0.22 | 1.93 | 11.51 | 4.55 |
> | **EHGNN-C** | 0.34 | 0.29 | 0.20 | **0.06** | 0.30 | **0.13** | **0.08** | **0.85** | **0.08** | **0.05** | 0.18 | 0.33 | 0.22 | 1.93 | 15.45 | 4.84 |
> | **EHGNN-F (cond.)** | **0.31** | 0.29 | 0.18 | **0.06** | 0.30 | 0.30 | 0.15 | 2.51 | 0.16 | **0.05** | 0.18 | 0.34 | **0.21** | 1.90 | OOM | **4.52** |
> | **EHGNN-C (cond.)** | 0.34 | 0.29 | 0.20 | **0.06** | 0.30 | 0.15 | **0.08** | 0.90 | **0.08** | 0.06 | 0.18 | 0.33 | 0.22 | 1.94 | **10.09** | 4.84 |
> | **EHGNN-F (cond, LR)** | **0.31** | **0.28** | 0.18 | **0.06** | **0.29** | **0.13** | **0.08** | **0.85** | **0.08** | **0.05** | 0.18 | 0.32 | 0.22 | **1.89** | 11.49 | 4.53 |
>
> > No graph-to-hypergraph adaptations + Q9
>
> We did not pursue this direction as clique expansion severely distorts the hypergraph structure and leads to extreme densification. A hyperedge of size $k$ becomes a clique with $k(k-1)/2$ edges, producing graphs that are much more denser than the original hypergraph. This makes graph sparsification computationally infeasible on datasets such as Yelp and Walmart, and, more critically, the resulting sparsifier preserves the spectral properties of the expanded graph, not the hypergraph Laplacian used in HGNNs.

---

> ### Author Response · Authors · 2025-11-23
> **W5, Q8**
>
> > W5 (Node clustering performance on heterophilic hypergraphs) + Q8
>
> Yes. On strongly heterophilous datasets, the unsupervised node-clustering results in Appendix G (Table 12) show that all message-passing HGNNs perform poorly, including our sparsified variants. **This behavior is expected and is not specific to sparsification.** As noted in lines 1228–1233 of the revised manuscript, these datasets violate the smoothness assumptions underlying Laplacian-based propagation: each layer mixes node features with neighbors that belong to different classes, leading to difficulties in clusterability of the learned embeddings. This phenomenon affects the Full HGNN to the same extent as EHGNN-F and EHGNN-C, as all of them share the same Laplacian-style aggregation operator. Sparsification neither increases nor decreases this structural bias.
>
> **This limitation, which we acknowledged, only applies to the unsupervised clustering task.** In the supervised node-classification benchmarks (Tables 2–4), the heterophilous datasets, such as Actor and Pokec, do not show performance degradation due to sparsification. Thus, the limitation is due to the backbone architecture, not the sparsification mechanism. Addressing heterophily would require modifying the aggregation operator itself (e.g., heterophily-aware HGNNs), which is outside the scope of this work.

---

> ### Author Response · Authors · 2025-11-23
> **W6/Q10 , W7/Q7**
>
> > W6 (Parameter Overhead Not Properly Addressed) + Q10
>
> Thank you for this insightful question. Although Table 9 shows higher parameter counts for EHGNN-F (due to learnable scorer matrices), we verified that this overhead has a negligible effect on the runtime, memory, or convergence.
>
> As shown earlier in our response to W1 (Table F - memory breakdown), on the Yelp dataset, **the static model footprint (parameters + buffers) remains 3-6 MB, whereas activation tensors consume 11-20 GB.** Even after the optimizer states for Adam adding an overhead of roughly 2 x #parameters, this is still about 10-12 MB in total. Compared to 11-20 GB activation memory, this overhead is negligible. On extremely large hypergraphs (Yelp), the computation cost is dominated by the size of the incidence tensor size $t$, not by the parameter count.
>
> During our experiments, we observed no overfitting. In particular, across all 16 datasets, the train–validation and train–test gaps remain comparable to Full training, and on large datasets (Walmart, Yelp), accuracy even improves over the Full training.
>
> -----
>
> > W7 (Critical experimental details)
>
> We train the models for 500 epochs on all datasets, except for the newly added Trivago benchmark. On Trivago, we train all the competing models for 2000 epochs. Early stopping is used with patience = 200, based on validation loss.
>
> The sparsity budget is a user-specified control parameter that the sparsifiers treat as a constraint. In practice, this budget is dictated by computational limitations, such as how much of the incidence structure can fit in GPU memory. Our experiments, therefore, evaluate all methods under the same user budget (typically 50%) to ensure a fair assessment.
>
> > Stat. Significance.  (Q4)
>
> We conducted paired t-tests comparing EHGNN-C (k=0.5) with Full across all 16 datasets. The table below summarizes the results.
>
> ## Table F --- Statistical test
> *\* = statistically significant at p < 0.05, ** = highly statistically significant at p < 0.01)*
> | Dataset       | Full (Mean) | EHGNN-C (Mean) | $\Delta$ acc (EHGNN-C−full) | p-value   | Sig. | Better Method      |
> |---------------|-------------|----------------|-------------|-----------|------|---------------------|
> | **Cora**          | 78.23      | 73.18         | −5.05       | 1.47e-05  | **  | Full                |
> | **Actor**         | 64.53      | 77.20         | **+12.67**  | 3.90e-10  | **  | **EHGNN-C (k=0.5)** |
> | **Walmart**       | 94.78      | 95.68         | **+0.90**   | 8.82e-08  | **  | **EHGNN-C (k=0.5)** |
> | **Trivago**       | 61.68      | 51.41         | −10.27      | 2.08e-06  | **  | Full                |
> | **DBLP-CA**       | 90.77      | 89.30         | −1.46       | 2.21e-06  | **  | Full                |
> | **ModelNet40**    | 94.89      | 93.74         | −1.16       | 7.90e-06  | **  | Full                |
> | **20newsW100**    | 81.39      | 78.01         | −3.38       | 2.57e-05  | **  | Full                |
> | **House**         | 73.75      | 84.09         | **+10.34**  | 3.11e-05  | **  | **EHGNN-C (k=0.5)** |
> | **NTU2012**       | 87.67      | 84.65         | −3.02       | 4.52e-05  | **  | Full                |
> | **Mushroom**      | 97.75      | 94.14         | −3.61       | 7.57e-05  | **  | Full                |
> | **Pokec**         | 58.37      | 58.79         | **+0.42**   | 1.55e-04  | **  | **EHGNN-C (k=0.5)** |
> | **Twitch**        | 51.23      | 50.57         | −0.66       | 2.35e-04  | **  | Full                |
> | **Yelp**          | 29.75      | 31.75         | **+2.00**   | 6.30e-03  | **  | **EHGNN-C (k=0.5)** |
> | **PubMed**        | 85.60      | 85.42         | −0.17       | 7.05e-03  | **  | Full                |
> | **Cora-CA**       | 81.48      | 80.50         | −0.97       | 1.09e-02  | *    | Full                |
> | **Citeseer**      | 68.19      | 67.46         | −0.72       | 4.29e-02  | *    | Full                |

---

> ### Author Response · Authors · 2025-11-23
> **W8**
>
> ## W8 (Runtime needs clarification)
>
> Thank you for your attention to this aspect.
>
> > Table 4 shows EHGNN methods often slower than Full on small datasets.
>
> This is expected behavior.  **On small hypergraphs, the incidence count $t$ is already small, so sparsification does not reduce any significant message passing overhead for the downstream HGNN.** However, EHGNN still performs an extra MLP scoring step to decide which incidence pairs to keep, which adds a small fixed overhead, making it slightly slower than Full.
>
>
> > The explanation (mask-learning overhead) contradicts efficiency claims
>
> We have revised section 6.1 (IV) to clarify the fact that our efficiency claim applies only to medium to large-scale hypergraphs, where message passing dominates cost. In these regimes, sparsification reduces the number of active incidences, and EHGNN-F / EHGNN-C match or outperform Full (Table 4; Figure 3c). For small hypergraphs, we do not claim runtime improvements.
>
> > Figure 3c shows gains only on largest datasets, but these are the same ones with memory issues
>
> **After re-examining our implementation, we identified an issue in the way GPU memory usage was measured: the earlier memory numbers were inflated because of how and when memory usage was sampled. We have corrected this by synchronizing CUDA operations and using PyTorch’s peak-memory utilities, which provide consistent, accurate reporting of training time GPU memory usage.**
>
> With the corrected measurement, the updated Table 3 now shows that our proposed sparsifiers-- **EHGNN-C, EHGNN-F, and the low-rank variant EHGNN-F(cond, LR), consistently reduce peak GPU memory compared to full training across all benchmarks**. These variants achieve 15–40% lower memory usage depending on dataset complexity. Furthermore, EHGNN-C(cond) also yields reductions on all datasets except Citeseer.
>
> With this, these datasets are no longer the ones with memory issues. If the reviewer thinks that a complete table presenting all the methods' runtime/epoch on all 16 datasets would be beneficial to the readers, we are open to adding it to the appendix.

---

> ### Author Response · Authors · 2025-11-24
> **W3**
>
> ## W3 (Weak Theory), Q5
>
> > Assumes HGNN is L-Lipschitz w.r.t. Frobenius norm but provides no validation
>
> > Are standard HGNNs (HGNN, HyperGCN, etc.) actually Lipschitz? What is L?
>
> > What is the Lipschitz constant L for standard HGNNs empirically?
>
> In this work, we do not attempt to estimate a tight empirical value of the global Lipschitz constant $L$ for HGNNs, since such estimates are known to be numerically unstable and highly conservative [1]. Instead, we have computed an empirical upper bound on $L$ by estimating the spectral norms of the weight matrices.
>
> Concretely, for an HGNN layer $H^{(l+1)} = \sigma \left( P H^{(l)} W^{(l)} \right),$ the propagation operator $P$ (normalized hypergraph Laplacian) satisfies $\|\|P\|\|_2 \leq 1$ [4].
>
> Thus, an upper bound on the layer Lipschitz constant is $L_l \leq \|\|W^{(l)}\|\|_2$.
>
> We can therefore estimate the global upper bound as $L_{\text{upper}} \approx \prod_{l=1}^{L} \|\|W^{(l)}\|\|_2.$
>
> As expected for unconstrained neural networks (i.e., without spectral normalization on the weights), the per-layer spectral norms grow moderately during training. Therefore, the overall upper-bound $L_{upper}$ also increases over epochs. More importantly, $L_{upper}$ remains finite at all times, confirming that standard HGNNs are indeed Lipschitz continuous with respect to the Frobenius norm. **We empirically found $L_{upper} (\text{Cora}) \approx 38$, $L_{upper} (\text{Citeseer}) \approx 36$ and $L_{upper} (\text{Actor}) \approx 27$ at convergence.**  This behavior is consistent with standard GNNs where no spectral normalization is applied [2, 3].
>
> Our theoretical assumptions require only that $L$ be finite, and the empirical results confirm this. A tighter bound would require spectral normalization, which we did not apply.
>
> [1] Lipschitz regularity of deep neural networks: analysis and efficient estimation, NeurIPS 2018.
>
> [2] Training Graph Neural Networks Subject to a Tight Lipschitz Constraint, TMLR 2025.
>
> [3] Stability Properties of Graph Neural Networks, IEEE Transactions on Signal Processing, 2020.
>
> [4] Learning with Hypergraphs: Clustering, Classification, and Embedding, NIPS 2006.
>
>
> > Assumes gradient signs remain fixed throughout training—highly unrealistic
>
> > How often do gradient signs flip during training? Can you show this empirically?
>
> We agree that assuming perfectly fixed gradient signs throughout training would be unrealistic if interpreted as a property of the full model. **Theorem 5.2 uses this assumption only as a local analytical device to derive a closed-form convergence bound for the mask probabilities, not as a global description of how the entire network trains.** The theorem simply analyzes logit update under the straight-through estimator (STE) and characterizes its behavior when the surrogate gradient maintains a consistent sign.
>
> To evaluate whether this assumption is reasonable, we have computed the gradient sign flip fraction for the mask logits, i.e., the proportion of logits whose gradient sign changes between consecutive steps. If the STE surrogate gradients are unstable or random, we expect $\approx 50%$ sign flips per epoch. If they are stable enough to guide optimization, sign flips should remain low.
>
> We measured the flip fraction for EHGNN-F method on Cora and Actor. Results show consistently low flip rates with the following observations:
>
> - **Cora:** starts near 0.10 in very early epochs and stabilizes at 0.02-0.05, average $\approx$ 3-4%
>
> - **Actor**: Starts near 0.11 in very early epochs and stabilizes around 0.07-0.09, average $ \approx$ 8-9%
>
> - Maximum flips observed across runs: 6% (Cora) and 10% (Actor)
>
> This demonstrates that although some sign changes naturally occur (as expected in any non-convex optimization), the STE surrogate gradient is far from random and maintains coherent directions throughout training.
>
> > The bound shows variance decreases as p→0 or p→1, which is obvious—doesn't explain empirical behavior
>
> We are unclear about what you meant by “explaining empirical behavior”. Perhaps you meant “true” empirical stability of the learned sparsifier models (e.g., EHGNN-F or EHGNN-C) under a stochastically controlled but bounded perturbation to the incidence matrix?

---

> ### Author Response · Authors · 2025-11-24
> **W3 (continued)**
>
> > Missing Theory
>
> Thank you for posing these deep questions; they have opened several interesting directions for us to reflect on. While we are not yet ready to make fully formal statements (and accompanying proofs), we outline our current understanding below.
>
> - *When does task-aware sparsification improve over unsupervised methods?* : Unsupervised sparsifiers aim to preserve structural properties of the hypergraph, but they are agnostic to label-relevant patterns. As a result, they may prune edges that are semantically useful for prediction, and retain edges that are irrelevant or even harmful. Their advantages lie in computational efficiency and the fact that they are completely decoupled from downstream HGNN training. This makes them more useful when labels are scarce or unavailable (e.g., unsupervised settings). In a supervised setting, when sufficient labels are available, task-aware sparsification is preferable to unsupervised, heuristic sparsifiers.
>
> - *What properties of the loss landscape explain why removing edges helps in some cases? Connection between sparsification and generalization bounds*:  Noisy or irrelevant hyperedges mix unrelated nodes and introduces spurious message-passing paths. This injects contradictory information into the aggregation operator. This leads to high-variance, shallow or flat directions in the loss surface and poorly conditioned curvature in the loss surface. All these lead to optimization that is unstable or prone to poor local minima [1,2].
> These issues are further amplified on dense hypergraphs in which nodes participate in many large hyperedges. Empirically, we also observed that sparsification reduces the model’s Lipschitz upper bound (than Full training) at convergence:  $L_{upper} (\text{Cora}) \approx 36$, $L_{upper} (\text{Citeseer}) \approx 32$ and $L_{upper} (\text{Actor}) \approx 18$. A smaller Lipschitz constant implies a smoother, more stable training dynamic, and typically corresponds to better generalization. Finally, Figure 1 shows that sparsification yields more separable embeddings and improved classification margins, suggesting that removing spurious edges reduces over-mixing and preserves discriminative structure.
>
>
> [1] Graph Neural Networks Inspired by Classical Iterative Algorithms, ICML 2021.
>
> [2] Towards Robust Graph Neural Networks for Noisy Graphs with Sparse Labels, WSDM 2022.
>
> ---
> **Please note that we have colored the changes in the revision in blue**

---

### Official Review · Reviewer_1x5r · 2025-11-01

**Soundness:** 2
**Presentation:** 3
**Contribution:** 1
**Rating:** 2
**Confidence:** 5

**Summary:**

This work proposes EdgeMask-HGNN with two distinct masking strategies, i.e., fine-grained node–hyperedge masking and coarse-grained hyperedge-level masking, to address the scalability challenges of HGNNs. Theoretical analyses establish the perturbation stability and the convergence of the method’s mask parameters. Furthermore, the authors conduct comprehensive experiments on fifteen benchmark datasets, demonstrating that EdgeMask-HGNN reduces or maintains memory usage on both small- and large-scale hypergraphs without sacrificing accuracy.

**Strengths:**

1. The manuscript provides theoretical analysis on the proposed method to demonstrate the perturbation stability and convergence of mask parameters.
2. The overall presentation is clear and well-structured. Moreover, the authors provide source code to ensure reproducibility.

**Weaknesses:**

1. The use of edge masking for selective message passing is not novel. Prior studies in GNNs and HGNNs for semi-supervised learning have explored similar ideas, including CO-GNN [1] and HeteHG-VAE [2].
2. Key backbone methods are omitted, such as ED-HNN [3] and SheafHGNN [4].
3. The benchmark datasets are relatively small; the largest include Walmart (88,860 nodes) and Yelp (679,302 hyperedges).
4. For the datasets described as large-scale, the proposed method requires more GPU memory for training than using full hypergraphs, making it difficult to justify the necessity of EdgeMask-HGNN.
5. Furthermore, existing work [5] reports that these datasets can be trained as full hypergraphs on two NVIDIA Tesla P100 GPUs with 12 GB memory each, raising concerns about the necessity of the proposed method.
6. Although the authors provide theoretical analysis on perturbation stability and the convergence of mask parameters, additional experiments are needed to validate the edge masking strategy.
7. Based on the experimental results, the proposed method is more suitable to tackle the heterophilic issue in hypergraphs.

[1] Cooperative Graph Neural Networks. ICML'24.

[2] Heterogeneous Hypergraph Variational Autoencoder for Link Prediction. T-PAMI'21.

[3] Equivariant hypergraph diffusion neural operators. ICLR'23.

[4] Sheaf hypergraph networks. NeurIPS'23.

[5] You are AllSet: A Multiset Function Framework for Hypergraph Neural Networks. ICLR'22.

**Questions:**

See weaknesses.

---

> ### Author Response · Authors · 2025-11-22
>
> We thank the reviewer for his constructive feedback on our work and his appreciation of our write-up.
>
> > W1 (Related works)
>
> We thank the reviewer for highlighting Co-GNN and HeteHG-VAE. We agree that selective masking of message-passing channels is not new. Indeed, early works such as SparseGAT (https://arxiv.org/pdf/1912.00552v1) already used edge-masking, predating both CO-GNN and HeteHG-VAE. Our intention is not to claim the general idea of masking as novel, but rather to clarify that our goals and mechanisms differ substantially from those of prior work.
>
> Co-GNN introduces a cooperative message-passing framework in which each node is treated as a player that chooses actions {STANDARD, LISTEN, BROADCAST, ISOLATE} at each layer; these actions determine whether messages are sent/received along edges. This produces a dynamic computation graph but does not learn incidence-wise masks nor induce a sparsified structure. In contrast, EdgeMask-HGNN (1) operates on hypergraphs, not graphs, (2) learns node–hyperedge incidence (EHGNN-F) or hyperedge (EHGNN-C) probabilities, not node-level actions on graphs, and (3) constructs an explicit task-informed sparsified hypergraph via incidence-wise (or hyperedge-wise) top-k selection that can be fed into any HGNN backbone.
>
> We clarify that HeteHG-VAE does not perform masking or selective message passing. Its “hyperedge attention” component assigns weights to different node types within each hyperedge (e.g., paper, author, Venue, etc), to fuse heterogeneous features into a hyperedge embedding. It does not learn per-incidence masks, does not prune the hypergraph, and does not perform top-k or threshold-based selection. HeteHG-VAE is an **unsupervised VAE** designed to reconstruct the full incidence matrix, not to perform task-driven sparsification. Its goals, granularity, and mechanisms therefore differ fundamentally from our work.
>
> ---
>
> > W2 (Backbone methods)
>
> Thank you for recommending these works. In the following, we evaluate the performance (accuracy and memory reduction) of EHGNN with the ED-HNN backbone. We have also added this backbone in Table 5 of our revision.
>
> As shown in Table C, ED-HNN+EHGNN-F yields substantial accuracy gains on several datasets (e.g., +21.8% on Actor, +5.4% on Cora, +2.7% on House), while maintaining comparable performance on datasets such as ModelNet40 and NTU. More importantly, Table D shows that these improvements come with significant reductions in peak GPU memory usage, ranging from 13–37% on most datasets. Notably, on the large-scale Yelp dataset where ED-HNN runs out of memory, adding EHGNN-F enables successful training with a 33.02% accuracy. This shows that our approach extends the scalability of ED-HNN to hypergraphs where the base model fails to run.
>
> ## Table C —  Accuracy comparison: ED-HNN vs. ED-HNN+EHGNN-F
> *(sparsity = 50%)*
>
> | Models              | Actor              | Cora               | Cora-CA            | House             | ModelNet40          | NTU2012                | PubMed             | yelp|
> |--------------------|--------------------|--------------------|--------------------|--------------------|----------------------|---------------------|---------------------|---------------------|
> | **EDHNN**          | 67.55 ± 0.1       | 78.05 ± 0.51       | **82.39 ± 0.62**       | 96.84 ± 2.4       | **97.42 ± 0.06**         | **89.54 ± 0.61**        | **88.07 ± 0.24**        | OOM|
> | **EDHNN + EHGNN-F** | **82.26 ± 0.16**   | **80.61 ± 0.76**   | 80.56 ± 0.76   | **99.50 ± 0.28**   | 97.38 ± 0.07     | 89.46 ± 0.37    | 87.87 ± 0.26    | **33.02 ± 0.4**
>
> ## Table D —   Peak GPU Memory Comparison (GB): ED-HNN vs. ED-HNN+EHGNN-F
> *(sparsity = 50%)*
>
> | Models      | Actor   | Cora    | Cora-CA | House   | ModelNet40 | NTU2012     | PubMed | yelp |
> |---------------------|---------|---------|---------------------------|---------|------------|---------|---------|---------|
> | **EDHNN (MB)**         | 640.44  | 153.65  | 149.09                    | 158.36  | 860.54     | 160.53  | 771.90  | OOM |
> | **EDHNN + EHGNN-F (MB)**    | 553.01  | 127.68  | 126.66                    | 99.30   | 543.33     | 107.73  | 604.35  | 12907 |
> | **Reduction (MB)**    | **87.43** | **25.97** | **22.43**                | **59.06** | **317.21** | **52.80** | **167.55** | - |
> | **Reduction (%)**     | **13.65%** | **16.90%** | **15.04%**              | **37.28%** | **36.85%** | **32.90%** | **21.71%** | - |
>
> ---

---

> ### Author Response · Authors · 2025-11-22
>
> > W3 (Datasets)
>
> In addition to the datasets reported in the main paper, we have now included results on the much larger Trivago hypergraph, which contains 172,738 nodes and 233,202 hyperedges, substantially larger than Walmart (88,860 nodes) and comparable to Yelp in hyperedge count (see Tables 2-3 in the revision). This experiment demonstrates that our sparsification framework remains both effective and scalable on significantly larger hypergraphs.
>
> As shown in Table E below, EdgeMask-HGNN variants achieve strong accuracy, substantially outperforming unsupervised sparsifiers (Random, Degdist) and staying competitive with the full HGNN, while also achieving **up to 27% peak GPU memory reduction** and **lower runtime per epoch.** Note that, while the newly added SOTA method, HSL [1], achieves the highest accuracy (68.69 ± 1.06), it does so by preserving the full hypergraph during message passing, which results in significantly higher memory and runtime costs (e.g., 2.64 seconds/epoch, compared with 0.15–0.20 seconds/epoch for EHGNN variants).
>
> ## Table E —   Accuracy and Peak GPU Memory Comparison (GB) on Trivago
>
> | Metric | Full       |  Random      |  Degdist       |  Spectral         | HSL    | EHGNN-F    | EHGNN-C    | EHGNN-F(cond) | EHGNN-C(cond) | EHGNN-F(cond,LR) |
> |:--------|:--------------------|:--------------------|:--------------------|:--------------------|:--------------------|:--------------------|:--------------------|:--------------------|:--------------------|:--------------------|
> | **Accuracy** | 61.68 ± 0.28 | 26.78 ± 0.05 | 26.78 ± 0.05 | OOM | **68.69 ± 1.06** | 50.38 ± 1.40 | 51.41 ± 0.72 | 46.22 ± 1.21 | 50.14 ± 0.72 | 49.87 ± 0.95 |
> | **Memory (GB)** | 6.2  | **4.44 (28% ↓)** | **4.44 (28% ↓)** | OOM | 5.53 (11% ↓) | 4.55 (27% ↓) | 4.84 (22% ↓) | 4.52 (27% ↓) | 4.84 (22% ↓) | 4.53 (27% ↓) |
> | **Avg. runtime/epoch (sec)** | 0.27 | **0.11** | **0.11** | OOM | 2.64 | 0.15 | 0.16 | 0.2 | 0.18 | 0.15 |
> | **% sparsified** | 0% | 50% | 50% | OOM | 0% | 50% | 50% | 50% | 50% | 50% |
>
> [1] Cai, Derun, et al. "Hypergraph Structure Learning for Hypergraph Neural Networks." IJCAI. 2022.
>
> ---
> > W4 (GPU memory issue)
>
> We thank the reviewer for pointing out the discrepancy in our original memory and runtime results. After re-examining our implementation, we identified an issue in the way GPU memory usage was measured: the earlier memory numbers were inflated because of how and when memory usage was sampled. We corrected the measurement procedure by synchronizing CUDA operations and using PyTorch’s peak memory utilities, ensuring consistent and accurate GPU memory reporting.
>
> With the corrected measurement, the updated Table 3 now shows that our proposed sparsifiers-- **EHGNN-C, EHGNN-F, and the low-rank variant EHGNN-F(cond, LR), consistently reduce peak GPU memory compared to full training across all benchmarks**. These variants achieve 15–40% lower memory usage depending on dataset complexity. Furthermore, EHGNN-F(cond) , discussed at length in Appendix C, also yields reductions on all datasets except Citeseer. In addition, we are finalizing a more memory-friendly version of EHGNN-C(cond), analogous to the low-rank approach used in EHGNN-F(cond, LR), and will include its results as soon as they are ready.
>
> ---
>
> > W5 (Necessity of EHGNN)
>
> We slightly disagree with the premise of the reviewer's question in this regard. Allset paper actually demonstrated that not all methods are equally scalable. Referring to Table 2 of their paper, we notice that CEGCN, CEGAT, and HAN (full) encounter OOM issues on datasets such as 20News, Walmart, and Yelp. They also evaluated the mini-batch setting of HAN and reported that minibatching harms its performance on Yelp and
> Walmart. They also remarked on Page 9 that **“.. a naive application of standard heterogeneous GNNs on large hypergraphs often fails”**.
>
> Our method benefits such methods that encounter OOM on large-scale hypergraphs, as we showed earlier in our comparison between ED-HNN and. ED-HNN+EHGNN-F.  We also showed that our method can reduce the memory requirement of ED-HNN on small - medium scale datasets by 13-37%. This suggests that even if one can run existing methods on these datasets, our approach can help these methods reduce precious memory resources.
>
> ---
>
> > W6 (Additional experiments to validate edge masking)
>
> We would be grateful if the reviewer could further elaborate on the specific experiments that could help us validate the edge masking strategy.
>
> ---
>
> > W7 (Heterophilic graph)
>
> We agree with the reviewer that the proposed method is indeed more useful in heterophilic hypergraphs than homophilic hypergraphs, as we observed in Table 2. To evaluate this claim, in Appendix F.2, Figure 7, we have demonstrated a strong negative correlation between Node heterophily vs. accuracy gain and Edge heterophily vs. accuracy gain w.r.t Full training.
>
> ---
> **Please note that we have colored the changes in the revision in blue**

---

> ### Author Response · Authors · 2025-11-27
> **Additional experiments we promised (Revised Table E)**
>
> **Additional experiments we promised earlier:** As we mentioned earlier that we were working on a low-rank reparameterization of EHGNN-C(cond) to improve its memory usage. Below, we present the revised Table E to include the results we obtained with this variant.
>
> *Please note the EHGNN-C(cond.) (as reported in Tables 2–4) deploy chunked scoring (chunk size = 32k), an optimization we attempted during rebuttal, to avoid OOM.* This optimization, though effective memory-wise, significantly increases end-to-end training time; for instance, it takes 2230.27 seconds on Yelp compared to 277 seconds by EHGNN-C(cond, LR). Thus, EGHNN-C (cond, LR) is a more balanced alternative to EHGNN-C(cond)- both memory- and runtime-wise.
>
> ## Revised Table E —   Accuracy and Peak GPU Memory Comparison (GB) on Trivago
>
> | Metric | Full       |  Random      |  Degdist       |  Spectral         | HSL    | EHGNN-F    | EHGNN-C    | EHGNN-F(cond) | EHGNN-C(cond) | EHGNN-F(cond,LR) | EHGNN-C(cond,LR) |
> |:--------|:--------------------|:--------------------|:--------------------|:--------------------|:--------------------|:--------------------|:--------------------|:--------------------|:--------------------|:--------------------|:--------------------|
> | **Accuracy** | 61.68 ± 0.28 | 26.78 ± 0.05 | 26.78 ± 0.05 | OOM | **68.69 ± 1.06** | 50.38 ± 1.40 | 51.41 ± 0.72 | 46.22 ± 1.21 | 50.14 ± 0.72 | 49.87 ± 0.95 | 48.18 ± 0.49 |
> | **Memory (GB)** | 6.2  | **4.44 (28% ↓)** | **4.44 (28% ↓)** | OOM | 5.53 (11% ↓) | 4.55 (27% ↓) | 4.84 (22% ↓) | 4.52 (27% ↓) | 4.84 (22% ↓) | 4.53 (27% ↓) | 4.84 (22% ↓) |
> | **Avg. runtime/epoch (sec)** | 0.27 | **0.11** | **0.11** | OOM | 2.64 | 0.15 | 0.16 | 0.2 | 0.18 | 0.15 | 0.16 |
> | **% sparsified** | 0% | 50% | 50% | OOM | 0% | 50% | 50% | 50% | 50% | 50% | 50% |
>
> ---
> *Please note that we have updated tables 2-4 in the revision with results from this newly added variant.*

---

### Official Review · Reviewer_gFef · 2025-11-01

**Soundness:** 2
**Presentation:** 2
**Contribution:** 2
**Rating:** 4
**Confidence:** 4

**Summary:**

In this paper, the authors propose EdgeMask-HGNN, a framework that sparsifies hypergraphs for more efficient hypergraph neural network learning. Specifically, it introduces a fine-grained node–hyperedge masking and a coarse-grained hyperedge-level masking mechanism to reduce hypergraph complexity while preserving task performance. Theoretical analysis demonstrates the stability of model outputs under stochastic masking and the convergence of retention probabilities during optimization. Extensive experiments on multiple node classification benchmarks show that EdgeMask-HGNN achieves comparable or superior accuracy to full HGNNs while significantly improving efficiency.

**Strengths:**

1. The paper is clearly written and easy to follow.

2. The proposed method is thoroughly validated across a wide range of datasets, results in Table 2 robustly demonstrate superiority over baselines (degree, random, spectral).

**Weaknesses:**

1. Several works focus on hypergraph structure learning, such as [1]. A comparison with these methods in the experiments would be valuable.

2. Some studies, such as [2–3], on sparse GNNs perform joint sparsification of both the graph structure and the neural network parameters; these works are related to this study and should be discussed.

3. The proposed masking method has already been well studied in existing works.

[1] Cai, Derun, et al. "Hypergraph Structure Learning for Hypergraph Neural Networks." IJCAI. 2022.

[2] Chen, Tianlong, et al. "A unified lottery ticket hypothesis for graph neural networks." International conference on machine learning. PMLR, 2021.

[3] Liu, Chuang, et al. "Comprehensive graph gradual pruning for sparse training in graph neural networks." IEEE Transactions on Neural Networks and Learning Systems 35.10 (2023): 14903-14917.

**Questions:**

See weaknesses

---

> ### Author Response · Authors · 2025-11-22
>
> Thank you for your comments. We sincerely appreciate your recognition of our paper writing. Here are the responses to the comments.
>
> > W1 (Comparison with HSL)
>
> Thank you for this suggestion. We compared EdgeMask-HGNN against Hypergraph Structure Learning (HSL) as proposed in Cai et al. (IJCAI 2022). HSL is an unsupervised structure-optimization framework that learns a new hypergraph prior to HGNN training. In contrast, EdgeMask-HGNN performs ***task-aware, end-to-end sparsification jointly with HGNN training**. Another significant difference between HSL and our method is in terms of problem setting: HSL does not have a provision for a user-specified sparsity budget, whereas EHGNN does. Thus, we added a regularizer with HSL’s loss function to make the comparison with EHGNN as fair as possible. Tables 2 and 3 in the revision show HSL’s performance.
>
> Below, we provide two tables comparing (1) accuracy, and (2) Memory reduction relative to HSL. These tables demonstrate that EHGNN variants are competitive or superior to HSL across diverse datasets, while offering significant memory reduction on large-scale Walmart, Yelp, and Trivago datasets.
> ## Table A — Accuracy Comparison of HSL vs. EHGNN Variants
> *(sparsity = 50%, Bold = highest accuracy per dataset)*
>
> | **Models** | 20News | ModelNet40 | Mushroom | NTU2012 | Actor | CiteSeer | Cora-CA | DBLP-CA | Cora | House | Pokec | PubMed | Twitch | Walmart | Yelp | Trivago |
> |------------|--------|-------------|----------|----------|--------|-----------|-----------|-----------|--------|--------|--------|---------|---------|----------|---------|----------|
> | **HSL** | **81.85 ± 0.10** | 86.36 ± 1.61 | **99.79 ± 0.03** | 80.45 ± 1.84 | 63.01 ± 0.24 | 62.20 ± 0.24 | 75.82 ± 1.26 | **90.11 ± 0.55** | 74.10 ± 1.62 | 75.85 ± 0.31 | 58.03 ± 0.67 | 78.88 ± 1.16 | **51.15 ± 0.41** | 63.59 ± 0.21 | 24.89 ± 0.03 | **68.69 ± 1.06** |
> | **EHGNN-F** | 77.99 ± 0.07 | 95.25 ± 0.06 | 96.39 ± 0.39 | **87.51 ± 0.22** | 77.00 ± 0.07 | 68.12 ± 0.52 | 79.53 ± 0.55 | 88.11 ± 0.07 | 74.33 ± 0.41 | 87.31 ± 1.58 | 58.78 ± 0.07 | **85.56 ± 0.04** | 50.74 ± 0.05 | 95.17 ± 0.02 | 32.31 ± 0.15 | 50.38 ± 1.40 |
> | **EHGNN-C** | 78.01 ± 0.34 | 93.74 ± 0.10 | 94.14 ± 0.60 | 84.65 ± 0.51 | 77.20 ± 0.07 | 67.46 ± 0.58 | **80.50 ± 0.15** | 89.30 ± 0.07 | 73.18 ± 0.36 | 84.09 ± 1.17 | 58.79 ± 0.04 | 85.42 ± 0.06 | 51.10 ± 0.04 | 95.68 ± 0.01 | 31.75  ± 0.29 | 51.41 ± 0.72 |
> | **EHGNN-F (cond.)** | 78.67 ± 0.06 | 95.59 ± 0.12 | 95.07 ± 0.69 | 87.32 ± 0.43 | **78.73 ± 0.05** | **68.36 ± 0.43** | 76.07 ± 0.66 | 86.27 ± 0.08 | 73.94 ± 0.39 | **100.00 ± 0.00** | **59.17 ± 0.09** | 85.53 ± 0.06 | 50.84 ± 0.06 | **98.37 ± 0.01** | OOM | 46.22 ± 1.21 |
> | **EHGNN-C (cond.)** | 74.63 ± 0.39 | 94.54 ± 0.15 | 97.52 ± 0.12 | 85.45 ± 0.29 | 77.57 ± 0.06 | 67.20 ± 0.57 | 75.51 ± 0.16 | 87.10 ± 0.14 | **76.22 ± 0.35** | 72.14 ± 0.73 | 59.02 ± 0.03 | 85.52 ± 0.05 | 50.67 ± 0.03 | 94.72 ± 0.04 | 29.21 ± 0.36 | 50.14 ± 0.72 |
> | **EHGNN-F (cond, LR)** | 76.34 ± 0.16 | **95.98 ± 0.01** | 96.91 ± 0.11 | 86.28 ± 0.20 | 78.00 ± 0.06 | 67.56 ± 0.42 | 76.93 ± 0.26 | 87.48 ± 0.09 | 74.77 ± 0.19 | **100.00 ± 0.00** | 58.67 ± 0.05 | **85.56 ± 0.06** | 50.47 ± 0.04 | 97.10 ± 0.01| **32.48 ± 0.19** | 49.87 ± 0.95 |
>
> ## Table B — Peak GPU Memory Comparison (GB)
> *(sparsity = 50%, Bold = lowest memory per dataset; ties allowed)*
>
> | **Models (GB)** | 20News | ModelNet40 | Mushroom | NTU2012 | Actor | CiteSeer | Cora-CA | DBLP-CA | Cora | House | Pokec | PubMed | Twitch | Walmart | Yelp | Trivago |
> |-----------------|--------|-------------|-----------|----------|--------|-----------|-----------|-----------|--------|--------|--------|---------|---------|----------|---------|----------|
> | **HSL** | 0.41 | 0.40 | **0.13** | 0.08 | 0.33 | 0.17 | 0.09 | 1.18 | 0.09 | 0.08 | **0.17** | **0.24** | **0.21** | 10.70 | 11.89 | 5.53 |
> | **EHGNN-F** | **0.31** | 0.29 | 0.18 | **0.06** | 0.30 | **0.13** | **0.08** | 0.89 | **0.08** | **0.05** | 0.18 | 0.33 | 0.22 | 1.93 | 11.51 | 4.55 |
> | **EHGNN-C** | 0.34 | 0.29 | 0.20 | **0.06** | 0.30 | **0.13** | **0.08** | **0.85** | **0.08** | **0.05** | 0.18 | 0.33 | 0.22 | 1.93 | 15.45 | 4.84 |
> | **EHGNN-F (cond.)** | **0.31** | 0.29 | 0.18 | **0.06** | 0.30 | 0.30 | 0.15 | 2.51 | 0.16 | **0.05** | 0.18 | 0.34 | **0.21** | 1.90 | OOM | **4.52** |
> | **EHGNN-C (cond.)** | 0.34 | 0.29 | 0.20 | **0.06** | 0.30 | 0.15 | **0.08** | 0.90 | **0.08** | 0.06 | 0.18 | 0.33 | 0.22 | 1.94 | **10.09** | 4.84 |
> | **EHGNN-F (cond, LR)** | **0.31** | **0.28** | 0.18 | **0.06** | **0.29** | **0.13** | **0.08** | **0.85** | **0.08** | **0.05** | 0.18 | 0.32 | 0.22 | **1.89** | 11.49 | 4.53 |
>
> Please note that EHGNN-F (cond,LR) is a lightweight low-rank parameterization of EHGNN-F (cond) that preserves feature-conditioning without the associated memory overhead. We have discussed it at length in Appendix C.

---

> ### Author Response · Authors · 2025-11-22
>
> > W2 (Inclusion of related works)
>
> Thank you for pointing out these relevant works. We have revised the Related Works section to clarify the distinction between our method and sparse-training approaches such as Chen et al. (2021) [2] and Liu et al. (2023) [3].
>
> These works focus on parameter sparsity of GNNs, i.e., pruning the weights of the neural network during training to obtain a compressed or lottery-ticket GNN. Although they sometimes also prune edges of the graph, the primary objective is network parameter compression (reducing FLOPs and model size). In contrast, our work focuses on hypergraph structural sparsification: learning to select a small but informative subset of node, hyperedge incidences (or hyperedges) while keeping the HGNN parameters as it is. Our masks, therefore, operate on the input hypergraph structure, not on the network parameters. Thus, our goal is to retain task-relevant incidence patterns rather than improve parameter efficiency.
>
> We acknowledge that both lines of work fall under the broad umbrella of sparse GNNs, but they act on different components of the model and different structures (graphs vs. hypergraphs).
>
> ---
>
> > W3 (well-studied existing works)
>
> We would be grateful if you could point us to some of these works.
>
> ---
> **Please note that we have colored the changes in the revision in blue**

---

### Author Response · Authors · 2025-12-04
**A summary of our rebuttal and revisions**

We thank the reviewers for their constructive feedback. We also appreciate the efforts of the Area Chair, especially under the changed circumstances of ICLR this year.

In the following, we discuss the concerns raised by the reviewers and summarize how we addressed them:

## 1. Additional Baselines and Clarifying Novelty

**Concerns:**

- Reviewer gFef: Missing comparison to HSL baseline.

- Reviewer 1x5r: Edge Masking for selective message passing is “not novel” due to works such as CO-GNN, HeteHG-VAE. Key backbone methods such as  ED-HNN, and SheafHGNN should be considered.

- Reviewer voUi: Better Spectral method implementation needed. Lack of other learnable/task-aware method.

**Our Actions:**
- New baseline: Added comparison with HSL (Reviewer voUi, gFef)
- New backbone: Added ED-HNN backbone experiments (Reviewer 1x5r)
- Clarified novelty: EdgeMask-HGNN performs task-aware structural sparsification, distinct from (i) parameter sparsification (Reviewer gFef), (ii) action-based message passing in CO-GNN (Reviewer 1x5r), and (iii) heterogeneous hyperedge attention in HeteHG-VAE (Reviewer 1x5r).
- We implemented an approximate spectral sparsifier, whose accuracy and memory consumption on large-scale hypergraphs are still inferior to EHGNN-C.

-----------

## 2. Scalability, Dataset Size, and Experimental Adequacy

**Concerns:**

- Reviewer 1x5r: Benchmark datasets are relatively small

- Reviewer g7jz: Need to show stronger scalability on larger datasets (>100k nodes).

**Our Actions:** Added experiments on the much larger Trivago hypergraph (172k nodes, 233k hyperedges). EHGNN achieves 22-27% memory reduction, lower runtime/epoch than full (0.15-0.20 s), and competitive accuracy.

--------

## 3. Peak memory usage in support of the scalability claim.

**Concerns:**

- Reviewer 1x5r: The proposed method requires more GPU memory than using full hypergraphs.

- Reviewer g7jz: The proposed method often requires more training time and GPU memory.

- Reviewer voUi: Fundamental Contradiction in Core Contribution. Requested to provide a detailed memory breakdown (parameters, activations, gradients, optimizer states), explaining why memory increases on dense hypergraphs (e.g., Yelp)

**Our Actions:**
After re-examining our implementation, we identified an issue in the way GPU memory usage was measured, due to which the earlier reported memory numbers were inflated. We corrected the measurement procedure and ensured consistent and accurate GPU memory reporting.
- With the corrected measurement, EHGNN-C, EHGNN-F, and the low-rank variant EHGNN-F(cond, LR), EHGNN-C(cond, LR) consistently reduce peak GPU memory compared to full training across all benchmarks. These variants achieve up to 49% lower memory usage depending on dataset complexity.
- A detailed memory breakdown (Table F in the rebuttal) confirms that the increased memory on Yelp comes exclusively from activation memory, not parameter memory.
------

## 4. Theoretical Clarifications

**Concerns:**

- Reviewer voUi: Justifying the assumptions behind Theorems 1 and 2. Theorem 1 => Validation for L-Lipschitzness. Are standard HGNNs (HGNN, HyperGCN, etc.) actually Lipschitz? What is L? Theorem 2 => Why does it assume that gradient signs remain fixed throughout training? How often do gradient signs flip during training? Can it be shown empirically?

- Reviewer 1x5r, voUi: Provide stronger justification for mask convergence and stability.

**Our Actions:**
- We clarified that Theorem 1 requires only the existence of a finite Lipschitz constant, not a tight bound. We provided empirical confirmation that standard HGNNs satisfy this assumption (upper bound for L for HGNN on Cora $\approx$ 38, Citeseer $\approx$ 36, and Actor $\approx$ 27).
- We explicitly showed that the gradient-sign-consistency assumption holds approximately and sufficiently in practice, and only locally (not globally). The empirical gradient sign flip rates ($\approx$ 3-4% on Cora, and $\approx$ 8-9% on Actor) support the validity of the theorem’s assumptions.
- In Appendix K, we showed empirical convergence and stability of the learned masks on Cora, Actor, and Yelp.  In particular, we showed that \%mask flips across consecutive epochs rapidly decrease after the initial few epochs and stabilize to very low values ($< 1 \%$). This shows that the learned discrete masks converge to a stable sparsification pattern.
--------

## 5. Model Architecture improvement

**Concerns:**

- Reviewer 1x5r + voUi: Pointed out that Feature-conditioned variants appear memory-heavy.

**Our Actions:** We introduced low-rank feature-conditioned variants (EHGNN-F(cond,LR), EHGNN-C(cond,LR)) that drastically reduce memory while preserving accuracy.

---

> ### Author Response · Authors · 2025-12-04
> **A summary of our rebuttal and revisions (cont.)**
>
> ## 6. Additional Insights on Heterophily
>
> **Concerns:**
> Reviewer 1x5r: Suggested that the proposed method seems more useful for heterophilic hypergraphs.
>
> **Our Actions:** Added heterophily analysis in Appendix F.2 (Figure 7), showing strong negative correlation between homophily ratio and accuracy gains.
>
> ------
>
> ## 7. Miscellaneous Concerns
>
> > Reviewer gFef:  Discussion on GNNs that perform joint structure and NN parameter sparsification should be added.
>
> **Our Actions:** We have revised the Related Works section to discuss these works.
>
> > Reviewer g7jz:  Clarify in the paper how our sampling process works and does not cut the gradient flow.
>
> **Our Actions:** We have added a Discussion on differentiability in section 4.
>
> > Reviewer 1x5r: Questions the necessity of EdgeMask-HGNN, referring to the “You are Allset” paper.
>
> **Our Actions:** -  We pointed out that many HGNNs (e.g., CEGCN, CEGAT, HAN) runs out of memory on Walmart, Yelp, as confirmed in AllSet. EdgeMask-HGNN enables such HGNNs to run on large-scale hypergraphs, where they otherwise fail.
>
> > Reviewer voUi: A principled discussion on when sparsification helps vs. hurts. Analysis of hypergraph properties (density, edge size distribution, homophily) correlated with success. A clear decision criteria for practitioners choosing between variants.
>
> **Our Actions:**
> Appendix J includes a principled discussion on choosing variants. Appendix F.3 includes a correlation analysis of the accuracy gain by EHGNN with respect to 8 hypergraph properties: edge homophily ratio, node homophily ratio, average edge size, hypergraph density, sparsity, the #nodes, #hyperedges, and #incidences.
>
> > Reviewer voUi:  Heterophilic Graphs: Acknowledged but Unresolved/ Poor performance on node-clustering task.
>
> **Our Actions:**
> We highlighted that this poor performance only applies to the unsupervised clustering task. In the supervised node-classification benchmarks (e.g., Actor and Pokec), performance does not degrade due to sparsification (see Table 2). Hence, the limitation is due to the backbone architecture, not the sparsification mechanism. Addressing unsupervised task performance on heterophilic hypergraphs would require modifying the aggregation operator itself (e.g., heterophily-aware HGNNs), which is outside our scope.
>
> > Reviewer voUi: Parameter Overhead Not Properly Addressed/ Discuss: Why the parameter size disparity between EHGNN vs Full training does not significantly affect training time and convergence.
>
> **Our Actions:**
> We highlighted that on the Yelp dataset, the static model footprint (parameters + buffers) remains 3-6 MB, whereas activation tensors consume 11-20 GB. Even with Adam’s roughly 2x overhead per parameter, this is still about 10-12 MB in total. Compared to 11-20 GB activation memory, this overhead is negligible. On extremely large hypergraphs (Yelp), the computation cost is dominated by the size of the incidence tensor size, not by the parameter count. Thus, this overhead has a negligible effect on the runtime, memory, or convergence.
>
>
>
> > Reviewer voUi: Clarification on reported runtime/epochs.
>
> **Our Actions:** We have revised section 6.1 (IV) to clarify the fact that our efficiency claim applies only to medium to large-scale hypergraphs, where message passing dominates cost. In these regimes, sparsification reduces the number of active incidences, and EHGNN-F / EHGNN-C match or outperform Full (Table 4; Figure 3c). We clarified that on small hypergraphs, we do not claim runtime improvements.
>
> > Reviewer voUi: Statistical significance tests.
>
> **Our Actions:**
> Appendix I now includes statistical significance tests.

---

### Meta-Review · Area_Chair_wLxw · 2026-01-07

**Summary:**

This paper proposes EdgeMask-HGNN, a task-aware sparsification framework for HGNNs with fine- and coarse-grained masking, supported by theoretical analysis and extensive experiments.

Reviewers found the paper generally well written, with thorough evaluations, ablations, and reproducibility efforts. However, they raised serious concerns about limited novelty relative to prior sparsification and structure-learning work, missing key baselines, weak and unrealistic theoretical assumptions, and contradictions between the scalability claims and experimental results.

**Reviewer Concerns:**

The incremental novelty, unrealistic theoretical assumptions, and lack of clear guidance on when the method is effective substantially limit the contribution.

**Reviewer Scores:**

There is not much discussion between reviewers and authors. Reviewers generally keep their score.

---

### Decision · Program_Chairs · 2026-01-26

Reject